# Characterization of AVHRR global cloud detection sensitivity based on CALIPSO-CALIOP cloud optical thickness information: Demonstration of results based on the CM SAF CLARA-A2 climate data record

Karl-Göran Karlsson[1], Nina Håkansson[1]

[1]Swedish Meteorological and Hydrological Institute, Folkborgsvägen 17, 601 76 Norrköping, Sweden

*Correspondence to*: Karl-Göran Karlsson (Karl-Goran.Karlsson@smhi.se)

**Abstract.** The sensitivity in detecting thin clouds of the cloud screening method being used in the CM SAF cloud, albedo and surface radiation dataset from AVHRR data (CLARA-A2) cloud climate data record (CDR) has been evaluated using cloud information from the Cloud-Aerosol Lidar with Orthogonal Polarization (CALIOP) onboard the CALIPSO satellite. The sensitivity, including its global variation, has been studied based on collocations of AVHRR and CALIOP measurements over a ten-year period (2006-2015). The cloud detection sensitivity has been defined as the minimum cloud optical thickness for which 50 % of clouds could be detected, with the global average sensitivity estimated to be 0.225. After using this value to reduce the CALIOP cloud mask (i.e., clouds with optical thickness below this threshold were interpreted as cloud-free cases), cloudiness results were found to be basically unbiased over most of the globe except over the polar regions where a considerable underestimation of cloudiness could be seen during the polar winter. The overall probability of detecting clouds in the polar winter could be as low as 50 % over the highest and coldest parts of Greenland and Antarctica, showing that also a large fraction of optically thick clouds remains undetected here. The study included an in-depth analysis of the probability of detecting a cloud as a function of the vertically integrated cloud optical thickness as well as of the cloud's geographical position. Best results were achieved over oceanic surfaces at mid-to-high latitudes where at least 50 % of all clouds with an optical thickness down to a value of 0.075 were detected. Corresponding cloud detection sensitivities over land surfaces outside of the polar regions were generally larger than 0.2 with maximum values of approximately 0.5 over the Sahara desert and the Arabian Peninsula. For polar land surfaces the values were close to 1 or higher with maximum values of 4.5 for the parts with the highest altitudes over Greenland and Antarctica. It is suggested to also quantify the detection performance of other CDRs in terms of a sensitivity threshold of cloud optical thickness which can be estimated using active lidar observations. Validation results are also proposed to be used in Cloud Feedback Model Intercomparison Project (CFMIP) Observation Simulation Package (COSP) simulators for cloud detection characterisation of various cloud CDRs from passive imagery.

## 1 Introduction

Monitoring the global amount, distribution and optical properties of clouds is increasingly important as a result of the increasing evidence that the parametrization of cloud processes and cloud-aerosol interactions including related climate feedbacks, are critical contributors to the uncertainty in climate change analysis and in predictions from climate models (Stocker et al., 2013). However, it is encouraging in this respect to note the steadily increasing amount of observations from spaceborne passive and active sensors (an excellent overview is available at https://www.wmo-sat.info/oscar/) and the prolonged growth of the observational records from the initial satellite sensors launched in the 1970's. These early satellite observations, which consist of spectral radiance measurements, can be used to retrieve information about clouds and other relevant Earth-Atmosphere parameters. Most importantly they have now evolved into time series of observations with lengths approaching four decades, which qualifies them for use as climate data records (CDRs). Examples of CDRs built

upon such observations are described by Rossow and Schiffer (1999), Karlsson et al. (2017), Heidinger et al. (2014) and
Stengel et al. (2017).

The advantage of using satellite-based observations for climate analysis is their global coverage. A similar coverage is very difficult to achieve with surface-based observations alone because of the sparsity of the surface-based observational network. This is particularly true for observations of cloudiness and cloud properties, where large parts of the Earth, especially
oceanic and polar regions, are still poorly covered. The different observation capabilities and conditions for space-based sensors and surface-based observations also leads to problems when trying to characterise the accuracy of space-based CDRs. Although the quality of observations may be estimated for selected Earth positions or for small regions with dense surface networks, it is very difficult to achieve a representative and homogenous view of the accuracy over the entire globe using surface observations. The quality of CDRs is especially important as observations used for climate monitoring must be
very accurate to allow the reliable estimation of potential climate change signals (Ohring et al., 2004), which is a central aspect in the planning and definition of the global climate observing system (Dowell et al., 2013). For this reason, there is also a  need to become more stringent in the description of the uncertainty of CDRs by following international metrological norms (Merchant et al., 2017).

One solution for achieving both the global coverage and an improved quality description is to make use of high-quality reference measurements from space-borne platforms (Dowell et al., 2013). This has already been successfully demonstrated by utilizing data delivered by the A-Train satellites, i.e., Afternoon Satellite Constellation or sometimes referred to as the Afternoon Train). This is a system of satellites operating in the same orbit configuration and having close to simultaneous observation times (Stephens et al., 2002).  The most important satellite in the A-train for the detection of clouds is the
CALIPSO satellite, which has the Cloud-Aerosol Lidar with Orthogonal Polarization (CALIOP) onboard (Winker et al., 2009). The sensitivity of CALIOP to clouds in the atmosphere is much higher than for other space-based sensors and this makes it a natural reference for evaluating the cloud detection efficiency in data records compiled from passive sensor data (e.g., as demonstrated by Heidinger et al., 2016).

This paper presents a detailed CALIOP-based evaluation of the cloud detection efficiency and the uncertainty of the cloudiness information provided by the CLARA-A2 (The CM SAF cloud, albedo and surface radiation dataset from AVHRR data - second edition) CDR (Karlsson et al. 2017). This CDR was released in 2017 by the Climate Monitoring Satellite Application Facility (CM SAF); a project belonging to the satellite ground segment of the European Organisation for the Exploitation of Meteorological Satellites (EUMETSAT, Schulz et al., 2009). The evaluation presented is based on an
original validation method described by Karlsson and Johansson (2013) which has been extended with several new features. The method was first updated to use the latest revision of the CALIPSO-CALIOP dataset (Version 4) and results showing the impact of this change are presented.  The study then takes advantage of the greatly extended CALIOP observation period (here covering almost 10 years) to monitor globally averaged cloud conditions in unprecedented detail. The achieved validation results, which cover approximately one third of the CLARA-A2 observation period, can be considered to be the
best currently available characterisation of the global quality of the CLARA-A2 cloud data record. A specific enhancement of the original validation method is the estimation of the geographical distribution of cloud detection probability as a function of cloud layer optical thickness. Section 2 describes the CLARA-A2 and CALIPSO datasets, Section 3 outlines the extended validation method and the compiled validation dataset and is followed by results in Section 4. Section 5 discusses the results and Section 6 provides conclusions and proposes potential future applications.

## 2 Data

### 2.1 The CLARA-A2 climate data record.

CLARA-A2 is constructed from historic measurements of the Advanced Very High Resolution Radiometer (AVHRR) operated onboard polar orbiting NOAA satellites and the Metop polar orbiters operated by EUMETSAT since 2006. AVHRR measures radiation in five spectral channels (two visible and three infrared channels) with an original horizontal field of view (FOV) size at nadir of 1.1 km. The data used in CLARA-A2 is a resampled version of these measurements at a reduced resolution of5 km, defined as global area coverage (GAC). The size is defined in this context as the approximate diameter (assuming a circular or elliptic shape) of the FOV and this definition will be used throughout this paper. Only resampled GAC data is available (i.e., being archived) globally over the full period since the introduction of the AVHRR sensor in space. The resampling of original data into GAC representation means that four out of five original FOVs are selected for the first scan line while the next two scan lines are ignored. Radiances for these four selected FOVs are then averaged and used to represent the GAC FOV consisting of 15 original full resolution FOVs. Thus, only about 25 % of the nominal GAC FOV is actually observed (see also visualization in Figure 1 in Section 3.2).

CLARA-A2 improves and extends the first version of the data record released in 2012 (Karlsson et al., 2013) and now covers a 34-year time period (1982-2015). Original visible radiances were inter-calibrated and homogenised, using MODIS (Moderate Resolution Imaging Spectroradiometer) data as a reference, before generating each component of the CLARA-A2 product portfolio. The inter-calibration was based on the method introduced by Heidinger et al. (2010), which has now been updated (MODIS Collection 6) and extended (six years have been added). This updated calibration is described by Devasthale et al. (2017). CLARA-A2 features of the following cloud products: cloud mask/cloud amount, cloud top temperature/pressure/height, cloud thermodynamic phase, and (for liquid and ice clouds separately) cloud optical thickness, particle effective radius and cloud water path. These cloud products are available as monthly and daily averages in a 0.25 by 0.25 degree latitude-longitude grid and also as daily resampled global products (Level 2b) in a 0.05 by 0.05 degree latitude-longitude grid. The daily resampled products are valid per satellite and orbit node (ascending or descending) while the daily average product is an average of all available daily resampled products and the monthly products are the averages of all the daily average products. Cloud parameter results are also presented as multi-parameter distributions (i.e., joint frequency histograms of cloud optical thickness, cloud top pressure and cloud phase) for daytime conditions. As well as cloud products CLARA-A2 also includes surface radiation budget and surface albedo products and examples of the CLARA-A2 products can be found in Karlsson et al. (2017).

In this study, we focus exclusively on the quality of the original AVHRR GAC cloud mask because of its central importance to the quality of all other CLARA-A2 products. Validation results for other CLARA-A2 products can be found in Karlsson et al. (2017) and in CM SAF 1 (2017). The CLARA-A2 cloud mask is generated using an improved and extended version of the method first proposed by Dybbroe et al. (2005) which enables reliable processing of the historic AVHRR GAC record. These improvements are described in detail in Karlsson et al. (2017) and in CM SAF 2, 2017).

### 2.2 The CALIPSO-CALIOP cloud information.

An extensive description of the existing CALIPSO-CALIOP cloud and aerosol datasets can be found in Vaughan et al. (2009). In short, the Cloud Layer product from CALIOP (denoted CLAY) used in this study provides information on up to 10 individual vertically displaced cloud layers. As the detection of a cloud layer requires that all layers above that layer are optically thin enough to allow the lidar signal to penetrate down to that particular layer, there can be a bias in the number of layers observed if overlaying clouds are optically thick. The CLAY product is provided in three different horizontal

resolutions (along track): 333 m ("single shot"), 1 km and 5 km. The resolutions coarser than 333 m are constructed through averaging over several single shots. This is done to increase the signal to noise ratio to allow detection of thinner clouds than could be achieved at the original single shot resolution. Thus, CALIOP products at coarser resolution will be capable of detecting more clouds than at finer resolutions and it is preferable that studies of thin Cirrus clouds should be based on products in the coarsest resolution 5 km (Vaughan et al., 2009). Note that the nominal single shot FOV size does not correspond to the true lidar FOV size but rather to the along-track sampling distance. As the true lidar FOV size is only 70 m (Winker et al., 2007), less than 5 % of the nominal single shot FOV is actually observed (see also Fig. 1 in Section 3.2).

An estimation of the cloud optical thickness of each layer is also provided but only for a FOV size of 5 km. However these values are only reliable for clouds with relatively low optical thickness (below approximately 3), because of signal saturation in optically thick clouds (Vaughan et al., 2009 and Sassen and Cho, 1992). In this study we have used the optical thickness interval 0-5 because the new CALIPSO CLAY dataset version 4.10 provides slightly increased cloud optical thickness values compared to previous versions. We interpret this change to represent underestimation in previous values. Despite this change there is still a high degree of uncertainty in values near the upper end of these limits and these may, in reality, include some clouds which are optically thicker.

The CALIPSO satellite follows the A-Train track in a sun-synchronous orbit with an equator-crossing local time of 01:30. This means that observations from the NOAA satellites can be matched to CALIPSO-CALIOP data in near-nadir conditions for a full orbit if they are in an orbit with the same or very close to the same equator-crossing time. For all other NOAA satellite orbits (and also the Metop satellites), matchups are only possible at high latitudes close to 70 degrees N/S. Since CALIPSO is operated in a slightly lower and faster orbit than the NOAA/Metop satellites (i.e., orbital period of CALIPSO is 98.5 minutes while NOAA/Metop period is 102 minutes), close matchups in time are found with a recurrence of approximately 2 days.

In this study, we have used the fourth reprocessed version of the CALIOP CLAY datasets (version 4.10), which was released in 2016. The main features of this updated version are described at
https://www-calipso.larc.nasa.gov/resources/calipso_users_guide/qs/cal_lid_l2_all_v4-10.php.

Regarding the basic CALIOP cloud mask, the most relevant changes affecting this study are
1.  Revised and improved basic cloud-aerosol-discrimination method
2.  Removal of mis-classifications of aerosols and dust as clouds at certain locations at high latitudes (as discussed by Jin et al., 2014)
3.  Inclusion of information on single shot cloud detection in the 5 km dataset, the implications of which are discussed in Section 3.3).

## 3 Validation analysis methods and datasets

### 3.1 Some theoretical considerations about clouds

Cloudiness is not an absolute well-defined quantity like other cloud properties or most other geophysical parameters. Firstly, it depends on the scale of interest, i.e., the areal extent over which cloud cover has to be calculated needs to be specified. Secondly, and perhaps more importantly, a definition of what is meant by a cloud is required to allow a subsequent quantitative use of the results. For example, how optically thin or thick should a cloud be to be called a cloud? This threshold is important when studying the cloud impact on components of the radiation budget (Charlson et al., 2007 and Barja and

Antuña, 2011). How to define clouds detected in satellite imagery is also related to the scale of individual clouds (Koren et al., 2008). The cloud definition aspect is often missing in studies describing various cloud data records. Typically, products and validation results are presented without any deeper discussion on for what clouds the results are really valid. A good example is found in comparison studies between satellite-derived and manual surface-observed cloudiness (e.g., Sun et al., 2015). Results from such studies are difficult to interpret because of the different observation geometries for the compared datasets and the lack of an objective and clear definition of the clouds being observed in either of the two datasets. Because of this ambiguity it has often been recommended to use parameters other than cloudiness or cloud cover (as mentioned in WMO 1, 2012) to instead describe the effect of clouds (e.g. "cloud albedo", "effective cloud cover" or "joint histograms of cloud top pressure and cloud optical thickness") in climate analysis and climate model evaluation studies. Nevertheless, the need to get the geographical distribution of modelled clouds correct is still a crucial requirement (as pointed out in WMO 1, 2012), particularly when considering that parameters describing the effect of clouds are still critically dependent on how you define the underlying cloud or cloud mask. This calls for continued studies of cloud cover from both the observational and modelling perspective. We claim here that the access to high-quality reference cloud observations from CALIPSO-CALIOP may help us to take a significant step forward regarding the use of a more strict quantitative definition of cloudiness. A detailed characterization of the clouds we are observing can be made using CALIOP data. Thus, the ability to observe similar clouds in data records based on passive imagery can then be assessed, which will augment the usefulness of these data records. The following sub-sections outline a new approach which will enhance the value of results from such cloud validation studies.

## 3.2 Basic CALIOP matching method and matching geometry

The underlying method for matching the two cloud datasets is described in detail by Karlsson and Johansson (2013). However, because of the importance for the understanding of method extensions and the achieved results in this study, we repeat here the most important aspects:

1. Positions where the orbital tracks cross are identified for the orbits of the two datasets to be collocated.
2. If the time difference of the two observations at the crossing point is within a certain maximum time difference $T_{diffmax}$ the observations at this position are denoted Simultaneous Nadir Observations (SNOs). Only orbits with SNOs satisfying the maximum $T_{diffmax}$ criterion are selected for further collocation studies. A $T_{diffmax}$ value of 45 seconds has been used in this study. As a consequence of a slightly shorter orbital period for the CALIPSO satellite, collocations could then be made with an approximate two-day repeat cycle.
3. For NOAA satellites flying in an afternoon orbit (which is similar or almost similar to the orbit of CALIPSO), it is possible to compare observations also before and after the SNO point since both satellites continue to observe the same points on Earth close in time. For example, if using a maximum observation time difference of 3 minutes, almost all observations during an entire orbit along the CALIPSO track can be inter-compared. Not all observations from the NOAA satellite afternoon orbits will be made in nadir conditions but relatively close to nadir (i.e., within 15 degrees). The current study has used afternoon orbit data with an observation time difference to CALIOP of 3 minutes to ensure global coverage.
4. For NOAA and METOP satellites flying in a morning orbit, the orbital tracks will cross almost perpendicularly and SNOs will then only occur at high latitudes (near 70 degrees N/S). A consequence of this is that collocations can only be made over distances limited by the AVHRR swath width. Furthermore, all individual collocations will then have varying AVHRR viewing angles along the matched track. Matchups with morning satellite data are not included in this study because of the limited geographical coverage.

In order to better understand the effects of different sensor sampling conditions and the collocation geometry, Fig. 1 shows
an idealised representation of CALIOP collocations with AVHRR GAC data for both afternoon and morning orbits. The
figure is idealised in the sense that it shows the perfect collocation, i.e., a collocation where the centre positions of both GAC
and CALIOP FOVs are perfectly matched. We repeat that the AVHRR GAC sampling means that four out of five original
FOVs are selected for the first scan line (marked as blue filled circular FOVs in Fig. 1) while the next two scan lines are
ignored (empty blue boxes in Fig. 1). Radiances for these four selected FOVs are averaged and then used to represent the
entire GAC FOV consisting of 15 original full resolution FOVs (schematically described as 3x5 blue boxes in Fig. 1). The 5
km CALIOP FOV observation is represented as an array of 15 original 333 m resolution red boxes in Fig. 1. Notice that the
true FOV of the CALIOP sensor is smaller in size. In Fig. 1 they are represented as red filled circles with 70 m size and
separated by 333 m distances. The 5 km CALIOP cloud observation is composed through averaging over the 15 original
measurements but also from averaging over measurements outside of the nominal 5 km distance. This is done to detect
optically very thin clouds (cirrus clouds) which could not be detected solely from data within the nominal 5 km FOV (as
described by Vaughan et al., 2009).

The different panels for afternoon and morning orbit collocations in Fig. 1 are meant to illustrate how collocation conditions
change from the along-track collocation mode for afternoon orbits to the across-track collocation mode for morning orbits.
As a contrast to afternoon satellites, the orbital tracks crosses then almost perpendicularly between CALIPSO and morning
orbit satellites, explaining the shift to a horizontal instead of a vertical orientation of the array of CALIOP measurements in
Fig. 1. The effects of the limited coverage of true AVHRR observations within the nominal GAC FOV and the different
orientations of the array of CALIOP FOVs for morning and afternoon satellites can be ignored if cloud elements have scales
larger than 5 km. However, for cases with smaller scale (sub-pixel) cloud elements or cases with cloud edges within the
GAC FOV, we can expect differences between AVHRR and CALIOP observations. The implications because of this for the
collocation and validation results will be discussed further in Section 5.

As explained by Karlsson and Johansson (2013), binary cloud masks for 5 km FOVs from AVHRR and CALIOP are inter-
compared and evaluated using a range of standard validation scores. However, prior to comparison, the content of the
original 5 km CALIOP FOV observation is adjusted to be consistent with the corresponding cloud mask defined at 1 km
resolution. This check was introduced after noting that global CALIOP-estimated cloudiness for individual orbits was not
always increasing when switching from the 1 km resolution dataset to the 5 km resolution dataset. Conceptually, cloudiness
should increase for the 5 km datasets as it is better able to detect also the optically thinnest cloud layers in addition to those
cloud layers detected at finer resolutions (Vaughan et al., 2009). However, a non-negligible fraction of cases (~ 3-5 % of all
investigated cases in a preparatory study) actually showed lower cloud amounts for the 5 km resolution. This inconsistency
comes as a side effect of the actual method used for creating the coarser resolution CALIOP datasets (Vaughan et al., 2009
and David Winker, CALIPSO Science Team, 2017, pers. comm.). Prior to performing the horizontal averaging of the
CALIOP scattering signal over several single shots, some single shot views are excluded from the analysis if they contain
strongly reflecting boundary layer clouds or aerosols. In the vast majority of cases, the number of these removed single shots
is less than 50 % of all single shot measurements within the 5 km FOV. Considering the official 5 km FOV CALIOP cloud
mask, this procedure would then still justify labelling of the 5 km FOV as cloud free if no other cloud layers are detected.
However, in some areas the frequency of small-scale convective clouds may be high and for these cases this could lead to
underestimated cloudiness in the 5 km products. Another important aspect is that strongly reflecting clouds on the sub-pixel
scale of AVHRR GAC data may still be detectable because of non-linear radiance contributions (with similarities to the "hot
spot" effect from fires) in the short-wave infrared channel at 3.7 µm (Saunders and Grey, 1985, and Saunders, 1986). Thus,
to not include these clouds in the CALIOP datasets might lead to too low or non-representative validation scores for some of

the investigated cases. Karlsson and Johansson (2013) showed that validation scores also improved for AVHRR-based cloud products when adding clouds from the 1 km datasets if 3 or more of the 1 km FOVs within the 5 km FOV were cloudy in cases when the original 5 km products were deemed cloud-free. For these added clouds from 1 km data, the 5 km cloud

optical thickness (not estimated in CLAY 1 km data) was set to 5, i.e., at the maximum upper end of realistically estimated cloud optical thicknesses. This is a justifiable approach as these clouds are by definition strongly reflecting and in most cases would lead to effective cloud optical thicknesses close to or above 5.

3.3 Adaptation to CALIPSO version 4 CLAY products

An important  objective of this study was to verify that the method used by Karlsson and Johansson (2013) would still be applicable to the new version 4 of the CALIOP CLAY product released in 2016 and to investigate whether the validation results changed in any systematic way. Despite the implemented modifications (mentioned at the end of section 2.2), the fundamental retrieval method for the CALIOP CLAY product has remained the same. Consequently, the above mentioned inconsistencies between fine and coarse resolution CALIOP datasets are likely to remain and would need a similar post-

processing adjustment as for previous version 3 products. However, the new version of the 5 km CALIOP cloud product (i.e., in this study we have used the standard CLAY product version 4.10) has been expanded to include full information on the single shots removed during the averaging process. Thus, the previous use of 1 km data in the method by Karlsson and Johansson (2013) could in principle be abandoned and replaced by the direct use of this single shot removal information (the latter method to be called "modified method" in the following). Another improvement found in the version 4.10 dataset is

that the removed single shot FOVs have also been labelled as being either cloudy or filled with thick aerosols. This separation was not available in version 3 where all removed single shot FOVs were assumed to be cloudy. An inter-comparison of version 3 and version 4 products is presented in section 4.1.

**3.4 Applied validation concept and validation scores**

Compared to the previous study by Karlsson and Johansson (2013) this study has access to CALIOP data for a much longer validation period; almost 10 years (2006-2015).  This means that it is now possible to calculate the geographical distribution of validation results, in addition to global mean conditions. Due to a sufficiently large amount of AVHRR-matched nadir looking CALIOP observations it is possible, for the first time, to evaluate the quality of a cloud CDR in a (close to) homogeneous way over almost the entire globe with the only exception being close to the poles where CALIOP

measurements are not available. Consequently, the validation results calculated in this paper are presented as global maps rather than as tables and figures with global mean values. For the plotting of these global maps the results have been rearranged and calculated using a Fibonacci grid with 28878 grid points evenly spread out around the Earth approximately 75 km apart. The resulting grid has almost equal area and almost equal shape of all grid cells making it preferable to traditional latitude-longitude grids which often introduce distortions near the poles. For further details on Fibonacci grids,

see González (2009) and Swinbank and Purser (2006).

We have used the same set of validation scores as those described and defined by Karlsson and Johansson (2013), namely:

-    Mean error (bias) of cloud amount (%), describing the systematic error of the mean

-    Bias-corrected Root Mean Square Error (RMS) of cloud amount (%), describing the random error of the mean
-    Probability of Detection ($0 \leq POD \leq 1$) for both cloudy and cloud-free conditions relative to all observed cloudy or clear cases

- False Alarm Rate ($0 \leq$ FAR) $\leq 1$) for both cloudy and cloud-free conditions relative to all predicted cloudy and clear cases

- Hitrate: Frequency (value between 0 and 1) of correct cloudy and clear predictions relative to all cases
- Kuiper's skill score ($-1 \leq$ KSS $\leq 1$) where value 1 means perfect agreement, value 0 means uncorrelated (random) results and value -1 means consistently opposite results (see Karlsson and Johansson for the exact definition).

The results are computed by treating both CLARA-A2 and CALIOP cloud masks as binary values, i.e., each FOV is
considered as either fully cloudy or cloud free. The Kuiper's skill score can be used to better identify cases of misclassifications when one of the categories is dominating. The KSS is sensitive to misclassifications even if they occur in only a small minority of the studied cases. The KSS score aims to answer the question of how well the estimation separated cloudy events from cloud-free events.

A minimum requirement for describing the accuracy of a parameter is to estimate the mean error or bias (giving the systematic error) and the variance of the error (giving the random error or dispersion) (Merchant et al. 2017). However, to enable the identification of specific problems with cloud identification it is necessary to look at the additional scores mentioned above, particularly in cases when one of the two categories ("cloudy" or "clear") is dominant. This is motivated by the fact that any cloud contamination (even if it is just a few cases) can have serious implications for parameter retrievals
further downstream in the processing. Therefore multiple validation scores are needed to correctly identify all problematic and critical cases.

### 3.5 Extension of the original validation method: enhanced analysis and introduction of cloud layer detection probability

The use of the CALIOP cloud mask for validation of cloud masking methods based on passive imagery is rewarding but also
challenging. It is known from previous results which used the original CALIOP cloud mask that there is a large difference in sensitivity between CALIOP (high sensitivity) and passive sensors (moderate to low sensitivity) which leads to the question: how can this sensitivity difference be managed to ensure the generation of useful results?

There are two major risks when comparing cloud masks retrieved from passive sensors to the original CALIOP cloud mask:

1. The CALIOP dataset will include sub-visible clouds (Martins et al., 2011) which are not possible to detect in passive imagery.
2. In areas where sub-visible clouds exist in abundance, a method may have been 'overtrained' or 'overfitted' (e.g., if trained with CALIOP data by statistical regression methods) to always predict clouds since this gives the best
overall validation scores.

These two problems can be handled by focusing on what happens for clouds that have different vertically integrated optical thicknesses as provided by the CALIOP 5 km cloud product. By applying successively reduced CALIOP cloud masks in the validation exercise we may exclude the thinnest clouds from the analysis by transforming them into cloud-free FOVs. This
also means that we can isolate clouds within finite cloud optical thickness intervals (i.e., by subtracting two adjacent restricted CALIOP cloud masks with different filtered cloud optical thickness) in order to calculate validation results exclusively for this sub-set of clouds. If the cloud optical thickness interval is sufficiently small and the number of samples within each interval is sufficiently high we may then estimate the method's efficiency in detecting a cloud (i.e., the cloud layer detection probability $POD_{cloudy}(\tau)$ where $\tau$ is the mean optical thickness or depth in the given interval) with this

particular cloud optical thickness. We may then expect to see low detection scores for small optical thicknesses with scores improving as cloud optical thickness values increase. We argue that a special situation occurs when this cloud layer detection probability for the first time exceeds 50 % for increasing cloud optical thicknesses. This marks an important performance point which could be seen as a minimum performance requirement: at this cloud optical thickness we detect at least 50 % of all clouds. In the following we will denote this value of the filtered cloud optical thickness as the method's ***cloud detection sensitivity***. There should also be a peak in the Hitrate parameter at exactly this point. For small optical thicknesses, scores would improve if we filter out thin clouds, while for larger optical thicknesses scores start to decrease as too many correctly detected clouds are transformed to the cloud-free case. We maintain that the best way to evaluate a cloud masking method is to estimate this cloud sensitivity parameter and to re-compute all validation scores after applying optical thickness filtering using exactly this value. This describes a method's optimal performance when using CALIOP cloud masks as the reference. The cloud detection sensitivity parameter defines the method's cloud detection capability in terms of the thinnest cloud that can confidently be detected. Furthermore, the validation scores computed at this value of the filtered optical thickness then define the method's optimal performance (in terms of the Hitrate) taking into account also false classifications. An important complementary parameter in this context is the false alarm rate in the unfiltered case ($FAR_{cloudy}(\tau=0)$) since this parameter does not depend on any filtering of optically thin clouds. $FAR_{cloudy}(\tau=0)$ can be used to investigate the degree of overtraining of a method (according to second bullet above). In the following Section 4, we present results of the cloud detection sensitivity and a range of validation scores computed at the point of the cloud detection sensitivity (i.e., using a CALIOP cloud mask filtered for thin clouds using the cloud detection sensitivity parameter as the optical thickness threshold) Most of these results are presented as global maps.

### 3.6 The final compiled validation dataset

We have matched a total number of 5747 global afternoon orbits of the NOAA-18 and NOAA-19 satellites with corresponding CALIPSO-CALIOP data in the time period October 2006 to December 2015. Due to increasing orbital drift of the NOAA-18 satellite after 2010 (with resulting deviation from the A-Train orbit and increasing off-nadir viewing angles for matchups), the matchup dataset contains a small fraction of observations with higher satellite zenith angles. The observation time difference is limited to 3 minutes and the spatial matchup error was maximised to 2.5 km (as a consequence of using the nearest neighbouring technique and after assuming negligible geolocation errors). This resulted in more than 23 million global matchups. The distribution of the matchups is shown in Fig. 2 using a Fibonacci grid resolution of 75 km.

Figure 2 shows a large variation in coverage as a function of latitude with a minimum number of matchups occurring at low latitudes and a maximum of matchups for the highest latitudes. Although the likelihood for a valid matchup to occur is the same everywhere on a particular matched orbit, the pattern of the matchup numbers is explained by the converging orbital tracks towards the poles. Furthermore, the large variation with some distinct features (e.g., over the Pacific Ocean) shows that it was not possible to extract all theoretically available matching cases (some periods with loss of data exist for both CALIOP and AVHRR). Although there is not fully homogeneous global coverage the dataset represents the best possible effort in that direction that we can make at present. Even at low latitudes the number of matches generally exceeds 300 for a grid resolution of 75 km, with only a few exceptions mainly located over the Pacific Ocean. In these locations the uncertainty in the results might be expected to be larger than for the rest of the globe.

## 4 Results

### 4.1 Results from inter-comparisons of validation results based on CALIPSO-CALIOP version 3 vs version 4

Results from the modified validation method were compared against results from the old method for a test dataset of 80 NOAA-18 CALIPSO-matched orbits between October and December 2006. These results are presented in Figs. 3 and 4. Figure 3 shows validation results for the two different approaches based exclusively on CALIOP CLAY version 4.10 products. The visualisation used here, showing the results for two validation scores (Hitrate and Kuipers score, see also discussion and definition in section 3.4) is identical to the approach seen in Karlsson and Johansson (2013). Results using the original CALIOP cloud mask are given by the leftmost value with a filtered cloud optical thickness of 0.0. The curves represent validations which use a successively reduced CALIOP cloud mask where clouds optically thinner than the values on the x-axis have been transformed from cloudy to clear cases. In this way we can calculate for which CALIOP cloud mask (i.e., for which filtered cloud optical thickness) we get the highest scores. Fig. 3 shows slightly improved results for the method using the single shot information, although they are practically identical. The slight improvement may be attributed to the improved cloud-aerosol labelling of removed single shots. Figure 4 shows the overall effect of introducing the new matching method and the new version 4 dataset compared to the results achieved using the former version 3 dataset and the previous matching method. There is a small increase in the overall results (maximum scores) and a progression of the maximum values towards larger optical depths. The improvement in results indicates an improved CALIOP product and the shifting of peak score values towards larger filtered cloud optical depths is indicative of more realistic and larger optical depths in CALIOP version 4.10 data (as confirmed by David Winker, CALIPSO Science Team, 2017, pers. comm.). These results are in line with expectations and demonstrate that the modified method is an appropriate basis for further validation studies based on the updated CALIOP CLAY dataset.

### 4.2 Results based on original CALIOP cloud masks compared to results excluding contributions from very thin clouds

Figure 5 shows the global distribution of the Hitrate parameter when comparing to the original CALIOP cloud mask. Results indicate a fairly good cloud screening capability over mid- to high latitudes (especially over oceans) but degraded results at most low latitudes and over the polar regions. The poorest results occur over Greenland and Antarctica.

Further analysis of results is complicated by the fact that the original CALIOP cloud mask includes all CALIOP-detected clouds as explained in Section 3.5. In particular, we suspect that the rather poor results in Fig. 5 in the tropical region may be significantly influenced by the presence of sub-visible clouds.

By using all available matchups, we can calculate $POD_{cloudy}(\tau)$ for all values of $\tau$ (Fig. 6) using the method outlined in Section 3.5. Calculations have been based on optical thickness intervals of 0.05 in the range $0.0 < \tau < 0.5$, intervals of 0.1 in the range $0.5 < \tau < 1.0$ and intervals of 1.0 in the range $1.0 < \tau < 5.0$ (results from the latter interval are not shown in Fig. 6.). Figure 6 shows that the cloud detection sensitivity (i.e., where a probability of 50 % is reached) is 0.225 for the investigated AVHRR-based results. Consequently, we will use this value to indicate the optimal Hitrate results, with the global distribution of these results presented in Fig. 7. As expected, the results improve considerably for most locations compared to Fig. 5, especially over low latitudes. Hitrates above 80 % are now achieved over most regions. The polar regions (at least the snow- and ice-covered parts) stand out as regions of poor quality with the worst results seen over central Greenland and Antarctica. There is also some degradation in the results over some regions at low-to-middle latitudes.

The results in Fig. 7 give a much clearer measure of the cloud detection capability of the CLARA-A2 cloud screening
method than those shown in Fig. 5, because they are now linked to a well-defined description of the involved clouds. We
will apply the same filtering approach to obtain the results shown in the next sub-section.

### 4.3 Additional validation scores

Figure 8 presents results for the systematic (bias) and random errors (bias-corrected RMS) of the CLARA-A2 cloud
amounts. It is clear that the cloud detection problems over the polar regions, as indicated by the Hitrate parameter in Fig. 7,
lead to a significant underestimation of cloud amounts, especially over those areas normally covered with snow or ice.
However, this is an overall mean (close to an annual mean) and the underlying results may be seasonally varying. For
example, cloud detection in the polar summer season is considerably better than during the polar winter (as shown by Fig. 6
in Karlsson et al., 2017). The results with least bias are found over mid-to-high latitudes while some overestimation is seen
over lower latitudes, particularly over oceanic surfaces. RMS values are high in the polar regions and over what can be
described as oceanic sub-tropical high regions. This agrees well with the corresponding Hitrate results seen in Fig. 6. RMS
values are low over dry desert regions but mostly as a consequence of the general lack of cloudy situations here.

To further investigate areas where there is significant misclassification of cloudy and clear conditions we can study results of
probability of detection of the cloudy and clear categories in Figure 9. For the cloudy category results are consistent with
those deduced from previous figures with the exception of the low probabilities of cloud detection over northern Africa and
the Arabian Peninsula. For the clear category we note high values over predominantly dry land portions of the world while
low values are seen over the tropical region and over oceanic storm track regions at high latitudes.

Results for the Kuipers score are shown in Fig. 10. This score does not show as much regional variability as the Hitrate
score. Again, we note low score values over the snow-covered polar regions and over some desert regions. The largest
difference to the Hitrate is seen over high-latitude oceanic regions where the Kuipers score show rather modest values while
Hitrate showed relatively high score values.

Figure 11 show the corresponding false alarm rates for cloudy and clear conditions. We note high false alarm rates for
cloudy conditions over tropical and sub-tropical regions (with some dominance for oceanic regions) while for clear
conditions the largest false alarm rates are found in the polar regions.

### 4.4 Estimating the global variability of cloud detection limitations

We have here presented validation results after having 'removed' (in the sense of interpreting them as cloud-free cases) all
clouds with smaller optical depths than the cloud detection sensitivity parameter. This leads to a clear improvement in the
results when compared to the original CALIOP cloud mask (i.e., comparing Figs. 5 and 7). However, the cloud detection
sensitivity value currently applied is a global average which could contribute to the large geographical variations in the
results. To investigate how serious this simplification is, we can plot the results of $\tau_{min}$(POD>50) calculated exclusively for
every Fibonacci grid point (Fig. 12). To reduce the uncertainty in this calculation due to low number of samples per grid
point as indicated in Fig. 2 for low latitudes, we have increased the radius of the Fibonacci grid from 75 km to 300 km.
Figure 12 shows a considerable variation in cloud detection sensitivity over the globe. It is clear that the cloud detection
sensitivity is considerably lower than the global average value of 0.225 over most oceanic areas as well as over tropical land
areas. On the other hand, values are generally larger than 0.225 over dry and desert-like regions and over high-latitude and

polar land areas. For the polar land areas the cloud detection sensitivity frequently exceeds 1 and for some grid points even reaches values close to 5. These values contrast with the global average value of 0.225, indicating that more representative (and most likely higher) validation scores could have been achieved if globally resolved cloud detection sensitivity values were used to re-calculate each of the validation scores. However, we have not taken this step here because of the relatively low number of samples in some grid points (even at the 300 km scale).


We can also visualise the variable cloud detection sensitivity by plotting the same kind of cloud layer probability curves as in Fig. 6 for a selection of individual grid points.. Figure 13 shows these curves for the three locations marked in Fig. 12. The blue curve in Fig. 13 shows cloud layer detection probabilities for a distant (from land) point in the North Atlantic Ocean. It marks a position where cloud detection is clearly most effective compared to the global average. The cloud detection

sensitivity value is 0.075 at this location demonstrating that even very thin clouds are well detected there. The cloud detection capability also reaches a maximum value of approximately 95 % by $\tau = 0.5$. This is considered to be as high as can be reached because of the limitations of the datasets, for instance the remaining and unavoidable AVHRR-CALIOP mis-location and matching problems (both in time and space). As a contrast, a grid point located in the Sahel region (green curve in Fig. 13) shows worse results with a cloud detection sensitivity of 0.375 and maximum cloud detection capability only

observed at $\tau = 3.5$ and higher. However, a more extreme case is the location over central Greenland (red curve in Fig. 13). The cloud detection sensitivity here is as large as 1.5 and even at a maximum $\tau$ value of 4.5 we can not come close to achieving an optimal cloud detection capability. Thus, over a snow-covered and often extremely cold location we cannot even detect all optically thick clouds which is consistent with the low $POD_{cloudy}$ results seen over Greenland and Antarctica in Fig. 9, upper panel).


The results in Fig. 13 again indicate that the validation matchup dataset slightly undersamples the true conditions for a limited number of grid points. This is indicated by the unexpected decrease in POD at some points for increasing $\tau$ values. Theoretically, one would expect a steady increase in POD as a function of $\tau$.

**5 Discussion**

There are several features of the results depicted in Figs 7-11 which warrant further attention and discussion. One of these is the reduction in performance observed over areas which are known to be dry and mostly cloud-free. The $POD_{cloudy}$ results in Fig. 9 show particularly low values over the Sahara Desert and the Arabian Peninsula. This indicates that in these particular areas, where cloudiness is generally low, CLARA-A2 still has difficulty detecting the few cloudy cases which occur. The exact reasons for this have to be investigated further but are likely linked to remaining uncertainties in the surface

emissivities used over these semi-arid regions and deserts.

Another feature to discuss is the overestimation of cloudiness over low and medium latitudes (especially over oceans) seen in the Bias plot in Fig. 8. This feature illustrates how it is difficult to find a simple representative way of evaluating results while also taking into account the existence of sub-visible clouds. The method applied in Fig. 8 (and in all Figs 7-11) is to

ignore cloud contributions in the CALIOP dataset for clouds having an optical thickness less than 0.225. But, as already mentioned in Section 4.4, the latter value is a global mean value and in many places on Earth clouds with smaller optical thicknesses are actually detected confidently. This is clearly demonstrated in Fig. 12 where the cloud detection sensitivity over oceanic surfaces is noticeably better (smaller) than the global mean of 0.225. This means that by applying the global value 0.225 as the filtering threshold of CALIOP-detected clouds, many clouds which were originally correctly detected in

CLARA-A2 will now be treated as being falsely detected. If a locally representative value of the cloud detection sensitivity

(as shown in Fig. 12) is used for the CALIOP filtering procedure, this apparent overestimation of clouds would largely disappear. However, to confidently apply such localised filtering a larger set of collocated observations is required to remove the sensitivity to low numbers of samples in individual grid points. Such a study will be possible in a few more years once an even larger matchup dataset has been collected. An extended dataset could also allow a further sub-division of the dataset to study the diurnal and seasonal variation of the validation results.

A more interesting and general feature is shown in Fig. 9: In areas where cloudiness is low (e.g., over sub-tropical ocean and land regions) $POD_{cloudy}$ is low and where cloudiness is high (e.g., over mid-latitude storm tracks and near the equator) $POD_{clear}$ is low. This explains to a large extent the fairly low values of the Kuipers' score over these regions (Fig. 10) leading to a slightly different distribution of results in comparison to the Hitrate (Fig. 7). However, we must remember that Hitrate is dominated by results for the dominating mode (cloudy or clear) while the Kuipers score highlights more clearly the existence of misclassifications of the minority mode. Figs 9 and 10 reveal that even if the dominantly cloudy and clear regions are generally captured very well the few cases of the opposing mode have a high frequency of misclassifications. This result is difficult to understand from the perspective of long-term experience of AVHRR cloud screening, as cloud screening works best over dark and warm ocean surfaces in good illumination. So, why are results not better here (e.g., over oceanic sub-tropical high regions)? We believe that this unexpected behaviour is a consequence of the limitations of both AVHRR GAC data and CALIPSO-CALIOP data when it comes to the sampling of the true conditions within the nominal 5 km FOV.

To understand this we have to go back to Fig. 1 displaying the conditions for the matching of AVHRR GAC and CALIOP observations and the overall collocation geometry. Sections 2.1 and 3.2, together with Fig. 1, clearly describes how only about 25 % of the nominal 5 km AVHRR GAC FOV is actually observed by AVHRR and that the corresponding figure for CALIOP single shot nominal FOV of size 330 meters is as low as 5 %. Notice that the latter means that CALIOP is only able to cover about 0.3 % of the nominal 5 km FOV. This has important consequences for all cases where we have cloud elements present which are smaller in size than the nominal 5 km FOV. We can first conclude that only in those cases containing cloud elements larger than the nominal 5 km FOV can we be confident that AVHRR and CALIOP observations will be comparable. For all other cloud situations involving clouds smaller than 5 km or when a cloud edge occurs within the GAC FOV, the two data sources will give different results since the sensors will observe different parts of the 5 km FOV. The situation is compounded by the fact that the AVHRR scan lines are perpendicular to the CALIPSO track when matching the two datasets in the near-nadir mode (Fig. 1, upper panel). This means that the CALIOP sensor consistently probes a different part of the nominal 5 km FOV to AVHRR. Theoretically, a maximum of 3 CALIOP single shot measurements (out of a total of 15) would be able to measure the same spot on Earth as the AVHRR GAC measurement within the FOV size of 5 km. However, it is clear from Fig. 1 that in a non-negligible fraction of cases, the two sensors will not even observe any common part of the nominal GAC FOV. This occurs when the nearest-neighbour matching of GAC and CALIOP FOVs places the CALIOP FOV in the rightmost part of the GAC FOV (see Fig. 1, upper panel). A direct consequence of these differences between the actual AVHRR and CALIOP measurements is that, in the case of dominating fractional cloudiness with cloud size modes below the 5 km scale, the random errors and the false-alarm rates will increase even if the overall bias remains small (assuming that the cloud element distribution within the GAC FOV is random over a long time period, i.e., as expected for climate data records). This behaviour is exactly what is observed over the oceanic sub-tropical high regions (Fig. 8 and Fig. 11, upper panel) and also explains the degraded overall scores in this region (in particular the $POD_{cloudy}$ score in Fig. 9) relative to other surrounding regions.

These regions of interest also have a reduced total cloud amount in the annual mean (e.g., see Fig. 6 in Karlsson et. al., 2017), mainly because of the more stable atmospheric conditions here. The prevailing large-scale subsidence (poleward parts

of the Hadley cell) in these locations suppresses cloudiness in mid- to high layers and is conducive only to the formation of convective and stratiform boundary layer clouds. This boundary layer cloudiness consists mainly of scattered small-scale cumulus and stratocumulus clouds, i.e., typically the kind of clouds for which we would expect enhanced disagreement between the AVHRR and CALIOP datasets as a result of variability wihin the 5 km FOV. It is interesting to note that this feature is not exclusive to oceanic areas. In addition some eastern parts of continents show similar results, e.g. easternmost part of South America and Africa. This could indicate that scattered cumulus cloudiness is also the dominant mode of cloudiness in these locations. Finally, notice also that we can see exactly the same effect for fractional clear areas, e.g. over northern and southern hemisphere stormtracks at mid- to high latitudes as shown by the large $FAR_{clear}$ values in Fig. 11. We conclude that, because of the problems with correctly representing cases of both small-scale cloudiness and small-scale holes in cloud decks in the two datasets, the validation results could be underestimated (i.e., giving too low scores) over these dominantly cloudy or dominantly clear regions of the globe. This reduction of scores would then be largely attributed to mis-matches due to GAC FOV geolocation errors (which are not zero), matchup errors (explained by the nearest-neighbour matching of GAC and CALIOP FOVs) and to the different cloud representation in each dataset rather than to real cloud detection problems. Thus, examination of the cloud detection capability of a method should also take into account the scales of clouds being investigated. A consequence of this is that detailed studies of small-scale convective cloudiness should rather be based on original resolution AVHRR and CALIOP observations than on datasets with a coarse resolution data representation.

Finally, a specific problem with the applicability of the current method is the inability to assess the global quality of products from polar satellites in morning orbits (e.g., from the NOAA-17 and Metop satellites) as a consequence of CALIPSO following an afternoon orbit. Matchups with CALIPSO-CALIOP are consequently only possible at high latitudes leaving low-to-middle latitudes without reference observations for AVHRR products. Previous cloudiness comparisons for morning satellites at high latitudes (CM SAF 1, 2017) show good agreement with corresponding results from afternoon satellites (assuming that diurnal cycle cloud effects are small at high latitudes). Thus, for cloud amount information (in contrast to some other cloud parameters, like cloud effective radius) there is no reason to suspect large differences between morning and afternoon results even if morning orbit data is partly using measurements in another spectral band (at 1.6 μm) in the short-wave infrared spectral region. However, this needs to be confirmed in the future through the use of reference data from the Cloud-Aerosol Transport System lidar (CATS, https://cats.gsfc.nasa.gov/) on the International Space Station or by use of data from the Earth Cloud Aerosol and Radiation Explorer (EarthCARE) mission (http://m.esa.int/Our_Activities/Observing_the_Earth/The_Living_Planet_Programme/Earth_Explorers/EarthCARE/ESA_s_cloud_aerosol_and_radiation_mission) for new afternoon satellites with two coexisting short-wave infrared channels onboard (e.g. NOAA-20).

## 6. Conclusions

We have shown that with access to the latest cloud information provided by the high-sensitivity CALIPSO-CALIOP lidar (CALIOP Version 4.10 dataset, covering almost a full decade (2006-2015) it is possible to construct a detailed global analysis of the cloud detection sensitivity and other skill scores of the cloud screening method used in the AVHRR-based CLARA-A2 cloud climate data record. A wide range of validation scores, including those complementary to the essential scores describing systematic and random errors, have been used to get a very detailed picture of the cloud screening efficiency of CLARA-A2. Furthermore, by use of the CALIOP-derived information on cloud optical thickness, it has been possible to make a clear definition of which clouds have been observed and thus for which clouds the validation scores are valid. We believe this to be crucial to the further quantitative use of the results. The method is not specifically developed or

valid exclusively for the CLARA-A2 cloud masking method but is also applicable to any method utilizing CALIOP data as a reference. Consequently, we propose that this method be used in future inter-comparisons of results from different cloud masking methods and cloud CDRs (following the example by Stubenrauch et al., 2013).

It is necessary to specify the clouds being investigated because the CALIOP sensor is capable of detecting clouds which are fundamentally "sub-visible" for passive imaging sensors. Therefore, a globally estimated minimum cloud optical thickness value (denoted "Cloud detection sensitivity"), for which the majority of clouds would be detected, was estimated to be 0.225 for the CLARA-A2 cloud masking method. This value was used to remove contributions to validation scores from thinner clouds than this minimum optical thickness, thus maximising the validation scores. For example, by utilising this definition of detectable clouds, resulting cloud amounts were found to be unbiased over most locations of the world except for a major underestimation over the polar regions. For the latter, a large part of all clouds still remain undetected during the polar night and this fraction can be as high as 50 % over the coldest and highest portions of Greenland and Antarctica. Under these conditions not even optically thick clouds may be detected due to the very similar thermal characteristics of clouds and Earth surfaces. Land-ocean differences were generally small with only results over Greenland and Antarctica standing out as clear exceptions.

The study revealed some interesting reductions in performance over mainly sub-tropical ocean areas. In these locations random errors were elevated indicating a decrease in agreement between AVHRR and CALIOP observations despite otherwise very favourable cloud detection conditions (e.g., warm ocean temperatures and good illumination conditions).We argue that this is caused by the different sampling conditions within the studied 5 km FOV of the AVHRR and the CALIOP sensors, which is particularly evident in cases where small-scale boundary layer cloudiness dominates the cloud situation. Because of this we suspect that the cloud detection capability over these areas could actually be better than that shown by these results.

An important novel feature of this study compared to many previous validation efforts based on CALIPSO-CALIOP data is the estimation of the probability of detecting an individual cloud as a function of its vertically integrated optical thickness and its geographical position on Earth. This was accomplished by isolating finite optical thickness intervals in the CALIOP cloud information and calculating validation scores for this subset of data in a coarse global grid. Results show a substantial variation compared to the global mean optical thickness value of 0.225 for the thinnest retained cloud in the CALIOP cloud mask to give optimal global validation scores. The highest sensitivity to clouds in AVHRR data is generally found over mid-to-high latitude ocean surfaces. Here, clouds with cloud optical thicknesses as low as 0.075 can be detected efficiently. This is in comparison to a value of approximately 0.2 over tropical oceans and typically greater than 0.2 over most land surfaces. The latter value reaches 0.5 over some dry and desert-like regions (e.g., the Sahara Desert and the Arabian Peninsula) and increases towards or beyond 1 over polar regions with a highest value of 4.5 found over Greenland and Antarctica. These results indicate that not even optically thick clouds can be confidently identified over Greenland and Antarctica during the polar winter. While these are not entirely new findings (e.g., see Karlsson and Dybbroe, 2010), this study has increased the confidence in the validation results over the polar regions. Consequently, these results could help in optimizing the combined use of passive and active cloud observations over the polar areas in specific process and radiation studies (similar to earlier work by Kay and Gettelman, 2009, and Kay and L'Ecuyer, 2013).

The presented validation method can be viewed upon as a step towards a more stringent and universal validation method to be used consistently for cloud climate data records generated from passive imagery (as discussed in Wu et al., 2017). The more than decadal long CALIPSO-CALIOP cloud dataset should be used for benchmarking and for evaluation of current

CDRs and future revisions of them. The method presented here could be seen as one candidate method. The ability to derive globally distributed results makes it easier to define and test global quality requirements for the CDRs. For example, requirements could be formulated in terms of minimum global coverage within a certain quality threshold instead of today's often overly generalised global requirement which use one finite value or a value range (WMO 2, 2011).

One particular aim of this study was to provide a strict definition of the clouds being validated alongside the main validation results. This has been accomplished through the use of the CALIOP-derived cloud mask and the CALIOP-estimated optical thickness of clouds. As a result these validation results are more quantitatively useful. One obvious application would be to incorporate this information about strengths and limitations of cloud detection capabilities into the cloud dataset simulators of the Cloud Feedback Model Intercomparison Project (CFMIP) Observation Simulation Package (COSP, Bodas-Salcedo et al., 2011). Existing COSP simulators for cloud datasets generated from passive satellite imagery (e.g., ISCCP and MODIS) do not explicitly take into account these potential inherent cloud detection problems and instead, they concentrate on simulating some satellite-specific or retrieval-specific features (e.g., systematic underestimation of cloud top height of thin high clouds) leaving it to the user of the simulator to add existing knowledge on cloud detection efficiency in the final evaluation process. It would clearly be beneficial if aspects of cloud detection capabilities were to be explicitly accounted for in these simulators. A specific CLARA-A2 COSP simulator is therefore under development where the description of such quality aspects will be included based on the findings of this validation study.

Finally, we repeat our opinion that CALIPSO-CALIOP data is an invaluable asset for the current and future evaluation of cloud CDRs based on passive satellite imagery. At the same time, we must express our concern about the current uncertainty regarding the long-term planning of possible replacements of both the A-Train satellites and the upcoming EarthCARE mission. Without follow-on missions it will be very difficult to assess the critical long-term stability of these CDRs, which in turn increases the difficulty in assessing the reliability of any climate trends deduced from these CDRs. There is also a need to slowly transform CLARA-type data records to AVHRR-heritage data records, i.e., extend the AVHRR results into the future using results from similar spectral channels existing on other sensors (e.g., the VIIRS sensor on recently launched and future polar NOAA satellites). A continued access to observations from active space-born lidar systems is essential for the development of such AVHRR-heritage data records.

**Author contributions**

Karl-Göran Karlsson conducted the validation study and wrote the manuscript. Nina Håkansson prepared the calculation and visualisation of the global results.

**Competing interest**

The authors declare that they have no conflict of interests.

**Acknowledgements**

CALIPSO-CALIOP datasets were obtained from the NASA Langley Research Center Atmospheric Science Data Center. The authors want to thank Dr. David Winker in the CALIPSO Science Team for valuable advice regarding the use of CALIOP cloud information. The authors are also grateful to David Dufton for constructive comments on the manuscript.

This work was funded by EUMETSAT in cooperation with the national meteorological institutes of Germany, Sweden, Finland, the Netherlands, Belgium, Switzerland and United Kingdom.

The CLARA-A2 data record is (as all CM SAF CDRs) freely available via the website https://www.cmsaf.eu.

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

**AVHRR-CALIOP matching for afternoon orbits**

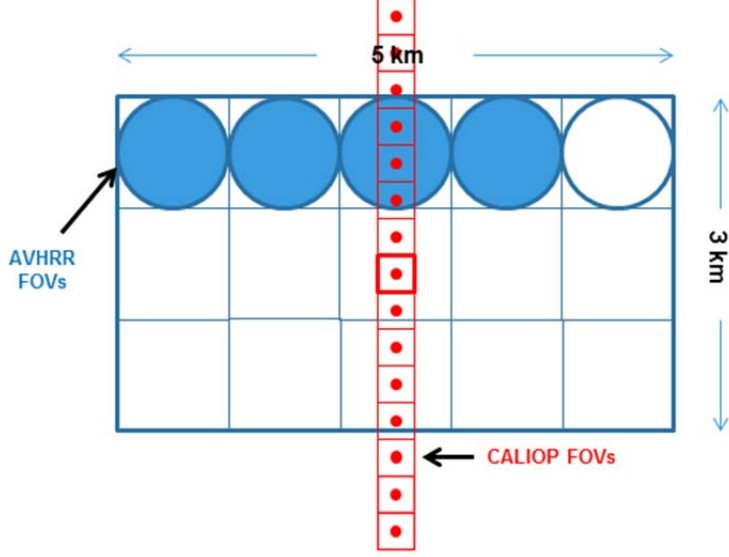

**AVHRR-CALIOP matching for morning orbits**

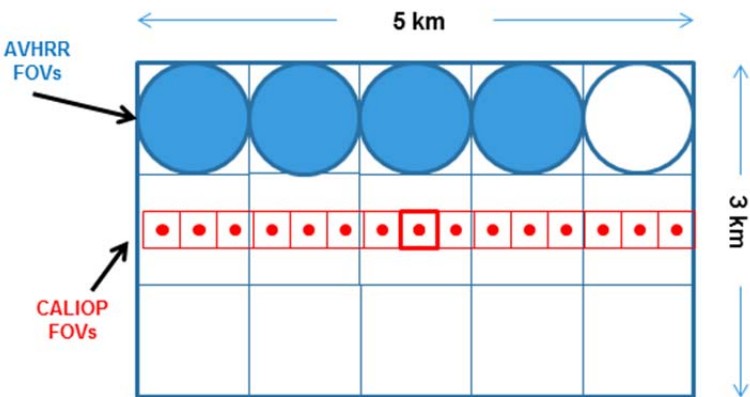

**Figure 1: Matchup geometry for perfectly collocated AVHRR GAC and CALIOP FOVs for afternoon satellites (top) and morning satellites (bottom). The GAC FOV is visualized as a rectangle with sides 3 km and 5 km and with individual full resolution AVHRR FOVs represented as 1 km squares. Blue circles indicate actual (more realistic) AVHRR measurements being used. Note that only the blue filled AVHRR FOVs are averaged to represent the full GAC FOV. Red squares denote 15 original nominal 333 m CALIOP FOVs which represent the CALIOP 5 km FOV coverage. The highlighted centre FOV marks the position of the perfect match (i.e., at the center of the GAC FOV). Note that the red filled circles describe actual CALIOP measurements. See text for a more detailed explanation.**


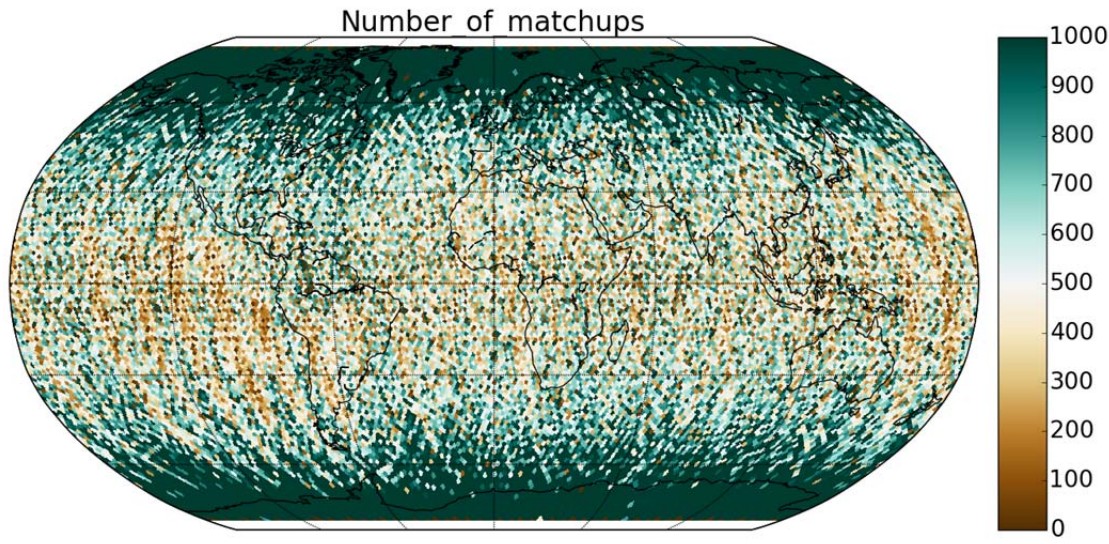

**Figure 2: Total number of CALIPSO-CALIOP matchups with NOAA-18 and NOAA-19 AVHRR observations in the time period**
**October 2006 to December 2015. Results are presented in a Fibonacci grid with 75 km resolution.**


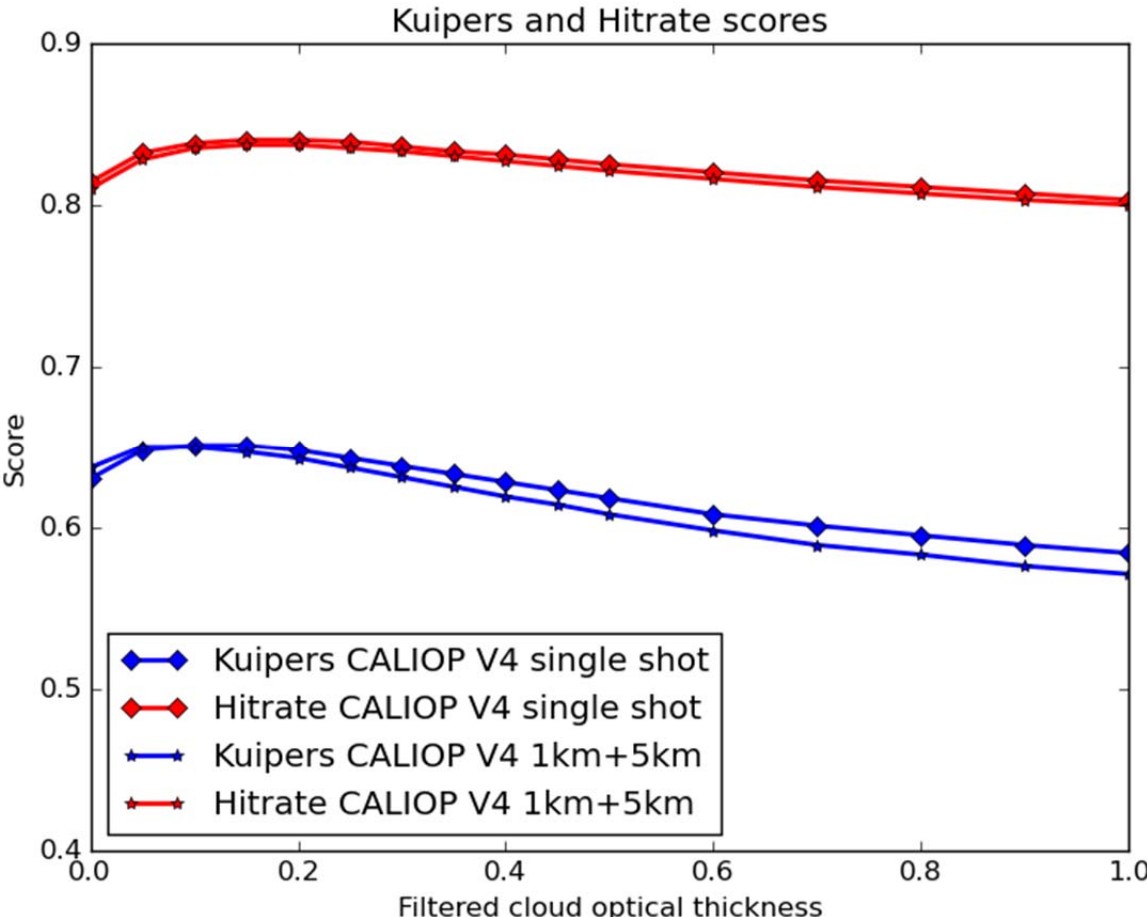

**Figure 3: CALIOP-based validation scores (Hitrate and Kuipers) as a function of filtered cloud optical thickness (see text for**
**explanation) for 80 matched NOAA-18 orbits between October and December 2006. Validation is based on CALIOP version 4.10**
**CLAY products and show results from two alternative validation methods (single shot or combined 1 km + 5 km, see text for**
**explanation).**


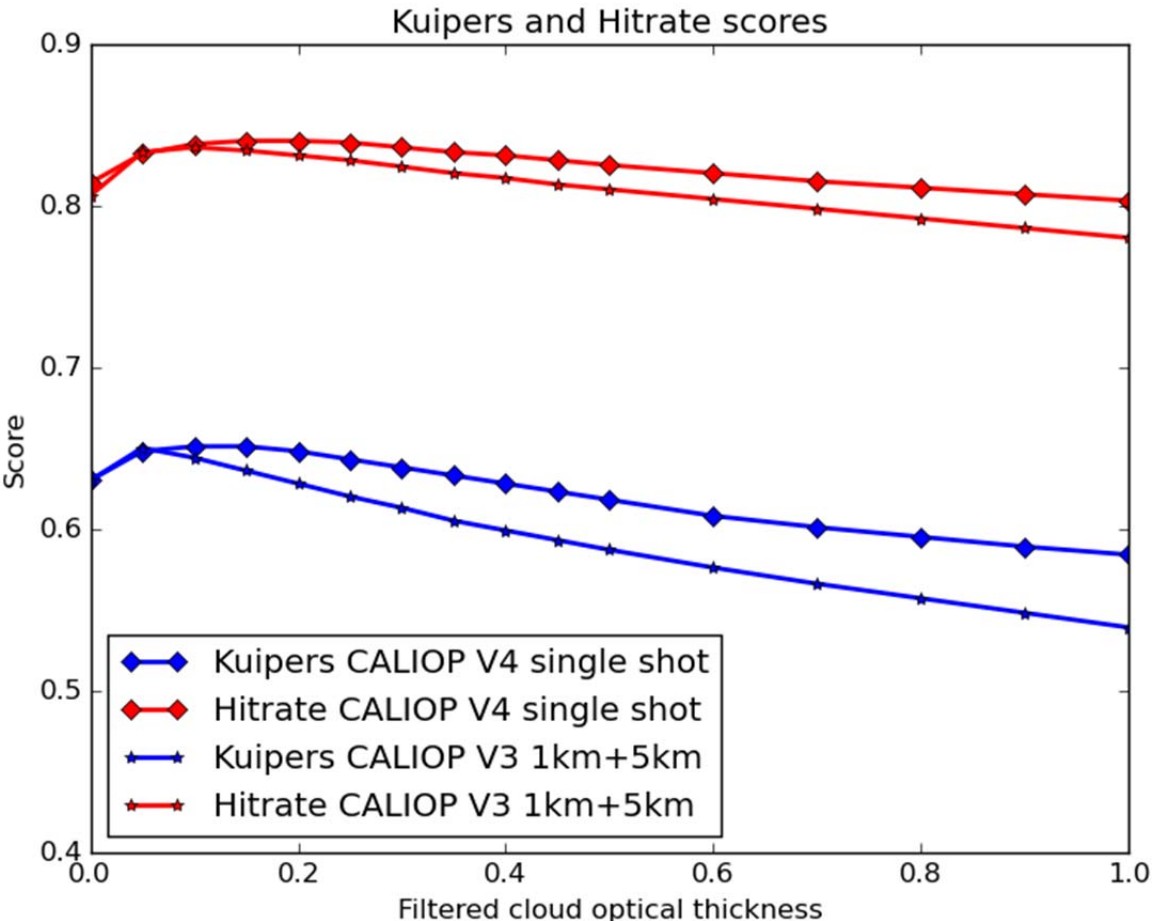

**Figure 4: CALIOP-based validation scores (Hitrate and Kuipers) as a function of filtered cloud optical thickness (see text for explanation) for 80 matched NOAA-18 orbits between October and December 2006. The curves compare results based on CALIOP version 4.10 CLAY products computed with the new method based on single shot information (denoted "CALIOP V4 single shot") with results based on CALIOP version 3.01 CLAY products computed with the old method based on combined 1 km + 5 km data (denoted "CALIOP V3 1km+5km").**



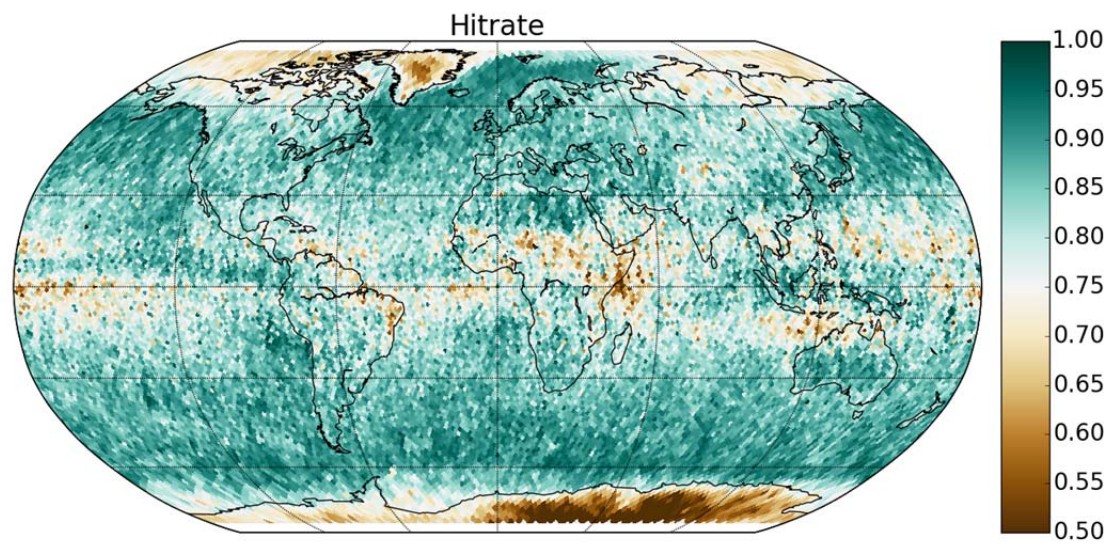

**Figure 5: Global presentation of the CLARA-A2 cloud mask Hitrate parameter with a horizontal Fibonacci grid resolution of 75 km. Validation results are based on comparisons with the original CALIPSO-CALIOP cloud mask. Same underlying matchup dataset as in Fig. 2.**




**Figure 6: Global estimation of the probability of detecting a cloud with a certain cloud optical thickness. Calculations are based on all available AVHRR-CALIOP matchups over the time period October 2006 to December 2015.**



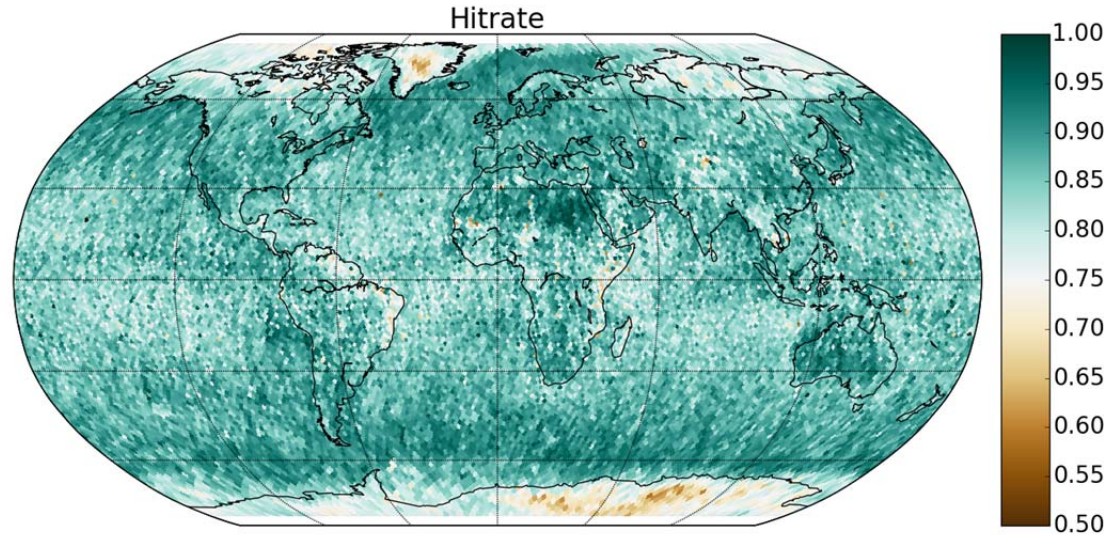

**Figure 7: Peak Hitrate results for the CLARA-A2 cloud mask achieved after filtering the CALIOP cloud mask with the cloud optical thickness value of 0.225. Same underlying matchup dataset as in Fig. 2. Results are presented in a Fibonacci grid with 75 km resolution.**



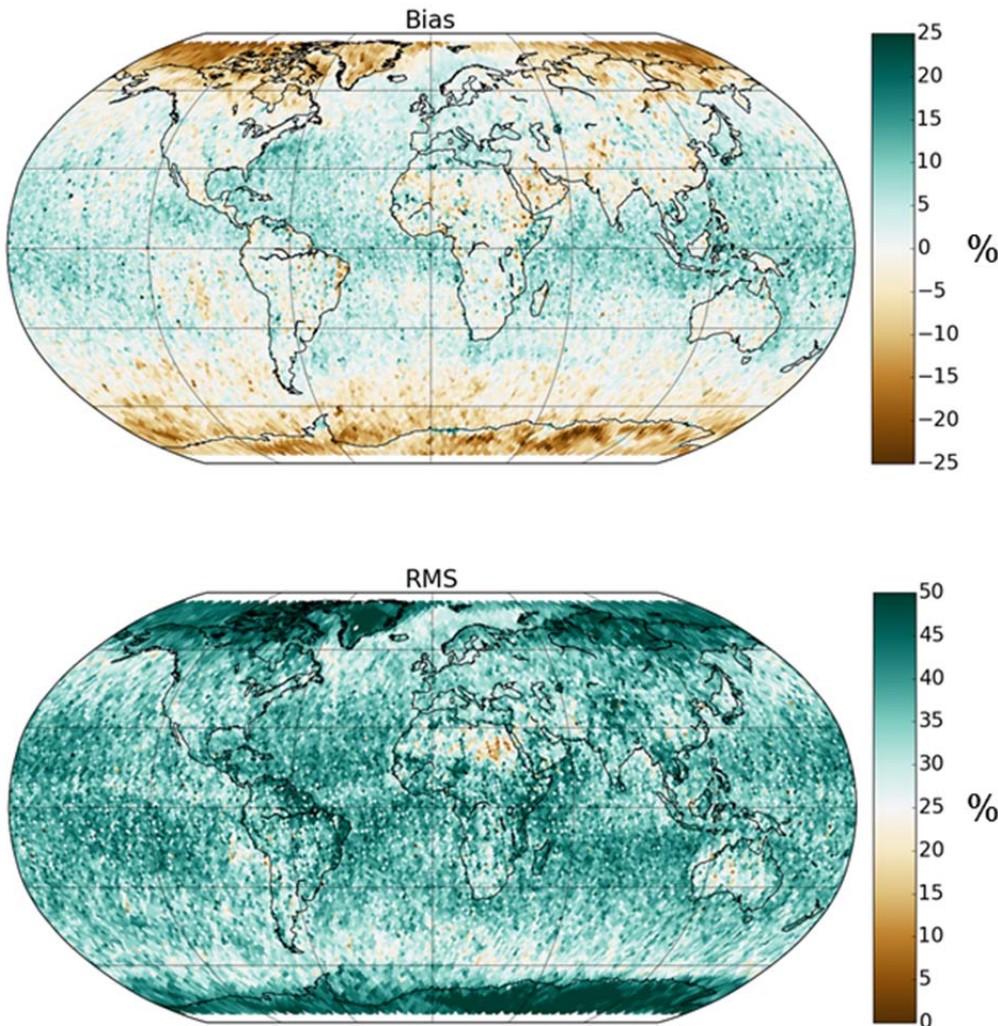

**Figure 8: Mean Error (Bias) and bias-corrected Root Mean Squared Error (RMS) for the CLARA-A2 cloud amount achieved
after filtering the CALIOP cloud mask with the cloud optical thickness value of 0.225. Same underlying matchup dataset as in Fig.
2. Results are presented in a Fibonacci grid with 75 km resolution.**


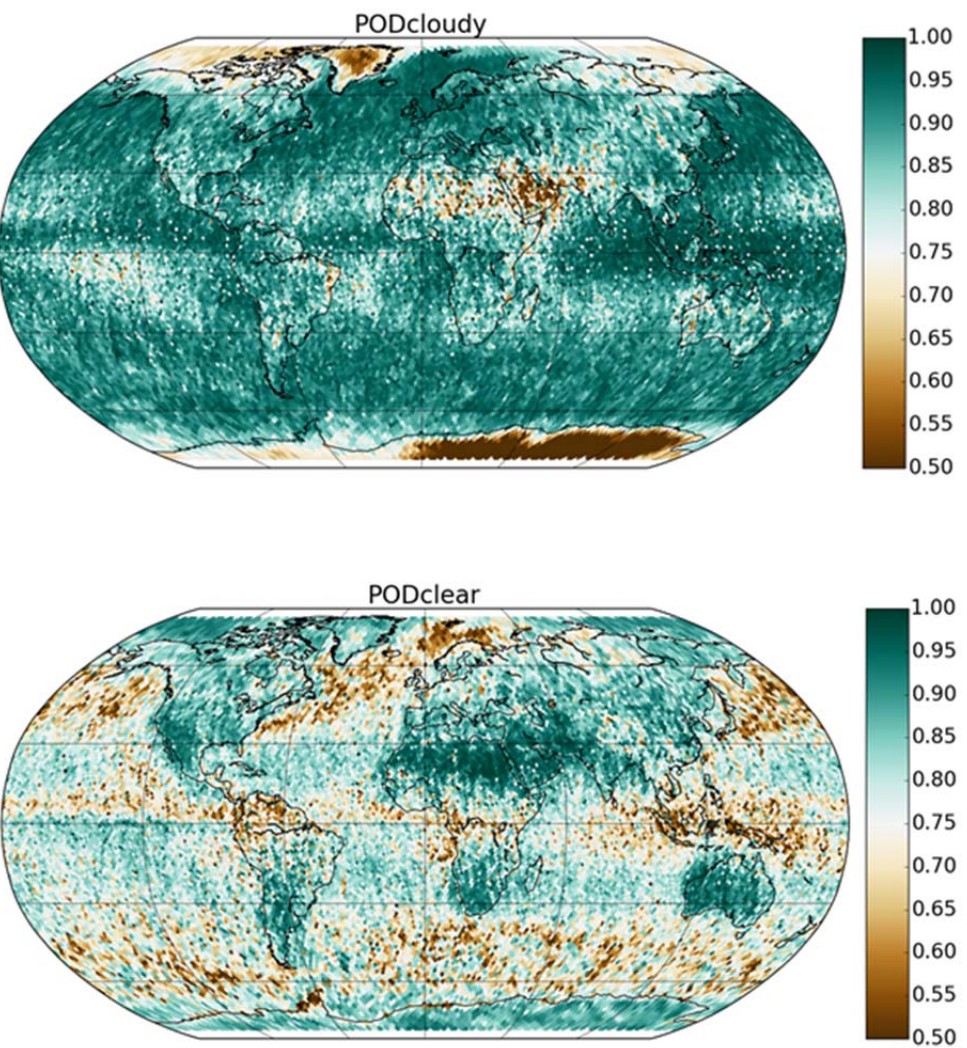

**Figure 9: Probability of detection of cloudy (top) and clear (bottom) conditions for the CLARA-A2 cloud mask achieved after filtering the CALIOP cloud mask with the cloud optical thickness value of 0.225. Same underlying matchup dataset as in Fig. 2. Results are presented in a Fibonacci grid with 75 km resolution.**


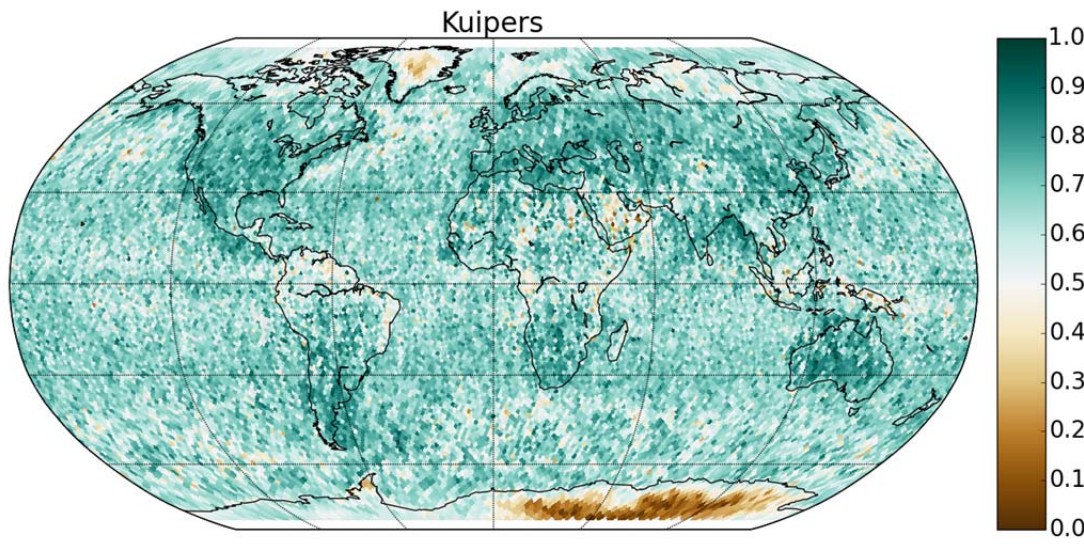

**Figure 10: Kuipers score for the CLARA-A2 cloud mask achieved after filtering the CALIOP cloud mask with the cloud optical thickness value of 0.225.  Same underlying matchup dataset as in Fig. 2. Results are presented in a Fibonacci grid with 75 km resolution.**



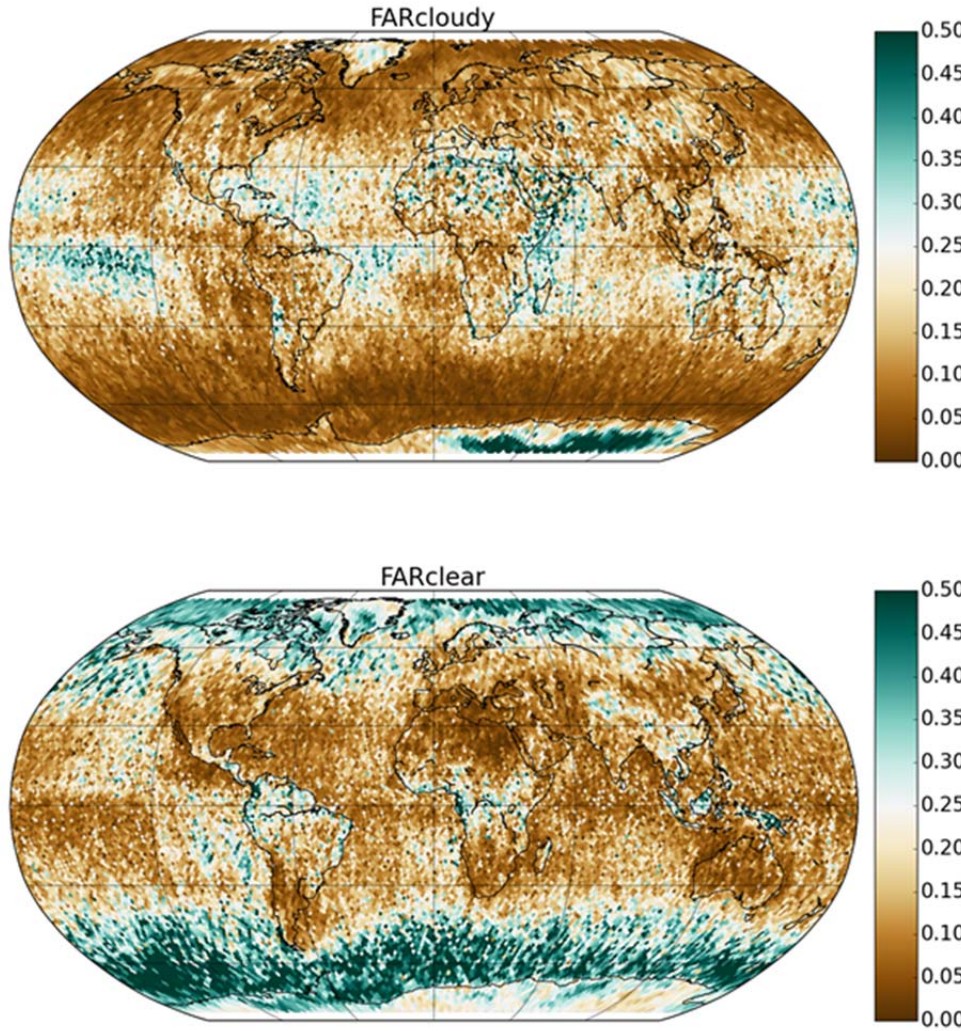

**Figure 11: False alarm rates for cloudy (top) and clear (bottom) predictions for the CLARA-A2 cloud mask achieved after filtering the CALIOP cloud mask with the cloud optical thickness value of 0.225. Same underlying matchup dataset as in Fig. 2. Results are presented in a Fibonacci grid with 75 km resolution.**



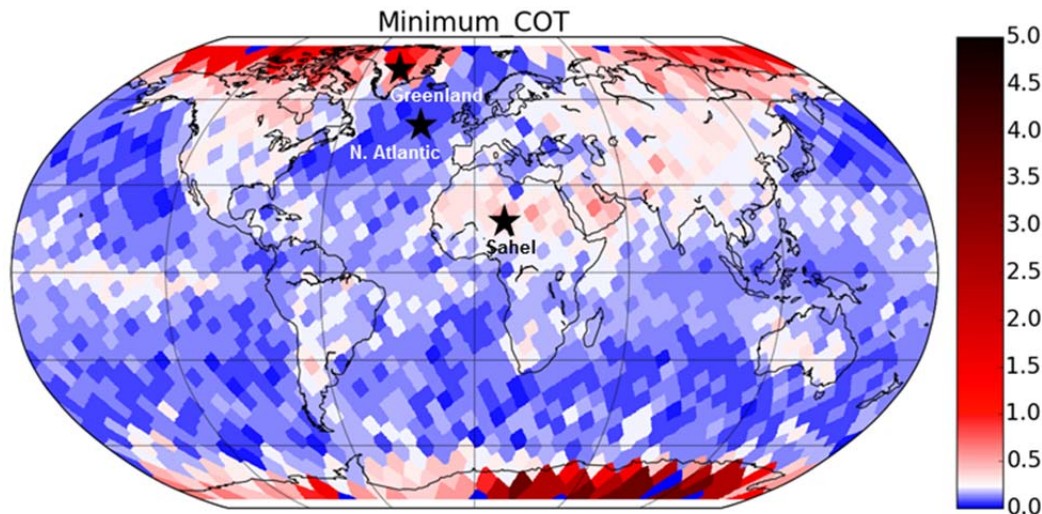

**Figure 12: Global map of estimated cloud detection sensitivity of the cloud mask of CLARA-A2 (see text for explanation). Results are calculated from the same dataset as visualized in Fig. 2 but in a coarser Fibonacci grid resolution of 300 km. Conditions in the three marked locations (black stars) are analysed further in Fig. 13. Values below the global mean value of 0.225 are coloured in blue shades and values above the global mean value are coloured in red shades.**

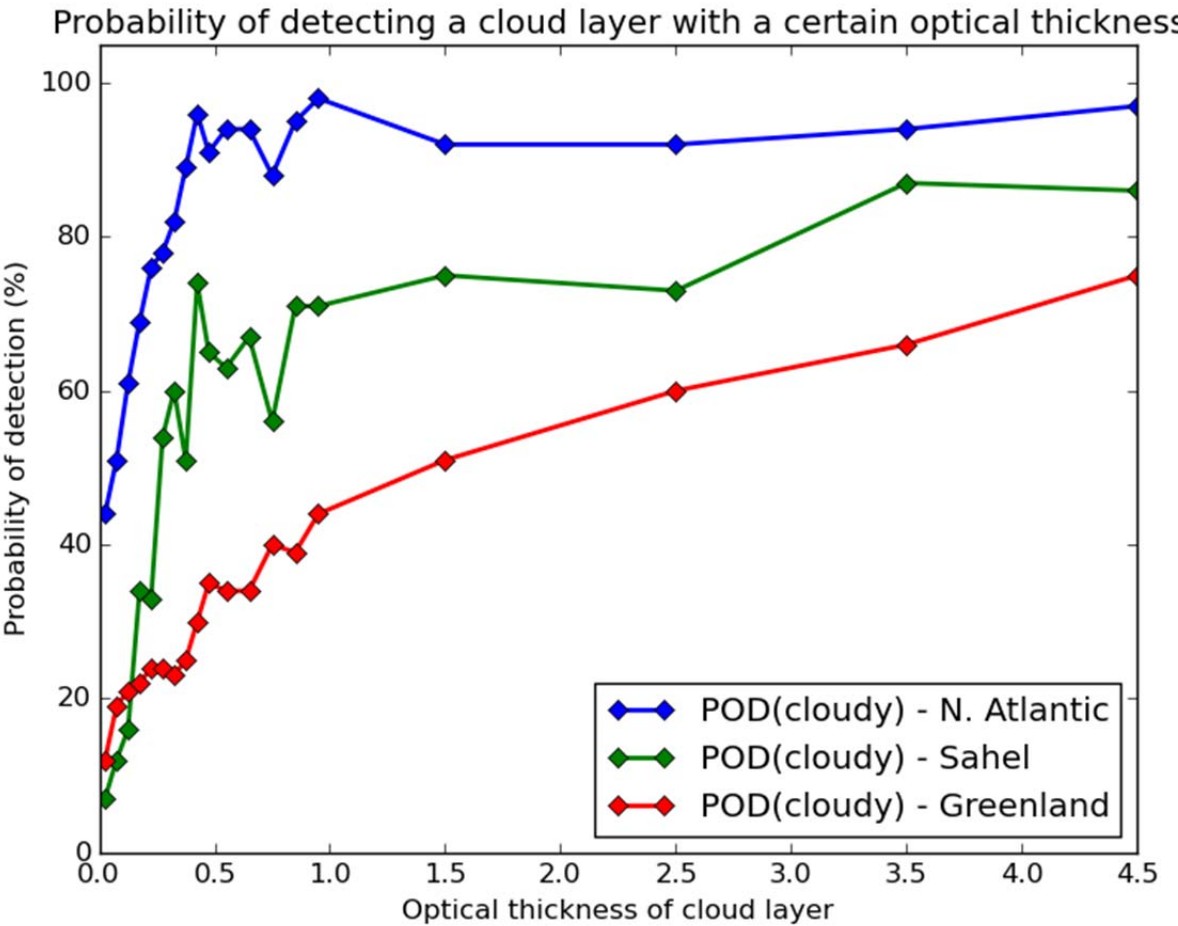

**Figure 13: Same as Fig. 6 but for the individual grid points marked in Fig. 12.**