# Peer review of "Characterization of AVHRR global cloud detection sensitivity based on CALIPSO-CALIOP cloud optical thickness information: Demonstration of results based on the CM SAF CLARA-A2 climate data record"

_Atmospheric Measurement Techniques, 2017_

## Referee Comment (RC1) · Anonymous Referee #1 · 13 Nov 2017

This manuscript evaluates the cloud mask of the CLARA-A2 climate data record (based on passive imagery from AVHRR polar orbiters) with collocated active cloud detections (CALIOP). Another, more general, paper has been published in ACP this year, and this AMT paper focuses exclusively on the cloud mask. This approach is sufficiently well justified, but the paper under review relies too much on the earlier publication (Karlsson et al., 2017; also to some extent on Karlsson et al., 2013) to explain the background. In order to qualify for publication in AMT, revisions need to be made to ensure that it can stand on its on, while not replicating too many of the science results.

In its current state, the paper is hard to review because some of the concepts are not explained sufficient well (specific examples are given below), and because details are left out. In addition, the manuscript is unnecessarily wordy in some places and has basic deficiencies with English/Grammar (for example, "were" is used instead of "where" throughout the manuscript; there are many run-on sentences; punctuation is used too sparingly; use of slang words such as "punish" for a statistical approach that are frequently used by the community, but should be used only where absolutely necessary). Before going into the copy/edit process at AMT, a native speaker should be consulted to ensure logical flow and readability of the manuscript overall.

Despite the criticism of the presentation quality, the content is interesting in that the cloud detection capability is studied as a function of optical thickness and region. Obviously, the POD (probability of cloud detection) depends on surface albedo and emissivity, mechanisms that are identified by the authors. Two comments here:

1) It should be stated more clearly where such findings have been made previously. The author make a point that the regional assessment is new, but there have been previous studies that focused on some of the problematic regions specifically in the Arctic with CALIOP that are not cited here (for example, studies by Gettelman, Kay, L'Ecuyer and a few others).

2) It remains unclear (partially because of the structural problems of the manuscript pointed out above) why there are some regions where cloud cover is overestimated by the passive imagers. One possible explanation is not sufficiently investigated: sub-grid resolution clouds that could be picked up by passive imagers but not by active imagers (if they are outside the FOV). There is some discussion of it, but it remains superficial. Also, active observations are portrayed as the ultimate "judge" for the performance of the cloud mask derived from passive observations, and they shouldn't be. As pointed out by the authors, active observations have their own limitations (sensitivity, FOV, day-vs-night contrasts). The truth is that active cloud observations afford a different perspective on clouds that happens to be less sensitive to the surface reflectivity and

emissivity than that of passive observations. This distinction (and the limitations of both approaches) should be made clear by the authors.

Sequential comments:

General language comments are provided below, but they are far from complete and should only serve as examples to go through the manuscript as a whole before submitting the revised version. Below, a few specific comments regarding the scientific content are given.

p2,L60: Why is CALIOP singled out as important for cloud observations, where in fact MODIS is flown in the A-Train as well. Wouldn't the MODIS observational record, in conjunction with CALIOP, lend itself to a similar study as the one presented here? Of course, its data record is much shorted, but on the other hand, MODIS and CALIOP are collocated all the time, by design.

p2,L60: The limitations of CALIOP (e.g., day time vs. night time detection, noise etc., strategies for thin cloud detection) should be discussed here.

p2,L70: The earlier study by Karlsson is cited here. It should be summarized in at least one paragraph since this paper needs to stand on its own. What was the scope of that manuscript? The extension by CALIOP, on the other hand, are well explained (with the caveats pointed out above).

p3,L79-87: This paragraph should be completely rewritten. The explanation of the field of view of the passive vs. the active instrument is vital for understanding this manuscript, yet it is incomplete. What is the GAC FOV vs. the FOV(passive) vs. the FOV(active, at native vs. aggregated resolution)? What data specifically are dropped? The best way to explain this would be through a simple illustration of the AVHRR pixels vs. the CALIOP FOV of single shots, as well as the aggregation of individual pixels/shots in the various products used in this study. Without this added figure, it will be hard to retrace the steps that were taken in this manuscript.

p3,L90: Which parameter retrievals? How is the radiance inter-calibration and data record homogenization done? Simply referencing Heidinger will not do because the specifics are missing. One of the clear requirements of AMT publications is that anybody reading the paper needs to be able to retrace the steps of a study from the original data to the findings. There is not sufficient detail provided here (or in other parts of the manuscript) to do that.

p4,I118-127: See comment above. These sections cannot be understood without better explanations of the FOVs, data aggregation and homogenization.

p4,I129-134: Provide description of specific NOAA orbits that were included (vs. those that were not). Also, why were MODIS observations NOT used? The minimum information for the NOAA observations are: (a) instrument/satellite names and short description; (b) orbit inclination and equator crossing time; (c) life time of satellite; (d) orbital shifts over time

p4,I148: The theoretical deliberations on cloud mask/cover are insufficiently backed by literature. The paper that comes to mind when talking about the meaning of a "small" or "thin" cloud is that by Koren ("How small is a small cloud"). A short literature study on the topic would be advisable, given that it is the main topic of this article.

p5,I184: "possibly punish AVHRR-based methods in an unfortunate and undeserved way...": three words (punish, undeserved, unfortunate) are inappropriate for a scientific publications. There are multiple occurrences of such "personalized" or "humanized" comments, which should all be translated into objective, rather than "punitive" language.

p5, 1187: The optical thickness threshold of 5 for CALIPSO is higher than usually assumed. If it is necessary for this study to work with such a high threshold, it should be justified, and it should be explained how this is possible (referring to literature where this has been done, or with a dedicated sub-section in this manuscript where it is shown that the lidar does, in fact, allow to go to COD 5, and under which circumstances). p6: There are multiple gaps on this page: The notion of "scores" (and different kinds) are used without sufficient (or any) explanation in this section, or in section 3.3. Too many questions remain, for example, which parameter of what satellite is validated with which other parameter, and how exactly as "score" (of any kind) is established. How is the aggregation done? Why are scores only plotted as a function of COD up to 1, where in fact CODs up to 5 are advertised? What is the "improvement"? If the figures are insufficiently explained, it is not possible to understand. What has been "transformed from cloudy to clear cases" (l212), and how is that done? What is the role of Kuiper vs. hit rate (should be spelled "hit rate", not "hitrate"). Each of the bulleted items of the list on p6/p7 need to be explained and supported with formulae where appropriate. Here again, terms such as "punishing" should be avoided if at all possible. After this paragraph, the reviewer was unable to give this a thorough review because the basics for understanding the remainder of the manuscript were not established. The review is willing to review another version of the manuscript where this has been fixed.

p7,I265: This question is a great one, and at the center of this manuscript. However, the method description below is insufficient. Terms from machine learning ("overtrained") are evoked without explanation how they relate to the manuscript content. Also, here again, CALIOP is represented as the "objective" instrument that AVHRR is validated by where possible – where in fact the two instrument just assess different aspects of a cloud (see comment above).

I278-I304: This seems to wordy and hard to follow since some of the concepts were not introduced.

I306: Now some of the orbits are introduced, but that is too late in the manuscript. In addition to NOAA-18 and NOAA-19, did other data go into the CDR under investigation?

p9,L350: insufficient introduction how systematic and random errors were establish

make it hard to understand Figure 7.

p9,I355-359: Add explanation why AVHRR gives higher cloud cover. It is easy to imagine a scenario where small cumulus clouds would be picked up by AVHRR (even if below its spatial resolution), but not by CALIOP products (for physical reasons). The statistical explanation given here does not seem to be complete and is hard to follow.

p10,369-371: What is Kuiper's score, what's the dominating mode in which case? At this point, some examples that help understanding one score vs. another are provided which is helpful, but that should be done (more systematically) earlier in the manuscript.

p11: "The cloud detection sensitivity is here as high as 1.5"; "all optically thick clouds"... Define what "high" and "thick" means (earlier in the manuscript).

p13, L495-500: Since specific orbits and satellites were not clarified, there's confusion here as to what was actually compared/validated. If it was equally applied to the morning and afternoon orbits (the wording leaves this open), one has to wonder how this would work because CALIOP operated in the afternoon orbit. How can morning cloud cover be "compared" to afternoon cloud cover, considering the significant diurnal cycle of clouds in most regions?

Language comments: p1: "considerably" -> "considerable"

p1,I20: "were" -> "where": multiple occurrences throughout the manuscript

p1: "geographically higher" -> use "surface elevation" instead?

p1,I23-I25: run-on sentence (multiple occurrences); at the very least use punctuation (in this case, a comma after "CDRs") to break it up. Better still, re-write.

p1: "sensor families": a bit unusual for science manuscript, consider revising "family"

p1: add comma before "which" (in most cases; multiple occurrences)

p1: "four decades..." -> "four decades, which qualifies them to be used in climate..."

p2,L51-53: "Linked to this..." Unclear: efforts by whom? stringent with regard to?

p2: The A-Train stands for "Afternoon constellation", not "Aqua Train"

p2: "a project being a part of" -> fix language

p2,I70-73: run-on sentence

p2,L91: MODIS: Introduce upon first occurrence

p3,I117: "including" -> "detecting"

p4, I121: "notice" -> "note"

p4,1148-165 and following: Avoid "you": Not only is this inconsistent with the style of this manuscript, but it is also not advisable for a science manuscript in general. This sounds more like a seminar or talk than a paper at this point in the manuscript. I recommend a complete re-write of this section, as well as a thorough discussion of the meaning of a "cloud mask" (see comment above).

p4,I148: "areal extension" -> "areal extent"

p5,I177-179: Revise English, hard to understand

p5,1184: "possibly punish AVHRR-based methods in an unfortunate and undeserved way..." - see comment above

p5,L199: "of which single shots that were removed..." fix English

p8,L314: "navigation" -> "geolocation"?

p10: "Validation results are probably underestimated" -> What does that mean?

p10: "compared to if only showing results based" -> fix English

---

## Referee Comment (RC2) · Anonymous Referee #2 · 14 Nov 2017

General comments

The paper presents an unprecedented evaluation of satellite-based cloud climatology (CMSAF's CLARA-A2) against CALIPSO/CALIOP performed at the global scale. Despite some limitations of CALIOP dataset discussed in the paper, it is the only currently considerable reference for cloud retrievals covering oceans, polar regions and other areas of very sparse cloud observations and measurements. Such evaluation has become possible with the sufficiently long CALIOP dataset. The authors also present an analysis of the CLARA-A2 cloud detection sensitivity, i.e. the threshold in the cloud

optical thickness (COT) above which the cloud detection algorithm detects more than 50% of clouds. Screening the CALIOP data with COT below the globally-averaged detection sensitivity allows for "more realistic" evaluation, i.e. taking into account the difference between the sensitivity of CALIOP (active sensor) and AVHRR (passive sensor). Therefore, the paper will be an important first step towards proposing described validation methodology for the list of standard validation activities performed before releases of new cloud climate data records.

While the content of the paper is novel, valuable and appropriate for the publication in AMT, the paper structure should be significantly improved. Finally, the paper has some grammar and language issues, which should be addressed. They are mostly related to the syntax, i.e. sentence length and inappropriate word order. Some examples are indicated in the following, but the whole manuscript should be revised.

Specific comments

(1) The title of the paper is a bit misleading. "Detailed characterization" suggests that the evaluation of the CDR is more detailed than the standard one, e.g. provided in CLARA-A2 validation report. However, the collocations of AVHRR and CALIOP are limited to NOAA-18 and NOAA -19, afternoon orbits and 10-year period only (from 30y+ of the CDR). Taking into account that one of the challenges in deriving CDR is stable performance in time, the evaluation presented in the manuscript cannot serve as an evaluation of CLARA-A2 CDR.

(2) Objectives of the study should be described better in the Introduction. In relation to (1), it should be clear if the aim is to present new methodology using a subset of CLARA-A2 as a an example or to evaluate CLARA-A2.

(3) The current discussion section is a mix of discussion remarks and conclusions. I recommend to separate the two. In the results' section, there are also interpretations, which are hypothetical (they often start with "we believe", "we claim") and should be moved to the discussion. Otherwise it is often difficult to judge which statements are

really supported by the results achieved in this study.

(4) The analysis of detection sensitivity reveals some interesting non-expected results. One is that CLARA performance is not better at dark and warm ocean surfaces (L374-375). The hypothesis this is due to sampling and geometry of AVHRR and CALIOP FOVs needs more explanation. The problem was detected here, because it leads to unexpected results. However, how to measure a possible effect of this issue on results in other situations, regions, etc.? I would consider a separate section (or paragraph) in the discussion.

(5) Is the cloud detection sensitivity a measure of CDR performance itself? There is no discussion if 0.225 signifies good or bad CLARA performance. One can imagine the same analysis (i.e. evaluation against screened CALIOP data), but with the estimated cloud detection sensitivity of, say, 0.5. Please elaborate on that. In addition, since the authors recommend the methodology to be widely used (e.g. in CFMIP), more detailed guidelines would be appreciated. For instance, when applied to different passive-sensor-based CDRs, should the cloud detection sensitivity be always recalculated?

L50, "be very accurate to be able.." - please be more specific, e.g. referring to GCOS recommendations

L82, "FOV resolution" - field of view does not have a resolution, I would keep FOV and remove 'resolution' (or 'size' in other places in the manuscript)

L92, 'various parameters retrieval' - be more precise

L117-119, "Thus CALIOP products..." - please provide a reference for this statement

L126-127, "...claiming that useful...seems to be available" - based on which results?

L140-L145, If these improvements are relevant for the study, please explain them better

L150, "..how thin or thick..." - do you mean optically, in height?

L151, "The second aspect..." - something is wrong with the syntax, please rephrase

L192, The investigation if the method used by Karlsson and Johansson (2013) is still applicable to the new CLAY version should be listed as one of the paper objectives (i.e. already in the introduction). The results (L206-223) should be moved from this paragraph to the Section 4.

L249, why 'CLARA-A2 cloud masks', i.e. in plural?

L250, "This approximation is acceptable.." - provide a reference

L288, Why 50% is an appropriate threshold for the cloud detection probability?

L326, "..but we still believe..." - what if the authors are wrong?

L328 and L349, Please consider giving different section names. These two are not very informative.

L369, "This contributes..." - it's not clear what is meant. Please rephrase.

L361-404 – It would be easier to follow the text divided in paragraphs

L381, "We first conclude..." - is it based on actual results or it is a hypothesis?

L406-407, Wrong syntax, please rephrase

L407, "...is undoubtedly a clear improvement", please explain why?

L436-438, Please explain better, preferably in a separate paragraph in the Discussion

Figure 11, it would be useful to have a different color scale (e.g. as in previous figures), with a shift between colours at 0.225. Otherwise it is difficult to see the 'edge' at 0.225

Figure 12, it would be useful to add FAR or KSS here. POD alone does not reveal the true performance of the cloud detection, as it gives no information about false alarms.

Technical corrections

many times in the manuscript, use a lower case after using a colon in the sentence

L11, should be "sensitivity of the detection"

L14, results of? Please rephrase.

L16, "portions" looks weird in this context

L23, use elevation or altitude instead of "highest"

L66, remove "

L132, 70 N/S

L200, remove second "be"

L230, should be 'where' not 'were'

L237, give colon after 'namely'

L317, "...a minimum of the number or matchups" should be "a minimum number of matchups"

L371, should be "Kuipers score"

L570, incorrect order of references

---

## Author Comment (AC1) · 17 Nov 2017

First reply to Referee 1's review of the AMTD paper

**"Detailed characterisation of AVHRR global cloud detection performance of the CM SAF CLARA-A2 climate data record based on CALIPSO-CALIOP cloud information"**
by
Karl-Göran Karlsson and Nina Håkansson, SMHI

**Repeating general comment, part 1:**

**This manuscript evaluates the cloud mask of the CLARA-A2 climate data record (based on passive imagery from AVHRR polar orbiters) with collocated active cloud detections (CALIOP). Another, more general, paper has been published in ACP this year, and this AMT paper focuses exclusively on the cloud mask. This approach is sufficiently well justified, but the paper under review relies too much on the earlier publication (Karlsson et al., 2017; also to some extent on Karlsson et al., 2013) to explain the background. In order to qualify for publication in AMT, revisions need to be made to ensure that it can stand on its on, while not replicating too many of the science results.**

**Reply:**

The two referred papers (especially Karlsson et al., 2013) are important papers in that they set the stage and define the framework for how to perform the matchups between AVHRR and CALIOP data. We get the feeling from some of the comments that we need to clarify the framework even further (i.e., that it is not enough to just provide the references). Thus, we will include a short summary of the most important points concerning the basic matching or collocation methodology (see also reply to **general comment, part 4** further below).

Regarding the justification of this paper and the question whether it adds anything new compared to the paper by Karlsson et al. (2017), we claim that one important objective of this study was to investigate the impact of upgrading the results by using the new CALIPSO-CALIOP version 4 dataset (which indeed is the main topic of the Special AMT Issue too which the paper was submitted). The previously mentioned validation efforts were all based on CALIPSO-CALIOP version 3 data. However, another and more important objective was to

show that the CALIPSO-CALIOP dataset can be used to investigate much more in detail the cloud detection limitations of one particular cloud screening method (like the one used for CLARA-A2) than what has been presented before. The concept of Cloud Detection Sensitivity (as illustrated by results in Figure 11) is a new approach which we hope can become a standard tool for a more objective evaluation of cloud climate data records in the future. Its main advantage is that it can be considered as a universal method, not depending specifically on the actually studied AVHRR dataset. It is a method based on a special organization of the CALIOP cloud dataset by use of estimated cloud optical thickness sub-categories. These results, being organized in cloud optical thickness sub-categories, can be compared to any other collocated satellite-based dataset.

We will emphasize better the objectives of the study and try to highlight better the results and the potential of the derived Cloud Detection Sensitivities in current and future studies.

**Repeating general comment, part 2:**

**In its current state, the paper is hard to review because some of the concepts are not explained sufficient well (specific examples are given below), and because details are left out. In addition, the manuscript is unnecessarily wordy in some places and has basic deficiencies with English/Grammar (for example, "were" is used instead of "where" throughout the manuscript; there are many run-on sentences; punctuation is used too sparingly; use of slang words such as "punish" for a statistical approach that are frequently used by the community, but should be used only where absolutely necessary). Before going into the copy/edit process at AMT, a native speaker should be consulted to ensure logical flow and readability of the manuscript overall.**

**Reply:**

As being non-native English authors, we admit limitations in the ability to produce perfect quality (scientific) English text. We thank the reviewer for pointing out the most common errors and we will do our best in eliminating them. We will certainly consult a native speaker before submitting the final version of the manuscript.

**Repeating general comment, part 3:**

**Despite the criticism of the presentation quality, the content is interesting in that the cloud detection capability is studied as a function of optical thickness and region. Obviously, the POD (probability of cloud detection)**

**depends on surface albedo and emissivity, mechanisms that are identified by the authors. Two comments here:**

**1) It should be stated more clearly where such findings have been made previously. The author make a point that the regional assessment is new, but there have been previous studies that focused on some of the problematic regions specifically in the Arctic with CALIOP that are not cited here (for example, studies by Gettelman, Kay, L'Ecuyer and a few others).**
**Reply:**

The knowledge of the dependency on surface characteristics (e.g. albedo or emissivity) for the possibility to separate clouds from Earth surfaces in satellite imagery is nothing fundamentally new. Rather, it is a well-established and well-known fact in the satellite user community. The reason is obvious: All cloud screening methods depend on the ability to find enough of contrast between clouds and underlying surfaces in the investigated images. This is valid for all spectral regions - be it visible, near-infrared, short-wave infrared or infrared. Multi-spectral methods will have the best capability since the use of many spectral channels increases the probability that at least one spectral channel will offer enough of contrast between clouds and Earth surfaces. This explains e.g. the high quality of cloud datasets from MODIS (with access to up to 36 useful spectral channels).

The challenges here are naturally largest at high latitudes and near the poles where we have both bright Earth surfaces (snow, ice) and very cold surface temperatures (very similar or even colder than clouds which normally are warmer than clouds in other regions). This explains the special interest here (as exemplified by the mentioned papers).

We can certainly add some of these references but the most important thing is to even stronger emphasize that the proposed method offers a method to monitor these problems globally and not just in specific regions. This is the big advantage of the method. Our statement about the novelty of the regional assessment should be interpreted as that the method offers both a monitoring of mean global conditions but also a regional monitoring including all regions on Earth and not just some selected ones.

**Repeating general comment, part 4:**

**2) It remains unclear (partially because of the structural problems of the manuscript pointed out above) why there are some regions where cloud cover is overestimated by the passive imagers. One possible explanation is**

**not sufficiently investigated: sub-grid resolution clouds that could be picked up by passive imagers but not by active imagers (if they are outside the FOV). There is some discussion of it, but it remains superficial.**

**Also, active observations are portrayed as the ultimate "judge" for the performance of the cloud mask derived from passive observations, and they shouldn't be. As pointed out by the authors, active observations have their own limitations (sensitivity, FOV, day-vs-night contrasts). The truth is that active cloud observations afford a different perspective on clouds that happens to be less sensitive to the surface reflectivity and emissivity than that of passive observations. This distinction (and the limitations of both approaches) should be made clear by the authors.**

**Reply:**

We agree that we could have been clearer in the discussion of aspects that are related to the different FOVs of AVHRR and CALIOP. Some discussion is included on page 10 (lines 366-404) and on page 12 (lines 465-471) but this can be improved. Since another reviewer also have pointed out more or less the same thing we suggest to do the following:

1. We will introduce a short summary of the underlying basic method of how we matched AVHRR and CALIPSO data. It seems the current referencing to the original paper by Karlsson and Johansson (2013) (which describes the matching method) is not enough for a full understanding. We need to recapitulate the method's most important aspects also in this paper.

2. We will add a clear illustration (new figure) of how matched high-resolution AVHRR FOVs relate to the CALIPSO-CALIOP FOVs within a nominal AVHRR GAC pixel. This would help understanding the problem.

3. We will expand the discussion of these results in a new Discussion section. Thus, the current Discussion section will be split into one separate Discussion section and one final Conclusion section. The problem of inter-comparing CALIOP data with other satellite data in cases of highly scattered and fractioned cloudiness needs to be discussed. In our opinion this aspect has been largely overlooked in many previous papers using CALIPSO-CALIOP data as the main validation source.

The question on why there seems to be regions where cloudiness is overestimated is interesting but the reasons behind this is beyond the scope of this paper. We do express some qualified guesses about the reasons for some of

the found deviations in the study but, basically, this is really up to the algorithm originators to further analyze and explain in subsequent studies. However, we think that it cannot really be related solely to "sub-grid resolution clouds that could be picked up by passive imagers but not by active imagers (if they are outside the FOV)". This mismatch can definitely occur for individual AVHRR GAC pixels and for individual orbits but when summed up in a climatology based on thousands of orbits such biases will end up to be either very low or non-existing. Simply since the opposite case (clouds picked up by active sensors and not by passive sensors) is just as likely to occur. We will explain that in relation to the illustration envisaged in point 2 above. But what is important is that the precision (variance) of the estimated mean cloud cover will degrade (i.e., higher RMS errors) when this occurs and this is emphasized in our discussion.

Regarding the choice of CALIPSO-CALIOP data as the "ultimate" judge, we both agree and disagree with the Reviewer's opinions. Admittedly, active data has its limitation where the FOV representability in relation to the AVHRR GAC FOV is perhaps one of the largest (as discussed above). However, for clouds with scales larger than the AVHRR GAC FOV (5 km) we still claim that no other observation reference can provide a better estimation of global cloud presence and distribution than the CALIPSO-CALIOP dataset. The big advantage with the CALIOP information is that we measure the lidar reflection from real cloud particles (in the CALIPSO-CALIOP version 4 dataset also quite confidently separated from aerosol particles) and from the backscatter energy we can also for the thinnest clouds estimate with high accuracy the optical thickness of the cloud layers (up to a certain maximum value). No other sensor can provide the same. MODIS data is an alternative but in our opinion the MODIS dataset share many of the problems experienced by dataset produces from most multispectral passive sensors (AVHRR, SEVIRI, VIIRS, ABI, etc.) and this is basically explained by the fact that the measurement always contain a mix of contribution from clouds, the atmosphere and the surface (especially in the cases of thin clouds). We cannot be sure that we only measure the impact of the cloud itself. For an active sensor we do not have the same problem. Nevertheless, we will try to include a better description of the difference and the pros and cons of the active and passive information. A final remark is that one must consider the importance of having very accurate estimations of cloud optical depth for the very thinnest clouds in order to carry out a study like this. Here the CALIPSO-CALIOP measurement is quite superior to MODIS for which estimations of cloud optical thickness of the thinnest clouds have high uncertainties.

**In the following we will address selected short comments (which are not simply editorial):**

**p2,L60: Why is CALIOP singled out as important for cloud observations, where in fact MODIS is flown in the A-Train as well. Wouldn't the MODIS observational record, in conjunction with CALIOP, lend itself to a similar study as the one presented here? Of course, its data record is much shorted, but on the other hand, MODIS and CALIOP are collocated all the time, by design.**

**Reply:**

We just gave some arguments in the reply above to **general comment, part 4**. It is our opinion that CALIOP data is a better reference in the sense that the measurement information is free from surface (and atmospheric water vapour and aerosol) dependence. However, even more important is that we cannot use MODIS data for the cloud detection sensitivity study since the cloud detection sensitivity of MODIS is probably not very different from AVHRR. More clearly, we repeat that we need access to very accurate cloud optical thickness estimations for very thin clouds in order to make such a study. The uncertainty of the MODIS-derived optical thickness in this optical thickness interval (values less than 1.0) is too high and at least much higher than for CALIOP-derived optical thickness. We will point this out more clearly in the manuscript to better justify the choice of CALIOP information as our reference.

It would actually be very interesting to do a similar study of the MODIS C6 cloud detection sensitivity with the same method as presented here. We would expect some improvements compared to CLARA-A2. Figure 6d in the CLARA-A2 paper in ACP indicates an almost constant bias in cloud cover of about 5 % for MODIS C6 over all latitudes (more clouds observed by MODIS). The question is then if the global distribution of the cloud detection sensitivity will decrease with a constant value everywhere or are there possibly regional differences?

**p2,L60: The limitations of CALIOP (e.g., day time vs. night time detection, noise etc., strategies for thin cloud detection) should be discussed here.**

**Reply:**

OK, we will provide more information here.

**p2,L70: The earlier study by Karlsson is cited here. It should be summarized in at least one paragraph since this paper needs to stand on its**

own. **What was the scope of that manuscript? The extension by CALIOP, on the other hand, are well explained (with the caveats pointed out above).**

**Reply:**

OK, see point 1 above in the reply to **general comment, part 4**.

**p3,L79-87: This paragraph should be completely rewritten. The explanation of the field of view of the passive vs. the active instrument is vital for understanding this manuscript, yet it is incomplete. What is the GAC FOV vs. the FOV(passive) vs. the FOV(active, at native vs. aggregated resolution)? What data specifically are dropped?**
**The best way to explain this would be through a simple illustration of the AVHRR pixels vs. the CALIOP FOV of single shots, as well as the aggregation of individual pixels/ shots in the various products used in this study. Without this added figure, it will be hard to retrace the steps that were taken in this manuscript.**

**Reply:**

OK, see point 2 above in the reply to **general comment, part 4**.

**p3,L90: Which parameter retrievals? How is the radiance inter-calibration and data record homogenization done? Simply referencing Heidinger will not do because the specifics are missing. One of the clear requirements of AMT publications is that anybody reading the paper needs to be able to retrace the steps of a study from the original data to the findings. There is not sufficient detail provided here (or in other parts of the manuscript) to do that.**

**Reply:**

The parameters we mention concern the different variables included in the entire CLARA-A2 dataset. Apart from cloud amount there are 7 different cloud properties, a surface albedo estimation and an estimation of surface radiation budget parameters.
It is true that there is still no follow-on paper to Heidinger et al. (2010) describing the upgraded calibration equations. But there is a recent publication in the GSICS Newsletter describing the associated PyGAC preprocessing software
(ftp://ftp.library.noaa.gov/noaa_documents.lib/NESDIS/GSICS_quarterly/v11_n

[o2_2017.pdf](o2_2017.pdf)) . PyGAC contains the final calibration which was used for the CLARA-A2 processing and is available as an open source package. We will add this reference to the manuscript.

**p4,l118-127: See comment above. These sections cannot be understood without better explanations of the FOVs, data aggregation and homogenization.**

**Reply:**

See the reply to **general comment, part 4**.

**p4,l129-134: Provide description of specific NOAA orbits that were included (vs. those that were not). Also, why were MODIS observations NOT used? The minimum information for the NOAA observations are: (a) instrument/satellite names and short description; (b) orbit inclination and equator crossing time; (c) life time of satellite; (d) orbital shifts over time**

**Reply:**

See the reply to **general comment, part 4**. We will try to cover all those aspects. The MODIS question has already been dealt with in the reply to the comment for **p2,L60.**

**p4,l148: The theoretical deliberations on cloud mask/cover are insufficiently backed by literature. The paper that comes to mind when talking about the meaning of a "small" or "thin" cloud is that by Koren ("How small is a small cloud"). A short literature study on the topic would be advisable, given that it is the main topic of this article.**

**Reply:**

Thanks for this advice. We will take a closer look and add the adequate literature references.

**p5,l184: "possibly punish AVHRR-based methods in an unfortunate and undeserved way: : :": three words (punish, undeserved, unfortunate) are inappropriate for a scientific publications. There are multiple occurrences**

of such "personalized" or "humanized" comments, which should all be translated into objective, rather than "punitive" language.

**Reply:**

We will remove the used non-scientific terminology.

**p5, l187: The optical thickness threshold of 5 for CALIPSO is higher than usually assumed. If it is necessary for this study to work with such a high threshold, it should be justified, and it should be explained how this is possible (referring to literature where this has been done, or with a dedicated sub-section in this manuscript where it is shown that the lidar does, in fact, allow to go to COD 5, and under which circumstances).**

**Reply:**

We admit that we do not have good support in the literature for stretching the useful upper limit of CALIPSO-derived COD to 5. However, in the description of the upgrade to CALIPSO-CALIOP version 4 it is also emphasized that previous cloud optical thicknesses in version 3 were generally underestimated. This is also clearly indicated in Figure 2 in the manuscript. Whether this increase entirely justifies moving the upper limit to 5 is still not clear.

We do have more indications from our own investigations that an adjustment of the upper limit seems possible. In a study related to a paper by Riihelä et al (2017) we investigated the correlation between CALIPSO-estimated and CLARA-A2 estimated CODs over various surfaces (with snow surfaces over Greenland as the main target). However, when isolating the collocated results over ice free ocean surfaces at high latitudes (noting that over a dark surface also the AVHRR-based estimations should be more accurate), we could clearly see a good correlation between the two estimations up to about COD=5 (see figure below):

[Figure]

Although this is not a perfect illustration (not included in Riihelä et al, 2017, but maybe considered for a follow-up paper) it shows how CLARA-A2-estimated optical depths compare to CALIOP-estimated optical depths in the range 0-15. Over a dark ocean surface the majority of values agree pretty well but what is clear is that an increasing number of cases (for higher optical depths) CLARA-A2 values saturates at 100 for CALIOP-values exceeding approximately 4 (noticeable at top of the figure). This reflects the inability of CALIOP to provide reasonable optical thicknesses for optically thick clouds. But, we made the conclusion that values compare pretty well even up to an optical thickness of 4-5 and this was one of the reasons why we decided to use the CALIOP interval 0-5 for this particular study (for AMT).

It is this finding that made us to use the maximum limit of 5 in this particular study. Unfortunately, in the end, we did not include this part of the inter-comparison in the finally published paper by Riihelä et al. (2017).

We propose that we keep the original maximum value of 5 in our plots but add a remark that values near this upper end are uncertain. The upper limit is not crucial for the findings of our study since in most cases the cloud detection sensitivity is considerably lower than 5. Only for some positions over Greenland

and Antarctica we approach these high values but whether the value is 3 or 5 here does not really matter since it deviates anyhow very much from the values found on other places (which is the main message).

The mentioned reference is the following:

Riihelä, A., Key, J. R., Meirink, J. F., Munneke, P. K., Palo, T., & Karlsson, K.-G. (2017). An intercomparison and validation of satellite-based surface radiative energy flux estimates over the Arctic. Journal of Geophysical Research - Atmospheres, 122(9), 4829–4848. https://doi.org/10.1002/2016JD026443

**p6: There are multiple gaps on this page: The notion of "scores" (and different kinds) are used without sufficient (or any) explanation in this section, or in section 3.3. Too many questions remain, for example, which parameter of what satellite is validated with which other parameter, and how exactly as "score" (of any kind) is established.**
**How is the aggregation done? Why are scores only plotted as a function of COD up to 1, where in fact CODs up to 5 are advertised? What is the "improvement"? If the figures are insufficiently explained, it is not possible to understand. What has been "transformed from cloudy to clear cases" (l212), and how is that done? What is the role of Kuiper vs. hit rate (should be spelled "hit rate", not "hitrate"). Each of the bulleted items of the list on p6/p7 need to be explained and supported with formulae where appropriate. Here again, terms such as "punishing" should be avoided if at all possible.**
**After this paragraph, the reviewer was unable to give this a thorough review because the basics for understanding the remainder of the manuscript were not established.**
**The reviewer is willing to review another version of the manuscript where this has been fixed.**

**Reply:**

We will improve the description here to improve the understanding of the method and the results. We will have these questions in mind when dealing with point 1 in the reply to **general comment, part 4**. We are certainly grateful for the reviewer's willingness to check the revised manuscript.

**p7,l265: This question is a great one, and at the center of this manuscript. However, the method description below is insufficient. Terms from machine learning ("overtrained") are evoked without explanation how they relate to the manuscript content. Also, here again, CALIOP is represented as the "objective" instrument that AVHRR is validated by where possible – where**

**in fact the two instrument just assess different aspects of a cloud (see comment above).**

**Reply:**

Yes, this is the core topic of the paper. In our opinion, the method of determining the Cloud Detection Sensitivity is a way of utilizing the sensitivity difference between the two sensors in the most optimal way.
We will improve the description here (the mentioned aspects have already been commented in replies to similar comments above).

**p7, l278-l304: This seems to wordy and hard to follow since some of the concepts were not introduced.**

**p8, l306: Now some of the orbits are introduced, but that is too late in the manuscript. In addition to NOAA-18 and NOAA-19, did other data go into the CDR under investigation?**

**Reply:**

To be dealt with as indicated in the reply to **general comment, part 4**. The used NOAA-18 and NOAA-19 data (being matched with CALIPSO) is exactly the same dataset as was used for the evaluation in the CLARA-A2 paper by Karlsson et al, 2017). However, in that study also results for morning satellite data (NOAA-17, METOP-A, METOP-B) were presented, although only valid over a small latitude band around 70 degree latitude. Since this study focus on global conditions we excluded the morning satellite part of the dataset. The matching with morning satellites will also introduce a new type of matching problems. This will be discussed in the revised manuscript in connection with the discussion of the figure to be introduced in point 2 of the reply to **general comment, part 4**.

**p9,L350: insufficient introduction how systematic and random errors were establish make it hard to understand Figure 7.**

**Reply:**

We will improve both the description how these quantities were derived and how they shall be interpreted in Figure 7.

**p9,l355-359: Add explanation why AVHRR gives higher cloud cover. It is easy to imagine a scenario where small cumulus clouds would be picked up by AVHRR (even if below its spatial resolution), but not by CALIOP**

**products (for physical reasons). The statistical explanation given here does not seem to be complete and is hard to follow.**

**Reply:**

We would claim that the opposite situation (i.e., clouds picked up by CALIOP but not by AVHRR) is also very likely (see reply to reply to **general comment, part 4**.). Thus, it is not obvious that one can use this explanation to explain higher AVHRR cloud cover.

Again, it is not the objective of this paper to explain the deviations we see but rather to provide an as sensible and trustworthy validation result as possible. The reasons for the deviations have to be explained by those being responsible for the actual algorithms. We do give some suggestions but in the end this has to be verified by the algorithm developers.

However, the results in Figure 7 illustrate a more general problem. One has to remember that all results in Figures 6-10 are based on comparison with a CALIOP cloud mask filtered at cloud optical thickness 0.225 (the latter being the global mean cloud detection sensitivity). But regionally, the value of the cloud detection sensitivity varies a lot (see Figure 11). We actually propose to change Figure 11 so that the relation to the mean value of 0.225 is made clearer. This is what we have in mind:

[Figure]

In this plot all values below the mean value 0.225 are plotted in blue colours and values above 0.225 in red colours. This colour representation also indicates better the highest values in the polar areas (not properly visualized in the current Figure 11).

Notice here that most oceanic areas are coloured blue and that the positive bias in Figure 7 is also mostly occurring over oceanic areas. By using a CALIOP cloud mask with cloud optical thickness being cut at 0.225 as our validation reference, we are then ignoring a substantial fraction of originally detected clouds below this cloud optical thickness limit over ocean surfaces. But these clouds are to a large extent actually detected in the CLARA-A2 results. Thus, it leads to an apparent overestimation of cloudiness over ocean in Figure 7 (notice that we are filtering CALIOP data and not CLARA-A2 data). This illustrates how difficult the estimation of general validation scores really is. More clearly, regardless of using a filtered CALIOP cloud mask or not when validating, there are always disadvantages. In that sense, the results expressed by the globally resolved cloud detection sensitivity is a much more objective visualization of the cloud detection performance provided that also a separate evaluation of false alarm rates are made. We repeat the following statement from section 3.4 (lines 295-300): "An important additional or complementary parameter in this context would be the false alarm rate in the unfiltered case (FARcloudy(tau=0)) since this parameter is not depending on any filtering of thin clouds". We will add this as an important result and recommendation in the Conclusions section.

Finally, the most correct way of calculating and plotting the Bias in Figure 7 would have been to actually use the derived grid-resolved Cloud Detection Sensitivities in Figure 11 as representing the most appropriate CALIOP cloud mask (i.e., the filtered cloud optical thickness) for validation. We had this option in mind but we realized that this probably needs a much larger sample dataset in order to calculate stable statistics (since it requires calculation based on only those samples existing for every single grid point). Figure 12 illustrates that the available number of samples in each grid point is still rather small which makes the estimation of statistical parameters on this scale rather uncertain. But it can be considered for the future if the time series of CALIPSO collocations can be extended with several more years.

**p10,369-371: What is Kuiper's score, what's the dominating mode in which case? At this point, some examples that help understanding one score vs. another are provided which is helpful, but that should be done (more systematically) earlier in the manuscript.**

**Reply:**

To be dealt with as indicated in the reply to **general comment, part 4**.

**p11: "The cloud detection sensitivity is here as high as 1.5"; "all optically thick clouds": : : Define what "high" and "thick" means (earlier in the manuscript).**

**Reply:**

OK.

**p13, L495-500: Since specific orbits and satellites were not clarified, there's confusion here as to what was actually compared/validated. If it was equally applied to the morning and afternoon orbits (the wording leaves this open), one has to wonder how this would work because CALIOP operated in the afternoon orbit. How can morning cloud cover be "compared" to afternoon cloud cover, considering the significant diurnal cycle of clouds in most regions?**

**Reply:**

We will try to clarify this better. This study only dealt with comparisons with afternoon satellites. This was clearly stated on page 8 lines 306-308 and the reason for restricting it to afternoon satellites have been mentioned several times above in various replies to comments and questions.

The reason that we still brought up the case of morning satellites at page 13 is explained by the fact that readers of this paper (and reviewers!) would most likely start wondering how these results will relate to morning satellite data (representing almost 40 % of the entire CLARA-A2 data record). Matchups between CALIPSO and morning satellites are possible but only near the high latitude of 70 degrees on both hemispheres where the orbital tracks crosses between the two satellites.
What maybe confuses the Reviewer is that we state that some comparisons had been done also for morning satellites. However, this relates to the results in the standard CLARA-A2 validation report (of which some were presented in the CLARA-A2 paper by Karlsson et al., 2017) and not to this particular study. We will emphazise this circumstance even clearer to avoid confusion.

The last question here is very relevant and interesting. The diurnal cycle of cloudiness is of course leading to differences which makes a direct comparison of results difficult (even at the latitude band around 70 degrees where we have matchups from both afternoon and morning satellites). Anyhow, we can first conclude that in the night part of an afternoon orbit and in the corresponding night part of morning orbits we have exactly the same AVHRR measurements. Thus, the only additional difference expected might come from diurnal cycle effects which probably are quite small for the dark part of the day at these high latitudes.

The largest differences are instead expected in the illuminated part of the day since we will then use AVHRR channel 3a (at 1.6 microns) for the morning satellites while AVHRR channel 3b (at 3.7 microns) is still used for the afternoon satellites. The comment on line 497 stating that we have seen good agreement here means that even for the illuminated case we have good correspondence between afternoon and morning satellites. For the region covered by morning matchups we do not see large differences with corresponding results from afternoon satellites. This is encouraging and it indicates that the two spectral channels provide more or less the same cloud screening information (while for cloud property estimations, like optical thickness, we expect much larger differences). We also think that at these high latitudes we will probably not be very much affected by the diurnal cycle in cloudiness.

The statement about the good agreement were not meant to be general in the global sense but just saying that, where we can inter-compare the data from the two orbit constellations, the agreement appears to be good. Additional studies are however needed to evaluate the global performance of morning satellites and we clearly indicate a way forward here (by using CATS data – see lines 500-502).

**FINAL REMARKS:**

- We are extremely grateful for the suggested editorial, syntax and language improvements. These are invaluable for non-native English writers like us!

- We also express our appreciation of the reviewer's large effort leading to this very detailed review.

---

## Author Comment (AC2) · 17 Nov 2017

First reply to Referee 2's review of the AMTD paper

**"Detailed characterisation of AVHRR global cloud detection performance of the CM SAF CLARA-A2 climate data record based on CALIPSO-CALIOP cloud information"**
by
**Karl-Göran Karlsson and Nina Håkansson, SMHI**

**Repeating general comments:**

The paper presents an unprecedented evaluation of satellite-based cloud climatology (CMSAF's CLARA-A2) against CALIPSO/CALIOP performed at the global scale. Despite some limitations of CALIOP dataset discussed in the paper, it is the only currently considerable reference for cloud retrievals covering oceans, polar regions and other areas of very sparse cloud observations and measurements. Such evaluation has become possible with the sufficiently long CALIOP dataset. The authors also present an analysis of the CLARA-A2 cloud detection sensitivity, i.e. the threshold in the cloud optical thickness (COT) above which the cloud detection algorithm detects more than 50% of clouds. Screening the CALIOP data with COT below the globally-averaged detection sensitivity allows for "more realistic" evaluation, i.e. taking into account the difference between the sensitivity of CALIOP (active sensor) and AVHRR (passive sensor). Therefore, the paper will be an important first step towards proposing described validation methodology for the list of standard validation activities performed before releases of new cloud climate data records.

While the content of the paper is novel, valuable and appropriate for the publication in AMT, the paper structure should be significantly improved. Finally, the paper has some grammar and language issues, which should be addressed. They are mostly related to the syntax, i.e. sentence length and inappropriate word order. Some examples are indicated in the following, but the whole manuscript should be revised.

**Reply:**

We thank the reviewer for this positive evaluation. We notice the request for a reorganization of the paper (also demanded by other reviewers) and we will do our best to accomplish this. We will reply to the specific comments below.

**Repeating specific comment 1:**

**The title of the paper is a bit misleading. "Detailed characterization" suggests that the evaluation of the CDR is more detailed than the standard one, e.g. provided in CLARA-A2 validation report. However, the collocations of AVHRR and CALIOP are limited to NOAA-18 and NOAA - 19, afternoon orbits and 10-year period only (from 30y+ of the CDR). Taking into account that one of the challenges in deriving CDR is stable performance in time, the evaluation presented in the manuscript cannot serve as an evaluation of CLARA-A2 CDR.**

**Reply:**

Yes, we understand this remark and we agree that the validation presented here cannot be fully representative of a validation of the entire 34-year CLARA-A2 data record. But we still argue that the validation presented here is improved and more detailed than the validation (i.e., the CALIPSO-CALIOP part) presented in the CLARA-A2 validation report. The reason is the use of CALIPSO version 4 datasets (version 3 was used in the CLARA-A2 validation report) and the introduction of the new concept evaluating the cloud detection sensitivity which is the core topic of this paper.

As regards the collocations with NOAA-18 and NOAA-19, these are exactly the same as for the standard CLARA-A2 validation (i.e., same number of collocations, about 5000 orbits). However, in this study we exclude collocations with the morning orbits of NOAA-17, Metop-A and Metop-B since these are only possible over a narrow latitude band close to 70 degrees. Thus, we want to focus on the global performance and that can best be studied based on afternoon orbit data.

The point about the necessity to evaluate the stability of a long-term data record is indeed an important aspect but also one of the most difficult ones to deal with. How can we find a suitable reference dataset of cloud observations with global coverage to perform this stability analysis? To be honest, there is no such reference dataset offering the required length and coverage of observations. The only candidate is surface (SYNOP) observations of cloudiness but they cannot fulfill the requirement of global coverage (e.g. oceanic and polar regions are largely not covered). They also have their own quality problems (e.g., lack of

knowledge of the thinnest cloud being observed, low quality at night-time and also hampered by being subjective in their character in that different observers have different opinions on how to interpret clouds and their coverage). Furthermore, the surface observation network has undergone rapid changes during the last decades due to automatization and this has caused problems in maintaining stable observation quality over time. With this background, we are of the opinion that there is no better reference than the 10-year CALIPSO dataset for evaluating the CLARA-A2 (and similar) satellite-derived data records, despite the fact that it only covers about one third of the CLARA-A2 observation period. It offers the global coverage (only excluding some areas in close proximity to the poles) and a high and stable quality of observations. Estimating the stability is still a challenge but we hope that on a longer term also this aspect will be properly dealt with assuming that the era of active cloud lidar observations from space can continue (e.g., with new data from EarthCARE and CATS replacing CALIPSO and hopefully also data from new lidar missions beyond the lifetime of EarthCARE).

In conclusion, we will add statements to the text based on the above reasoning justifying better why we think the results are still relevant for characterizing the entire CLARA-A2 data record. We can also propose a small change to the title as the following:

"Improved characterization of AVHRR global cloud detection performance based on CALIPSO-CALIOP cloud information: Demonstration of results based on the CM SAF CLARA-A2 climate data record"

**Repeating specific comment 2:**

**Objectives of the study should be described better in the Introduction. In relation to (1), it should be clear if the aim is to present new methodology using a subset of CLARA-A2 as an example or to evaluate CLARA-A2.**

**Reply:**

Yes, we will do that (with reference to the reply to 1).

**Repeating specific comment 3:**

**The current discussion section is a mix of discussion remarks and conclusions. I recommend to separate the two. In the results' section, there are also interpretations, which are hypothetical (they often start with "we believe", "we claim") and should be moved to the discussion. Otherwise it is**

**often difficult to judge which statements are really supported by the results achieved in this study.**

**Reply:**

Yes, we admit this weakness of the current manuscript. We will follow the recommendation of including both a Discussion section and a Conclusion section.

**Repeating specific comment 4:**

**The analysis of detection sensitivity reveals some interesting non-expected results. One is that CLARA performance is not better at dark and warm ocean surfaces (L374-375). The hypothesis this is due to sampling and geometry of AVHRR and CALIOP FOVs needs more explanation. The problem was detected here, because it leads to unexpected results. However, how to measure a possible effect of this issue on results in other situations, regions, etc.? I would consider a separate section (or paragraph) in the discussion..**

**Reply:**

Yes, we admit that this result deserves more attention. We also got a similar remark from the other reviewer. We suggest improving the description in three ways:

1. We will introduce a short summary of the underlying basic method of matching AVHRR and CALIPSO data. It seems the current referencing to the original paper by Karlsson and Johansson (2013) (which introduces the matching method) is not enough for a full understanding. We need to recapitulate the method's most important aspects also in this paper.

2. We will add a clear illustration (new figure) of how matched high-resolution AVHRR FOVs relate to the CALIPSO-CALIOP FOVs within a nominal AVHRR GAC pixel. This would help understanding the problem.

3. We will expand the discussion of these results in the new Discussion section. However, we believe that further studies on the full (global and local) impact of the differences of matched AVHRR and CALIOP FOVs could indeed deserve a paper on its own. Thus, we cannot dwell too much on this seemingly unexpected result since this would risk leading to a much too long paper. We only want to highlight the existence of this

problem which has (in our view) been largely overlooked in many previous papers using CALIPSO-CALIOP data as the main validation source.

**Repeating specific comment 5:**

**Is the cloud detection sensitivity a measure of CDR performance itself? There is no discussion if 0.225 signifies good or bad CLARA performance. One can imagine the same analysis (i.e. evaluation against screened CALIOP data), but with the estimated cloud detection sensitivity of, say, 0.5. Please elaborate on that. In addition, since the authors recommend the methodology to be widely used (e.g. in CFMIP), more detailed guidelines would be appreciated. For instance, when applied to different passive-sensor-based CDRs, should the cloud detection sensitivity be always recalculated?**

**Reply:**

Yes, even if it only concerns cloud detection performance, we believe that it is at least one very important piece of information for characterizing the entire CDR performance. Despite of the fact that it only deals with the cloud masking quality and not specifically with the quality of other parameters of CLARA-A2 (e.g. other cloud properties, surface albedo and surface radiation budget parameters), we also know that errors in cloud masking definitely will affect the quality of other parameters derived further down-stream in the processing of a data record. For example, incorrect cloud screening (missed clouds) over dark surfaces will inevitably lead to an overestimation of surface albedos. Exactly how the uncertainty in cloud masking is propagating into the uncertainty of other parameters is yet to be determined in more details than what is done today. However, to better describe this is one of the challenges in the CM SAF project when preparing the next version of the CLARA dataset (CLARA-A3). But for the current CLARA-A2 dataset (and which could also relevant for other similar type of datasets), this new description of the cloud detection performance can be seen as one important step towards a better uncertainty description.

The question whether the average cloud detection sensitivity at (cloud optical thickness) 0.225 represents a good or a bad performance has no clear answer. This is because this study is the first of its kind proposing such a measure defined in exactly this way (as described in the paper). However, one indication that it is probably not too bad is that the COSP (Cloud Feedback Model Intercomparison Project (CFMIP) Observation Simulator Package) satellite simulator for ISCCP uses a global cloud optical depth threshold of 0.3 to describe the cloud detection ability of the ISCCP dataset.

However, this quantity can only be evaluated when and if it is later put in relation to corresponding values (computed in the same way) for other datasets (like datasets from MODIS Collection 6, PATMOS-X, ISCCP or ESA-CLOUD-CCI). We encourage such studies since we think that this measure of performance is a universal one which has nothing to do with AVHRR data in particular. Instead, it should be applicable to any other global cloud dataset based on passive satellite imagery. And, yes, it should always be recalculated for every new dataset to be evaluated (answer to last question). These cloud detection sensitivities could then be inter-compared between different data records. This is the main point in promoting this method as a universal method.

The value 0.225 is only a global average calculated for CLARA-A2 (or to be strictly correct, for the 2006-2015 period of CLARA-A2) and it should only be inter-compared and evaluated with corresponding global averages derived for other cloud datasets. In that sense, the question about what happens if using the value 0.5 is not relevant. More interesting would rather be to compare the results of the global distribution of the cloud detection sensitivity (Figure 11) with corresponding distributions for other cloud datasets. This would be the most interesting aspect for use in a wider context since this would be able to reveal global differences (at a rather fine resolution) in performance for different algorithms and data records. Examples of such inter-comparisons are still rather few (with the GEWEX inter-comparison study by Stubenrauch et al. in BAMS July 2013 as the best example). A tentative repeated GEWEX inter-comparison study in the future could be imagined to include such global performance and difference maps valid for the entire period of CALIPSO data. That would really show how all these data records perform if using CALIPSO-CALIOP as representing the truth.

We will include some of these clarifications and proposals/suggestions in the new Discussion and Conclusion sections.

**FINAL REMARK:**

- We will clarify the unclear aspects listed among the short comments

- Finally, thanks for suggested editorial, syntax and language improvements. These are invaluable for non-native English writers like us!

---

## Referee Comment (RC3) · H. Deneke (Referee) · 22 Nov 2017

The manuscript provides an in-depth investigation of the cloud detection performance of the algorithm employed in the CLARA climate data record, utilizing CALIOP lidar observation as reference. The topic of the paper is interesting, presents novel results, and the approach is scientifically sound, hence I do recommend the paper for publication in AMT.

There are however a number of general comments/concerns which I'd like to see ad-

dressed/at least discussed in the manuscript before publication, which will further clarify the relevance of the results for readers. I also added a number of specific minor points/language corrections below, which is likely incomplete. I do recommend proofreading of the manuscript by a native English speaker.

General comments:

- Title: "Detailed characterisation" => from my point of view, the term "characterisation" mainly refers to a characterisation of performance in terms of CALIOP cloud optical thickness, I'd recommend adding COT to the title (e.g. "based on CALIPSO-CALIOP cloud optical thickness"), this is more specific than "cloud information" (what other information do you use?). I would also prefer the term "sensitivity" over "Performance", but that is definitely a matter of taste. Hence please consider modifying the title, taking these points into account.

- The authors should describe in more detail the cloud detection scheme and the changes between the CLARA-A1 and A2 data records, in particular with respect to cloud masking. The short paragraphs at the end of Section 2.1. seems somewhat too brief, considering that the aim of the paper is to characterize the performance of that scheme, and the findings might be different for other cloud screening methods. Has the cloud mask algorithm been changed/improved between the two versions of CLARA? Are changes in cloud detection performance expected, is it possible to quantify such changes using the validation approach? Do the calibration updates affect the cloud mask performance? Has the analysis of Karlsson et al.,2013, helped to improve the algorithm, i.e. have you been able to tune the algorithm based on the results of the previous validation study? Do you expect that your results are specific to this cloud masking method, or do you expect them to be linked to fundamental characteristics of the AVHRR observations you are using, so your findings would apply similarly to other AVHRR-based cloud detection algorithms? If the latter, how would this translate to other sensors as e.g. MODIS/SUOMI NPP/geostationary observations?

-In general, I find the approach of looking at the COT regardless of observing conditions somewhat too simple. I expect the detection performance to be very different during daylight/nighttime conditions, and also depend on cloud type/phase (viewing angle might be another important influencing factor). Additionally, the cloud detection scheme relies on a combination of tests, which will show different sensitivities to thin/thick/low/high clouds (it might be interesting to look at the sensitivity for each individual test separately). While it is nice to quantify the geographic variation of detection performance, what are the dominating factors for those variations (I guess surface albedo, cloud type?). Here, I urge the authors to discuss their results with more focus on the underlying physical effects (suggested plot: using a global surface albedo map e.g. from MODIS, show an x-y plot of threshold COT vs. surface albedo), and at least discuss if considering day/night different cloud types separately would add new insights.

-Due to GAC sampling, the comparability of CALIOP and AVHRR observations likely suffers. Can you quantify this effect using spatially complete data, e.g. by use of MODIS data to simulate GAC sub-sampling, in particular for those regions where clouds with significant small-scale variability are expected (i.e. the sub-tropical ocean). Even an analysis on limited data might shed some more insights in the context of the rather speculative disuccsion on page 10 ("We believe"...).

-In the conclusions, the author's stress that long-term availability of active observations from space would be benefical in the conclusions. While I generally support this point, due to the inherent value of active observations, I am not convinced that this indeed adds value to the aims of this paper. Do the authors expect the performance of the cloud mask to change over time? If so, what factors could change? Why is not a once-only characterization sufficient?

-Finally, I do think that the language/wording of the article can be significantly improved, both in terms of English language use and in terms of being stricter/more consistent in terminology (some examples: use of terms "parameters" vs. "scores", "performance"

vs. "sensitivity", "cloud screening" vs. "cloud detection" vs. "cloud masking", using the abstract term "detection sensitivity" instead of COT). Please do revise the paper once more carefully with respect to this points.

Detailed/language comments (disclaimer: I am not a native speaker myself...):

-L10 : "including their global distribution" => "regional variation"(?) (results is unspecific, so it remains unclear what a "distribution" of results actually refers to)

-L11 "sensitivity of the results" => which results? This opens up the possiblity for misunderstanding, please change "the results" to "the cloud detection performance" or name the statistical score you are referring to.

-L 11: "cloud optical thicknesses" => "thickness"

-L 21: "sensitivities ... were larger than 0.2" => please make it clear that COT is used as measure for sensitivity, and hence 0.2 is value of COT!

-L22 "over Sahara" => "over the Sahara"

-L23-L24: "The validation method', "validation results are proposed". This is fairly unspecific. Why not mention exlicitely "It is suggested to also quantify the detection performance of other CDRs in terms of a sensitivity threshold of cloud optical thickness which can be estimated using active lidar observations"

-L28: "appear increasingly important", do not use "appear", or do the author's doubt the value of their own work?

-L29: "cloud description and ... feedback processes" => suggested re-phrasing "the parametrization of cloud processes and cloud-aerosol interactions including related climate feedbacks."

-L37: I suggest to drop the part "in combination with ...", I do think satellite observations have sufficient value even without complementary ground-based observations

[Figure]

-L41: "the global view" => "their global coverage"

-L57: "Aqua train" => I have never heard this term, all references I can come up with translate A-Train to "Afternoon train"

-L162: "A very strict definition" => I do not think this is a definition, but a characterization (this point also applies to other similar uses later in the manuscript)

-L235: "behave in a strange way" => maybe "introduce distortions"

-L341/342: places=> regions/locations

-L442: performance parameters => be more consistent in terminology, do you mean skill scores, or the threshold in COT?

-L448: "The method . . . is not . . . valid for the CLARA-2 . . . method": from my reading, this statement seems to invalidate the whole paper, and does not make sense. Do the authors mean: "The method of using CALIOP data as reference is applicable"

-L449-450: "Because of this...": I do not understand the meaning of this sentence, please clarify it.

-L495: "A specific problem with the current method": its not an inherent problem of the method, but of data availability of active observations, I would thus suggest to use a different wording.

---

## Author Comment (AC3) · 19 Dec 2017

Final reply to Hartwig Deneke's review of the AMTD paper

**"Detailed characterisation of AVHRR global cloud detection performance of the CM SAF CLARA-A2 climate data record based on CALIPSO-CALIOP cloud information"**
**by**
**Karl-Göran Karlsson and Nina Håkansson, SMHI**

**Note: All line numbers referred to below are relevant for the revised manuscript version written in Word change track mode and named "CLARA_A2_validation_AMT_2017_version2_tracked_changes".**

**Repeating general comments:**

**The manuscript provides an in-depth investigation of the cloud detection performance of the algorithm employed in the CLARA climate data record, utilizing CALIOP lidar observation as reference. The topic of the paper is interesting, presents novel results, and the approach is scientifically sound, hence I do recommend the paper for publication in AMT.**
**There are however a number of general comments/concerns which I'd like to see addressed/at least discussed in the manuscript before publication, which will further clarify the relevance of the results for readers. I also added a number of specific minor points/language corrections below, which is likely incomplete. I do recommend proofreading of the manuscript by a native English speaker.**

**Reply:** Thanks for this positive evaluation. We will address all points in the following. The final manuscript has been checked by a native English speaker.

**General comment 1:**
**- Title: "Detailed characterisation" => from my point of view, the term "characterisation" mainly refers to a characterisation of performance in terms of CALIOP cloud optical thickness, I'd recommend adding COT to the title (e.g. "based on CALIPSO-CALIOP cloud optical thickness"), this is more specific than "cloud information" (what other information do you use?). I would also prefer the term "sensitivity" over "Performance",**
**but that is definitely a matter of taste. Hence please consider modifying the title, taking these points into account.**

**Reply:** We got similar remarks from other reviewers. We have changed the title as follows:

"Characterization of AVHRR global cloud detection sensitivity based on CALIPSO-CALIOP cloud optical thickness information: Demonstration of results based on the CM SAF CLARA-A2 climate data record"

**General comment 2:**

**- a) The authors should describe in more detail the cloud detection scheme and the changes between the CLARA-A1 and A2 data records, in particular with respect to cloud masking. The short paragraphs at the end of Section 2.1. seem somewhat too brief, considering that the aim of the paper is to characterize the performance of that scheme, and the findings might be different for other cloud screening methods. Has the cloud mask algorithm been changed/improved between the two versions of CLARA?**

**b) Are changes in cloud detection performance expected, is it possible to quantify such changes using the validation approach?**

**c) Do the calibration updates affect the cloud mask performance?**

**d) Has the analysis of Karlsson et al.,2013, helped to improve the algorithm, i.e. have you been able to tune the algorithm based on the results of the previous validation study?**

**e) Do you expect that your results are specific to this cloud masking method, or do you expect them to be linked to fundamental characteristics of the AVHRR observations you are using, so your findings would apply similarly to other AVHRR-based cloud detection algorithms? If the latter, how would this translate to other sensors as e.g. MODIS/SUOMI NPP/geostationary observations?**

**Reply:**

a) We disagree here in the sense that the CLARA-A2 paper by Karlsson et al., (2017) does exactly what is asked for here, i.e., it explains what has been done to algorithms (not only cloud retrievals) and calibration methods for the upgrade to the CLARA-A2 data record. We cannot repeat this here considering the length of the paper and the need to dwell deeper on other more serious subjects brought up by reviewers. However, we added a statement making it more clear where descriptions of algorithm changes can be found (lines 126-129).

**b)** Definitely. The paper by Karlsson et al. (2017) gives already some validation results (e.g. comparisons with MODIS Collection 6 results in Figure 6d in that paper). It also refers to the weaknesses of the CLARA-A1 cloud detection which largely have been solved by the new methods in CLARA-A2. However, the purpose of this paper is not to evaluate the improvement in the cloud detection algorithm from CLARA-A1 to CLARA-A2. Rather it introduces a method for a more detailed characterization of cloud detection sensitivity.

**c)** Yes. The cloud screening methods use fixed or pre-calculated thresholds which mean that if calibration drifts (i.e., visible reflectances changes) cloud detection results will also change. However, the used cloud detection scheme uses thresholds in the short-wave infrared and infrared regions with a higher priority than the visible thresholds. In that sense the sensitivity to visible thresholds is small (but not negligible).

**d)** Absolutely! It helped in finding the largest weaknesses of the cloud screening algorithm (e.g. the problems found over semi-arid regions) and the validation method has been heavily used to evaluate the impact of subsequent and final algorithm changes. We consider it as maybe the most important tool in the development work. But, of course, the CALIOP data itself (i.e., the access to almost one full decade of CALIOP data) is the most important aspect here.

**e)** Of course, these presented results are specific to the cloud screening method used for CLARA-A2. However, we believe that the evaluation method itself is universal and not specifically linked to AVHRR data or AVHRR-based methods. We state this very clearly in the Conclusions section on lines 720-724 and on lines 765-772. All satellite observations/retrievals which can be matched/collocated with CALIOP data can be evaluated in the same way. We think it is a strong point to suggest the use of one such universal method for determining the cloud detection sensitivity. It can facilitate how to inter-compare results from different methods and different satellite sensors.

Regarding the mentioned sensors (MODIS/SUOMI NPP/geostationary) we see no particular problem in trying to repeat the same kind of study. In fact, we are planning to do it ourselves in the near future, with the highest priority on evaluating measurements from the Suomi-NPP and NOAA-20 VIIRS sensors.

**General comment 3:**

-In general, I find the approach of looking at the COT regardless of observing conditions somewhat too simple. I expect the detection performance to be very different during daylight/nighttime conditions, and also depend on cloud type/phase (viewing angle might be another important influencing factor). Additionally, the cloud detection scheme relies on a combination of tests, which will show different sensitivities to thin/thick/low/high clouds (it might be interesting to look at the sensitivity for each individual test separately). While it is nice to quantify the geographic variation of detection performance, what are the dominating factors for those variations (I guess surface albedo, cloud type?). Here, I urge the authors to discuss their results with more focus on the underlying physical effects (suggested plot: using a global surface albedo map e.g. from MODIS, show an x-y plot of threshold COT vs. surface albedo), and at least discuss if considering day/night different cloud types separately would add new insights.

**Reply:** We definitely agree with the reviewer here regarding the potential for deeper and more detailed studies. But we have to stress (which is mentioned several times in the paper, e.g. on lines 637-640), that for doing this we need to have a more extensive dataset. Already with the present dataset we have identified problems in getting enough of samples to get statistically reliable results at the individual gridpoint level (here, we use 300 km resolution grid points). See for example the discussion about the results of Figure 13 in the revised manuscript (lines 615-618). The sparseness of data is mostly found at low latitudes which can be explained by the way samples are collected and the used polar orbits. To further sub-divide our dataset, e.g., into daytime and night-time portions, will probably lead to extended areas with lack of collocations.

Furthermore, we don't think it is really our job to explain why we have these validation results in terms of the cloud screening algorithm details. This is up to the development team of each investigated algorithm to discuss and understand. This study is mainly a validation study which may highlight algorithm weaknesses but it can neither explain the weaknesses nor provide solutions to overcome them.

In conclusion: More detailed studies may come later after receiving a longer time period of data and possibly if using less stringent matching criteria (i.e., allowing a temporal difference of 10 minutes instead of 3 minutes). But here, we prefer to stay with the current approach of making a first attempt to derive global results as a demonstration of the potential and only give a few examples of more local results (Figure 13).

**General comment 4:**

**-Due to GAC sampling, the comparability of CALIOP and AVHRR observations likely suffers. Can you quantify this effect using spatially complete data, e.g. by use of MODIS data to simulate GAC sub-sampling, in particular for those regions where clouds with significant small-scale variability are expected (i.e. the sub-tropical ocean). Even an analysis on limited data might shed some more insights in the context of the rather speculative disuccsion on page 10 ("We believe"...).**

**Reply:** We got similar questions from the other reviewers. We concluded that we need to improve our description and discussion of the matching methodology and better illustrate the geometrical aspects and consequences of matching the AVHRR GAC and CALIOP FOV observations. We have done that in three ways:

1. We introduced a short summary of the underlying basic method of how we matched AVHRR and CALIPSO data (first part of Section 3.2). It seems the current referencing to the original paper by Karlsson and Johansson (2013) (which describes the matching method) is not enough for a full understanding. We need to recapitulate the method's most important aspects also in this paper.

2. We added an illustration (new Figure 1) of how matched high-resolution AVHRR FOVs relate to the CALIPSO-CALIOP FOVs within a nominal AVHRR GAC pixel. The consequences for the matching of the two datasets are described in the second part of Section 3.2.

3. We expanded the discussion of these results in the new Discussion section (Section 5, lines 642-695). Thus, the current Discussion section will be split into one separate Discussion section (Section 5) and one final Conclusion section (Section 6). The problem of inter-comparing CALIOP data with other satellite data in cases of highly scattered and fractioned cloudiness needs to be discussed. In our opinion this aspect has been largely overlooked in many previous papers using CALIPSO-CALIOP data as the main validation source.

**General comment 5:**

*-In the conclusions, the author's stress that long-term availability of active observations from space would be benefical in the conclusions. While I generally support this point, due to the inherent value of active observations, I am not convinced that this indeed adds value to the aims of this paper. Do the authors expect the performance of the cloud mask to change over time? If so, what factors could change? Why is not a once-only characterization sufficient?*

**Reply:** Yes, in principle a once-only characterization is probably OK for an individual data record like CLARA-A2. But for its evolution over time (i.e., upcoming new versions of CLARA, like the currently planned CLARA-A3 to be released in 2021-2022) there is a need for new evaluations. Especially, future versions of CLARA will have to be transformed into an AVHRR-heritage type of data record since the AVHRR instrument itself will soon be missing on upcoming satellites. The last AVHRR will be launched on METOP-C (scheduled for 2019) which effectively means that no AVHRR measurements can be expected beyond the 2025-2030 time frames. However, AVHRR-heritage datasets are still possible if utilizing AVHRR-like spectral channels on other sensors, e.g. the VIIRS sensor of the JPSS satellites. But to evaluate and get a smooth transition of the data record in this way we need to repeat studies like this with the existing data from active (lidar) measurements. We have added a comment on this (lines 806-809).

However, there is also a very important aspect in that we currently lack good reference data to estimate the stability of data records (mentioned on lines 804-806). An extension of missions with active lidar instruments in space will eventually allow more accurate estimations of the data records stability over time.

**General comment 5:**

*-Finally, I do think that the language/wording of the article can be significantly improved, both in terms of English language use and in terms of being stricter/more consistent in terminology (some examples: use of terms "parameters" vs. "scores", "performance" vs. "sensitivity", "cloud screening" vs. "cloud detection" vs. "cloud masking", using the abstract term "detection sensitivity" instead of COT). Please do revise the paper once more carefully with respect to this points.*

**Reply:** Certainly, we are aware of language limitations and mistakes in the manuscript. We have taken these aspects into account and also in the end we

used native English speaking people for a final check of the manuscript. We are grateful for all language comments and suggestions in the following.

**Detailed/language comments (disclaimer: I am not a native speaker myself...):**

**-L10 : "including their global distribution" => "regional variation"(?) (results is unspecific,so it remains unclear what a "distribution" of results actually refers to)** Rephrased (lines 13-15)

**-L11 "sensitivity of the results" => which results? This opens up the possiblity for misunderstanding, please change "the results" to "the cloud detection performance" or name the statistical score you are referring to.** Rephrased (lines 16-17)

**-L 11: "cloud optical thicknesses" => "thickness"** Corrected (line 19)

**-L 21: "sensitivities : : : were larger than 0.2" => please make it clear that COT is used as measure for sensitivity, and hence 0.2 is value of COT!** The quantity "cloud detection sensitivity" is clearly defined in the text (lines 16-17) as a COT value. No change.

**-L22 "over Sahara" => "over the Sahara"** Corrected.

**-L23-L24: "The validation method', "validation results are proposed". This is fairly unspecific. Why not mention exlicitely "It is suggested to also quantify the detection performance of other CDRs in terms of a sensitivity threshold of cloud optical thickness which can be estimated using active lidar observations"** Adopted.

**-L28: "appear increasingly important", do not use "appear", or do the author's doubt the value of their own work?** "appear" is replaced with "are".

**-L29: "cloud description and : : : feedback processes" => suggested re-phrasing "the parametrization of cloud processes and cloud-aerosol interactions including related climate feedbacks."** Adopted.

**-L37: I suggest to drop the part "in combination with ...", I do think satellite observations have sufficient value even without complementary ground-based observations** Adopted.

**-L41: "the global view" => "their global coverage"** Corrected.

**-L57: "Aqua train" => I have never heard this term, all references I can come up with translate A-Train to "Afternoon train"**
Corrected (lines 67-68).

**-L162: "A very strict definition" => I do not think this is a definition, but a characterization (this point also applies to other similar uses later in the manuscript)** Rephrased (lines 197-198).

**-L235: "behave in a strange way" => maybe "introduce distortions"** Adopted (lines 332).

**-L341/342: places=> regions/locations** Corrected.

**-L442: performance parameters => be more consistent in terminology, do you mean skill scores, or the threshold in COT?** Rephrased (line 714).

**-L448: "The method : : : is not : : : valid for the CLARA-2 : : : method": from my reading, this statement seems to invalidate the whole paper, and does not make sense. Do the authors mean: "The method of using CALIOP data as reference is applicable"** Adding the word "exclusively" after "valid" (line 720-721) clarifies that we (of course) don't want to invalidate the whole paper.

**-L449-450: "Because of this...": I do not understand the meaning of this sentence, please clarify it.**
Reformulated (lines 723-724) and adding reference to Stubenrauch et al., 2013.

**-L495: "A specific problem with the current method": its not an inherent problem of the method, but of data availability of active observations, I would thus suggest to use a different wording.**
Rephrased (lines 697-710).

---

## Author Comment (AC4) · 19 Dec 2017

Final reply to Referee 1's review of the AMTD paper

**" Detailed characterisation of AVHRR global cloud detection performance of the CM SAF CLARA-A2 climate data record based on CALIPSO-CALIOP cloud information"**
by
**Karl-Göran Karlsson and Nina Håkansson, SMHI**

**Note: All line numbers referred to below are relevant for the revised manuscript version written in Word change track mode and named "CLARA_A2_validation_AMT_2017_version2_tracked_changes".**

**Repeating general comment, part 1:**

**This manuscript evaluates the cloud mask of the CLARA-A2 climate data record (based on passive imagery from AVHRR polar orbiters) with collocated active cloud detections (CALIOP). Another, more general, paper has been published in ACP this year, and this AMT paper focuses exclusively on the cloud mask. This approach is sufficiently well justified, but the paper under review relies too much on the earlier publication (Karlsson et al., 2017; also to some extent on Karlsson et al., 2013) to explain the background. In order to qualify for publication in AMT, revisions need to be made to ensure that it can stand on its on, while not replicating too many of the science results.**

**Reply:**

The two referred papers (especially Karlsson et al., 2013) are important papers in that they set the stage and define the framework for how to perform the matchups between AVHRR and CALIOP data. We get the feeling from some of the comments that we need to clarify the framework even further (i.e., that it is not enough to just provide the references). Thus, we have included a short summary of the most important points (new section 3.2) concerning the basic matching or collocation methodology (see also reply to **general comment, part 4** further below).

Regarding the justification of this paper and the question whether it adds anything new compared to the paper by Karlsson et al. (2017), we claim that one important objective of this study was to investigate the impact of upgrading the

results by using the new CALIPSO-CALIOP version 4 dataset (which indeed is the main topic of the Special AMT Issue too which the paper was submitted). The previously mentioned validation efforts were all based on CALIPSO-CALIOP version 3 data. We have now added this objective in the Introduction section (lines 82-83).

However, another more important objective was to show that the CALIPSO-CALIOP dataset can be used to investigate much more in detail the cloud detection limitations of one particular cloud screening method (like the one used for CLARA-A2) than what has been presented before. The concept of Cloud Detection Sensitivity (as illustrated by results in the original Figure 11, now Figure 12 in the revised manuscript) is a new approach which we hope can become a standard tool for a more objective evaluation of cloud climate data records in the future. Its main advantage is that it can be considered as a universal method, not depending specifically on the actually studied AVHRR dataset. It is a method based on a special organization of the CALIOP cloud dataset by use of estimated cloud optical thickness sub-categories. These results, being organized in cloud optical thickness sub-categories, can be compared to any other collocated satellite-based dataset.

We have emphasized better the objectives of the study in the Introduction section (lines 76-91) and highlight better the results and the potential of the derived Cloud Detection Sensitivities in current and future studies in the Conclusions section (lines 720-724, 765-772 and 787-799).

**Repeating general comment, part 2:**

**In its current state, the paper is hard to review because some of the concepts are not explained sufficient well (specific examples are given below), and because details are left out. In addition, the manuscript is unnecessarily wordy in some places and has basic deficiencies with English/Grammar (for example, "were" is used instead of "where" throughout the manuscript; there are many run-on sentences; punctuation is used too sparingly; use of slang words such as "punish" for a statistical approach that are frequently used by the community, but should be used only where absolutely necessary). Before going into the copy/edit process at AMT, a native speaker should be consulted to ensure logical flow and readability of the manuscript overall.**

**Reply:**

As being non-native English authors, we admit limitations in the ability to produce perfect quality (scientific) English text. We thank the reviewer for pointing out the most common errors and we have done our best in eliminating

them. We have also consulted a native speaker before submitting the revised version of the manuscript.

**Repeating general comment, part 3:**

**Despite the criticism of the presentation quality, the content is interesting in that the cloud detection capability is studied as a function of optical thickness and region. Obviously, the POD (probability of cloud detection) depends on surface albedo and emissivity, mechanisms that are identified by the authors. Two comments here:**

**1) It should be stated more clearly where such findings have been made previously. The author make a point that the regional assessment is new, but there have been previous studies that focused on some of the problematic regions specifically in the Arctic with CALIOP that are not cited here (for example, studies by Gettelman, Kay, L'Ecuyer and a few others).**

**Reply:**

The knowledge of the dependency on surface characteristics (e.g. albedo or emissivity) for the possibility to separate clouds from Earth surfaces in satellite imagery is nothing fundamentally new. Rather, it is a well-established and well-known fact in the satellite user community. The reason is obvious: All cloud screening methods depend on the ability to find enough of contrast between clouds and underlying surfaces in the investigated images. This is valid for all spectral regions - be it visible, near-infrared, short-wave infrared or infrared. Multi-spectral methods will have the best capability since the use of many spectral channels increases the probability that at least one spectral channel will offer enough of contrast between clouds and Earth surfaces. This explains e.g. the high quality of cloud datasets from MODIS (with access to up to 36 useful spectral channels).

The challenges here are naturally largest at high latitudes and near the poles where we have both bright Earth surfaces (snow, ice) and very cold surface temperatures (very similar or even colder than clouds which normally are warmer than clouds in other regions). This explains the special interest here (as exemplified by the mentioned papers).

We have added some of these references (lines 760-763) since we agree that they absolutely need to be mentioned in this context. However, the most important thing is that we even stronger have emphasized that the proposed method offers a universal method (which could become a standard method) to

monitor these problems globally and not just in specific regions (see reply above to **general comment, part 1**). This is the big advantage of the method. Thus, our statement about the novelty of the regional assessment should be interpreted as that the method offers both a monitoring of mean global conditions but also a regional monitoring including all regions on Earth and not just some selected ones.

**Repeating general comment, part 4:**

**2) It remains unclear (partially because of the structural problems of the manuscript pointed out above) why there are some regions where cloud cover is overestimated by the passive imagers. One possible explanation is not sufficiently investigated: sub-grid resolution clouds that could be picked up by passive imagers but not by active imagers (if they are outside the FOV). There is some discussion of it, but it remains superficial.**
**Also, active observations are portrayed as the ultimate "judge" for the performance of the cloud mask derived from passive observations, and they shouldn't be. As pointed out by the authors, active observations have their own limitations (sensitivity, FOV, day-vs-night contrasts). The truth is that active cloud observations afford a different perspective on clouds that happens to be less sensitive to the surface reflectivity and  emissivity than that of passive observations. This distinction (and the limitations of both approaches) should be made clear by the authors.**

**Reply:**

We agree that we could have been clearer in the discussion of aspects that are related to the different FOVs of AVHRR and CALIOP. Some discussion is included on page 10 (lines 366-404) and on page 12 (lines 465-471) but this can be improved. Since other reviewers also have pointed out more or less the same thing we have done the following:

1. We introduced a short summary of the underlying basic method of how we matched AVHRR and CALIPSO data (Section 3.2). It seems the current referencing to the original paper by Karlsson and Johansson (2013) (which describes the matching method) is not enough for a full understanding. We need to recapitulate the method's most important aspects also in this paper.

2. We added a clear illustration (new Figure 1 in Section 3.2) of how matched high-resolution AVHRR FOVs relate to the CALIPSO-CALIOP FOVs within a nominal AVHRR GAC pixel. This would help understanding the problem.

3. We expanded the discussion of these results in a new Discussion section (lines 642-695). Thus, the previous Discussion section is now split into one separate Discussion section and one final Conclusion section. The problem of inter-comparing CALIOP data with other satellite data in cases of highly scattered and fractioned cloudiness is now discussed in more depth in the new Discussion section. In our opinion this aspect has been largely overlooked in many previous papers using CALIPSO-CALIOP data as the main validation source.

The question on why there seems to be regions where cloudiness is overestimated is interesting but the reasons behind this is at least partly beyond the scope of this paper. We do express some qualified guesses about the reasons for some of the found deviations in the study but, basically, this is really up to the algorithm originators to further analyze and explain in subsequent studies. However, we think that it cannot really be related solely to "sub-grid resolution clouds that could be picked up by passive imagers but not by active imagers (if they are outside the FOV)". This mismatch can definitely occur for individual AVHRR GAC pixels and for individual orbits but when summed up in a climatology based on thousands of orbits such biases will end up to be either very low or non-existing. Simply since the opposite case (clouds picked up by active sensors and not by passive sensors) is just as likely to occur. We have explained that in relation to the illustration envisaged in point 2 above (Figure 1, Section 3.2 and lines 642-695 in the Discussion section). But what is important is that the precision (variance) of the estimated mean cloud cover will degrade (i.e., higher RMS errors) when this occurs and this is emphasized in our discussion.

Another important thing is that the indicated overestimation may actually be caused by an inappropriate value of the global mean cloud detection sensitivity (i.e., minimum cloud optical thickness) for regions were cloud detection is very efficient. This is discussed in lines 627-640 in the Discussion section.

Regarding the choice of CALIPSO-CALIOP data as the "ultimate" judge, we both agree and disagree with the Reviewer's opinions. Admittedly, active data has its limitation where the FOV representability in relation to the AVHRR GAC FOV is perhaps one of the largest (as discussed above). However, for clouds with scales larger than the AVHRR GAC FOV (5 km) we still claim that no other observation reference can provide a better estimation of global cloud presence and distribution than the CALIPSO-CALIOP dataset. The big advantage with the CALIOP information is that we measure the lidar reflection from real cloud particles (in the CALIPSO-CALIOP version 4 dataset also quite

confidently separated from aerosol particles) and from the backscatter energy we can also for the thinnest clouds estimate with high accuracy the optical thickness of the cloud layers (up to a certain maximum value). No other sensor can provide the same. MODIS data is an alternative but in our opinion the MODIS dataset share many of the problems experienced by dataset produces from most multispectral passive sensors (AVHRR, SEVIRI, VIIRS, ABI, etc.) and this is basically explained by the fact that the measurement always contain a mix of contribution from clouds, the atmosphere and the surface (especially in the cases of thin clouds). We cannot be sure that we only measure the impact of the cloud itself. For an active sensor we do not have the same problem. However, most important in this context is that for a study like this it is very important to have access to very accurate estimations of cloud optical depth for the very thinnest clouds in order to carry out a sensitivity study like this. Here the CALIPSO-CALIOP measurement is quite superior to MODIS. For the latter sensor, estimations of cloud optical thickness of the thinnest clouds have high uncertainties due to the strong dependency on radiance contributions from the underlying surface and atmosphere. This is the main reason for using CALIOP data instead of MODIS data. We have explained the importance of having access to accurate estimations of cloud optical thicknesses (lines 87-89, 146-154, 195-200 and the entire section 3.5) in order to carry out our study. With this information and background we think there is no need to discuss why we have chosen CALIOP instead of MODIS in this paper.

**In the following we will address selected short comments (which are not simply editorial):**

**p2,L60: Why is CALIOP singled out as important for cloud observations, where in fact MODIS is flown in the A-Train as well. Wouldn't the MODIS observational record, in conjunction with CALIOP, lend itself to a similar study as the one presented here? Of course, its data record is much shorted, but on the other hand, MODIS and CALIOP are collocated all the time, by design.**

**Reply:**

We just gave some arguments in the reply above to **general comment, part 4**. It is our opinion that CALIOP data is a better reference in the sense that the measurement information is free from surface (and atmospheric water vapour and aerosol) dependence. However, even more important is that we cannot use MODIS data for the cloud detection sensitivity study since the cloud detection sensitivity of MODIS is probably not very different from AVHRR. More clearly, we repeat that we need access to very accurate cloud optical thickness estimations for very thin clouds in order to make such a study. The uncertainty

of the MODIS-derived optical thickness in this optical thickness interval (values less than 1.0) is too high and at least much higher than for CALIOP-derived optical thickness. We have pointed out the requirement of very accurate optical thickness information for very thin clouds (see reply to previous comment) which we think is enough for justifying the choice of CALIOP as our reference.

It would actually be very interesting to do a similar study of the MODIS C6 cloud detection sensitivity with the same method as presented here. We would expect some improvements compared to CLARA-A2. Figure 6d in the CLARA-A2 paper in ACP indicates an almost constant bias in cloud cover of about 5 % for MODIS C6 over all latitudes (more clouds observed by MODIS). The question is then if the global distribution of the cloud detection sensitivity (=minimum detected cloud optical thickness) will decrease with a constant value everywhere or are there possibly regional differences?

**p2,L60: The limitations of CALIOP (e.g., day time vs. night time detection, noise etc., strategies for thin cloud detection) should be discussed here.**

**Reply:**

We have been using CALIPSO-CALIOP cloud information in cloud validation activities ever since 2007 (shortly after data was made available). For our applications, we have not encountered or noticed any specific problems regarding the efficiency in CALIOP cloud detection between day and night. CALIOP daytime results are a bit more noisy due to some reflected sunlight contaminating the signal but we believe that the enhanced noise is mostly relevant and serious for studies of very weak signals, e.g. from very thin aerosols. In the CALIOP version 4 dataset, a better Cloud and Aerosol discrimination method was introduced and the previous problems in misclassifying heavy aerosols as clouds over specific regions of the world has been taken care of (see lines 170-174). Consequently, we see no reason to add any deeper discussion on the quality of the CALIOP measurements. We think that the representativeness issues (i.e., that AVHRR and CALIOP probes different parts of the GAC FOV) as discussed in Section 2 and extensively discussed in the Discussion section is actually more serious than actual uncertainties of the CALIOP measurement.

**p2,L70: The earlier study by Karlsson is cited here. It should be summarized in at least one paragraph since this paper needs to stand on its own. What was the scope of that manuscript? The extension by CALIOP, on the other hand, are well explained (with the caveats pointed out above).**

**Reply:**

Done, see point 1 above in the reply to **general comment, part 4**.

**p3,L79-87: This paragraph should be completely rewritten. The explanation of the field of view of the passive vs. the active instrument is vital for understanding this manuscript, yet it is incomplete. What is the GAC FOV vs. the FOV(passive) vs. the FOV(active, at native vs. aggregated resolution)? What data specifically are dropped?**
**The best way to explain this would be through a simple illustration of the AVHRR pixels vs. the CALIOP FOV of single shots, as well as the aggregation of individual pixels/ shots in the various products used in this study. Without this added figure, it will be hard to retrace the steps that were taken in this manuscript.**

**Reply:**

Done, see point 2 above in the reply to **general comment, part 4**.

**p3,L90: Which parameter retrievals? How is the radiance inter-calibration and data record homogenization done? Simply referencing Heidinger will not do because the specifics are missing. One of the clear requirements of AMT publications is that anybody reading the paper needs to be able to retrace the steps of a study from the original data to the findings. There is not sufficient detail provided here (or in other parts of the manuscript) to do that.**

**Reply:**

The parameters we mention concern the different variables included in the entire CLARA-A2 dataset and we have clarified this (lines 113-123). Apart from cloud amount there are 7 different cloud properties, a surface albedo estimation and an estimation of surface radiation budget parameters.
It is true that there is still no follow-on paper to Heidinger et al. (2010) describing the upgraded calibration equations. But there is a recent publication in the GSICS Newsletter describing the associated PyGAC preprocessing software
([ftp://ftp.library.noaa.gov/noaa_documents.lib/NESDIS/GSICS_quarterly/v11_n](ftp://ftp.library.noaa.gov/noaa_documents.lib/NESDIS/GSICS_quarterly/v11_n) [o2_2017.pdf](o2_2017.pdf)) . PyGAC contains the final calibration which was used for the

CLARA-A2 processing and is available as an open source package. We have added this reference to the manuscript (line 113 and lines 837-840).

**p4,l118-127: See comment above. These sections cannot be understood without better explanations of the FOVs, data aggregation and homogenization.**

**Reply:**

See the reply to **general comment, part 4**.

**p4,l129-134: Provide description of specific NOAA orbits that were included (vs. those that were not). Also, why were MODIS observations NOT used? The minimum information for the NOAA observations are: (a) instrument/satellite names and short description; (b) orbit inclination and equator crossing time; (c) life time of satellite; (d) orbital shifts over time**

**Reply:**

See the reply to **general comment, part 4**. We have tried to cover all those aspects. The MODIS question has already been dealt with in the reply to the comment for **p2,L60.**

**p4,l148: The theoretical deliberations on cloud mask/cover are insufficiently backed by literature. The paper that comes to mind when talking about the meaning of a "small" or "thin" cloud is that by Koren ("How small is a small cloud"). A short literature study on the topic would be advisable, given that it is the main topic of this article.**

**Reply:**

Thanks for this advice. We have taken a closer look and added some adequate literature references (lines 177-185). What is clear, though, is that there are several aspects of this topic. The paper by Koren et al. (2008) discusses primarily the impact of varying sizes of small Cumulus clouds in fine resolution satellite imagery (e.g. Landsat) and this is perhaps not directly applicable to AVHRR data in the comparably much coarser GAC resolution. For GAC data, we are perhaps more interested in when large scale (in contrast to small cumulus) clouds become so optically and geometrically thin that they are not detectable any longer. This is the probably the most important aspect for GAC

data. Nevertheless, also when cloud elements begin to approach a scale that is much finer than the GAC resolution (analogous to the cumulus case described by Koren et al. (2008)) we will also lose detectability. This is a very important aspect when trying to understand the implications of the matching geometry depicted in the new Figure 1. Consequently, we have expanded our discussion here on those aspects (Section 3.2 and lines 642-695 in Section 5).

**p5,l184: "possibly punish AVHRR-based methods in an unfortunate and undeserved way: : :": three words (punish, undeserved, unfortunate) are inappropriate for a scientific publications. There are multiple occurrences of such "personalized" or "humanized" comments, which should all be translated into objective, rather than "punitive" language.**

**Reply:**

We have removed the use of non-scientific terminology.

**p5, l187: The optical thickness threshold of 5 for CALIPSO is higher than usually assumed. If it is necessary for this study to work with such a high threshold, it should be justified, and it should be explained how this is possible (referring to literature where this has been done, or with a dedicated sub-section in this manuscript where it is shown that the lidar does, in fact, allow to go to COD 5, and under which circumstances).**

**Reply:**

We admit that we do not have good support in the literature for stretching the useful upper limit of CALIPSO-derived COD to 5. However, in the description of the upgrade to CALIPSO-CALIOP version 4 it is also emphasized that previous cloud optical thicknesses in version 3 were generally underestimated. This is also clearly indicated in Figure 2 (new Figure 3) in the manuscript. Whether this increase entirely justifies moving the upper limit to 5 is still not clear.

We do have more indications from our own investigations that an adjustment of the upper limit seems possible. In a study related to a paper by Riihelä et al (2017) we investigated the correlation between CALIPSO-estimated and CLARA-A2 estimated CODs over various surfaces (with snow surfaces over Greenland as the main target). However, when isolating the collocated results over ice free ocean surfaces at high latitudes (noting that over a dark surface also the AVHRR-based estimations should be more accurate), we could clearly see a

good correlation between the two estimations up to about COD=5 (see figure below):

[Figure]

Although this is not a perfect illustration (not included in Riihelä et al, 2017, but maybe considered for a follow-up paper) it shows how CLARA-A2-estimated optical depths compare to CALIOP-estimated optical depths in the range 0-15. Over a dark ocean surface the majority of values agree pretty well but what is clear is that an increasing number of cases (for higher optical depths) CLARA-A2 values saturates at 100 for CALIOP-values exceeding approximately 4 (noticeable at top of the figure). This reflects the inability of CALIOP to provide reasonable optical thicknesses for optically thick clouds. But, we made the conclusion that values compare pretty well even up to an optical thickness of 4-5 and this was one of the reasons why we decided to use the CALIOP interval 0-5 for this particular study (for AMT).

It is this finding that made us to use the maximum limit of 5 in this particular study. Unfortunately, in the end, we did not include this part of the inter-comparison in the finally published paper by Riihelä et al. (2017).

We propose that we keep the original maximum value of 5 in our plots but add a remark that values near this upper end are uncertain (lines 146-154). The upper

limit is not crucial for the findings of our study since in most cases the cloud detection sensitivity is considerably lower than 5. Only for some positions over Greenland and Antarctica we approach these high values but whether the value is 3 or 5 here does not really matter since it deviates anyhow very much from the values found on other places (which is the main message).

The mentioned reference is the following:

Riihelä, A., Key, J. R., Meirink, J. F., Munneke, P. K., Palo, T., & Karlsson, K.-G. (2017). An intercomparison and validation of satellite-based surface radiative energy flux estimates over the Arctic. Journal of Geophysical Research - Atmospheres, 122(9), 4829–4848. https://doi.org/10.1002/2016JD026443

**p6: There are multiple gaps on this page: The notion of "scores" (and different kinds) are used without sufficient (or any) explanation in this section, or in section 3.3. Too many questions remain, for example, which parameter of what satellite is validated with which other parameter, and how exactly as "score" (of any kind) is established.**
**How is the aggregation done? Why are scores only plotted as a function of COD up to 1, where in fact CODs up to 5 are advertised? What is the "improvement"? If the figures are insufficiently explained, it is not possible to understand. What has been "transformed from cloudy to clear cases" (l212), and how is that done? What is the role of Kuiper vs. hit rate (should be spelled "hit rate", not "hitrate"). Each of the bulleted items of the list on p6/p7 need to be explained and supported with formulae where appropriate. Here again, terms such as "punishing" should be avoided if at all possible.**
**After this paragraph, the reviewer was unable to give this a thorough review because the basics for understanding the remainder of the manuscript were not established.**
**The reviewer is willing to review another version of the manuscript where this has been fixed.**

**Reply:**

We have improved the description here to improve the understanding of the method and the results. We had these questions in mind when dealing with point 1 in the reply to **general comment, part 4**. However, regarding the exact definition of the used validation scores (bullets on pages 6-7) we insist on that the reference to the paper by Karlsson and Johansson should be enough (although we have also added some clarifying comments on lines 338-346). This previous paper defines all these scores with illustrations and formulas. The current revised manuscript is already extended substantially as a consequence of

all the requests from reviewers and we think that further extensions shall be avoided where it is possible. Furthermore, all scores except the Kuipers score are standard scores provided with short text descriptions. Regarding the Kuipers score we have added a comment on how values should be interpreted (lines 345-346).

We are certainly grateful for the reviewer's willingness to check the revised manuscript.

**p7,l265: This question is a great one, and at the center of this manuscript. However, the method description below is insufficient. Terms from machine learning ("overtrained") are evoked without explanation how they relate to the manuscript content. Also, here again, CALIOP is represented as the "objective" instrument that AVHRR is validated by where possible – where in fact the two instrument just assess different aspects of a cloud (see comment above).**

**Reply:**

Yes, this is the core topic of the paper. In our opinion, the method of determining the Cloud Detection Sensitivity is a way of utilizing the sensitivity difference between the two sensors in the most optimal way. Regarding the use of terms "overtrained" and "overfitted" we actually insist on keeping them here. Bullet number 2 is important and expresses a general problem of cloud screening methods (not particularly the CLARA-A2 method) and how to train them (especially statistical regression methods and artificial neural networks). Again, we repeat that we think that the described method can be applied to investigate any cloud screening method and not just the one used in CLARA-A2. For that reason, this bullet is important.

We will improve the description here (the mentioned aspects have already been commented in replies to similar comments above).

**p7, l278-l304: This seems to wordy and hard to follow since some of the concepts were not introduced.**

**p8, l306: Now some of the orbits are introduced, but that is too late in the manuscript. In addition to NOAA-18 and NOAA-19, did other data go into the CDR under investigation?**

**Reply:**

To be dealt with as indicated in the reply to **general comment, part 4**. The used NOAA-18 and NOAA-19 data (being matched with CALIPSO) is exactly the same dataset as was used for the evaluation in the CLARA-A2 paper by

Karlsson et al, 2017). However, in that study also results for morning satellite data (NOAA-17, METOP-A, METOP-B) were presented, although only valid over a small latitude band around 70 degree latitude. Since this study focus on global conditions we excluded the morning satellite part of the dataset. Nevertheless, we keep some discussion on morning satellites since they are important for CLARA-A2 (almost 40 % of the CLARA-A2 data consists of morning satellite data). The matching with morning satellites will introduce a new type of matching problems. This will be discussed in the revised manuscript in connection with the discussion of the figure to be introduced in point 2 (new Figure 1 in Section 3.2) of the reply to **general comment, part 4**. We have also kept a discussion on how to deal with the validation of morning satellite data in the new Discussion section (lines 697-710).

**p9,L350: insufficient introduction how systematic and random errors were establish make it hard to understand Figure 7.**

**Reply:**

We think that the understanding and statistical definition of systematic and random errors should be well known in the scientific community. Systematic errors concern the error of the mean value (normally described as "bias") while random errors define the typical variations (variability or variance) around a particular mean value. We insist on keeping the current formulations but we have introduced the terms "systematic" and "random" already in the descriptions of the used validation scores in Section 3.4 (lines 338-339).

**p9,l355-359: Add explanation why AVHRR gives higher cloud cover. It is easy to imagine a scenario where small cumulus clouds would be picked up by AVHRR (even if below its spatial resolution), but not by CALIOP products (for physical reasons). The statistical explanation given here does not seem to be complete and is hard to follow.**

**Reply:**

We claim that the opposite situation (i.e., clouds picked up by CALIOP but not by AVHRR) is also very likely (see reply to reply to **general comment, part 4**.). Thus, it is not obvious that one can use this explanation to explain higher AVHRR cloud cover.

Again, it cannot be the objective or task of this paper to explain why we see the deviations we but rather to provide an as sensible and trustworthy validation result as possible. The reasons for the deviations have to be explained by those

being responsible for the actual algorithms. We do give some suggestions but in the end this has to be verified by the algorithm developers.

However, the results in Figure 7 (new Figure 8) illustrate a more general problem. One has to remember that all results in Figures 6-10 (new Figures 7-11) are based on comparison with a CALIOP cloud mask filtered at cloud optical thickness 0.225 (the latter being the global mean cloud detection sensitivity). But regionally, the value of the cloud detection sensitivity varies a lot (see Figure 11 or new Figure 12). We have also changed the colour scheme in Figure 11 (new Figure 12) so that the relation to the mean value of 0.225 is made clearer:

[Figure]

In the new Figure 12 all values below the mean value 0.225 are plotted in blue colours and values above 0.225 in red colours. This colour representation also indicates better the highest values in the polar areas (not properly visualized in the old Figure 11).

Notice here that most oceanic areas are coloured blue and that the positive bias in Figure 7 (new Figure 8) is also mostly occurring over oceanic areas. By using a CALIOP cloud mask with cloud optical thickness being cut at 0.225 as our validation reference, we are then ignoring a substantial fraction of originally detected clouds below this cloud optical thickness limit over ocean surfaces. But these clouds are to a large extent actually detected in the CLARA-A2 results. Thus, it leads to an apparent (but largely false) overestimation of cloudiness over ocean in Figure 7 (new Figure 8 - notice that we are filtering CALIOP data and not CLARA-A2 data). We have added a discussion of this aspect in the Discussion section (lines 627-640).

This illustrates how difficult the estimation of general validation scores really is. More clearly, regardless of using a filtered CALIOP cloud mask or not when

validating, there are always disadvantages. In that sense, the results expressed by the globally resolved cloud detection sensitivity is a much more objective visualization of the cloud detection performance provided that also a separate evaluation of false alarm rates are made. We repeat the following statement from section 3.4 (lines 295-300): "An important additional or complementary parameter in this context would be the false alarm rate in the unfiltered case (FARcloudy(tau=0)) since this parameter is not depending on any filtering of thin clouds". We have added this as an important result and recommendation in the Conclusions section (lines 408-414).

Finally, the most correct way of calculating and plotting the Bias in Figure 7 (new Figure 8) would have been to actually use the derived grid-resolved Cloud Detection Sensitivities in Figure 11 (new Figure 12) as representing the most appropriate CALIOP cloud mask (i.e., the filtered cloud optical thickness) for validation. We had this option in mind but we realized that this probably needs a much larger sample dataset in order to calculate stable statistics (since it requires calculation based on only those samples existing for every single grid point). Figure 12 (new Figure 13) illustrates that the available number of samples in each grid point is still rather small which makes the estimation of statistical parameters on this scale rather uncertain. But it can be considered for the future if the time series of CALIPSO collocations can be extended with several more years. The existing undersampling at grid point level (especially at low latitudes) is commented on lines 615-618 and on lines 637-640.

**p10,369-371: What is Kuiper's score, what's the dominating mode in which case? At this point, some examples that help understanding one score vs. another are provided which is helpful, but that should be done (more systematically) earlier in the manuscript.**

**Reply:**

See reply to comment for **p6** above.

**p11: "The cloud detection sensitivity is here as high as 1.5"; "all optically thick clouds": : : Define what "high" and "thick" means (earlier in the manuscript).**

**Reply:**

The cloud detection sensitivity is clearly defined earlier in the manuscript (lines 396-399). However, since it represents an optical depth we have generally replaced the word "high" with "large" (and 'small' for the opposite case) throughout the text.

**p13, L495-500: Since specific orbits and satellites were not clarified, there's confusion here as to what was actually compared/validated. If it was equally applied to the morning and afternoon orbits (the wording leaves this open), one has to wonder how this would work because CALIOP operated in the afternoon orbit. How can morning cloud cover be "compared" to afternoon cloud cover, considering the significant diurnal cycle of clouds in most regions?**

**Reply:**

We have clarified this in the revised manuscript. This study only dealt with comparisons with afternoon satellites. This was clearly stated on page 8 lines 306-308 and the reason for restricting it to afternoon satellites have been mentioned several times above in various replies to comments and questions. Still, we have discussed also the morning satellite case in the Discussion section (lines 697-710) since CLARA-A2 is based on both afternoon and morning satellite data.

The reason that we still brought up the case of morning satellites at page 13 is explained by the fact that readers of this paper (and reviewers!) would most likely start wondering how these results will relate to morning satellite data (representing almost 40 % of the entire CLARA-A2 data record). Matchups between CALIPSO and morning satellites are possible but only near the high latitude of 70 degrees on both hemispheres where the orbital tracks crosses between the two satellites (as explained on lines 220-224 in the revised manuscript).
What maybe confuses the Reviewer is that we state that some comparisons had been done also for morning satellites. However, this relates to the results in the standard CLARA-A2 validation report (of which some were presented in the CLARA-A2 paper by Karlsson et al., 2017) and not to this particular study. We have added a discussion about the morning satellite matchups in Section 5 (lines 697-710) where we have also clearly explained that the discussed results for morning satellites was published in earlier papers. But we wanted to mention these results in relation to this discussion and especially pointing out the need for addressing this issue later (lines 705-710).

The last question here is very relevant and interesting. The diurnal cycle of cloudiness is of course leading to differences which makes a direct comparison of results difficult (even at the latitude band around 70 degrees where we have matchups from both afternoon and morning satellites). Anyhow, we can first conclude that in the night part of an afternoon orbit and in the corresponding night part of morning orbits we have exactly the same type of AVHRR

measurements (same spectral channels used). Thus, the only additional difference expected might come from diurnal cycle effects which probably are quite small for the dark part of the day at these high latitudes.

The largest differences are instead expected in the illuminated part of the day since we will then use AVHRR channel 3a (at 1.6 microns) for the morning satellites while AVHRR channel 3b (at 3.7 microns) is still used for the afternoon satellites. The comment on line 497 stating that we have seen good agreement here means that even for the illuminated case we have good correspondence between afternoon and morning satellites. For the region covered by morning matchups we do not see large differences with corresponding results from afternoon satellites. This is encouraging and it indicates that the two spectral channels provide more or less the same cloud screening information (while for cloud property estimations, like optical thickness, we expect much larger differences). We also think that at these high latitudes we will probably not be very much affected by the diurnal cycle in cloudiness. A short comment on this has been added on line 702.

The statement about the good agreement were not meant to be general in the global sense but just saying that, where we can inter-compare the data from the two orbit constellations, the agreement appears to be good. Additional studies are however needed to evaluate the global performance of morning satellites and we clearly indicate a way forward here (by using CATS data – see lines 705-710 in the revised manuscript).

**Language comments:**

**p1: "considerably" -> "considerable" Corrected**

**p1,l20: "were" -> "where": multiple occurrences throughout the manuscript Comment: We found 2 occurrences of this error (typos) in the manuscript. The reviewer is too critical here, in our opinion. The typos have been corrected.**

**p1: "geographically higher" -> use "surface elevation" instead? Reformulated.**

**p1,l23-l25: run-on sentence (multiple occurrences); at the very least use punctuation (in this case, a comma after "CDRs") to break it up. Better still, re-write. Reformulated (broken into two sentences)**

**p1: "sensor families": a bit unusual for science manuscript, consider revising "family"**

**Comment: OK, we have removed this term (despite often being used in the remote sensing community)**

**p1: add comma before "which" (in most cases; multiple occurrences)** **Comment: We tried our best to improve here but it is not always obvious where to use a comma.**

**p1: "four decades: : :" -> "four decades, which qualifies them to be used in climate: : :" Corrected**

**p2,L51-53: "Linked to this: : :" Unclear: efforts by whom? stringent with regard to? Reformulated**

**p2: The A-Train stands for "Afternoon constellation", not "Aqua Train" Corrected**

**p2: "a project being a part of" -> fix language Corrected**
**p2,l70-73: run-on sentence Corrected**
**p2,L91: MODIS: Introduce upon first occurrence Corrected**
**p3,l117: "including" -> "detecting" Corrected**
**p4, l121: "notice" -> "note" Corrected**
**p4,l148-165 and following: Avoid "you": Not only is this inconsistent with the style of this manuscript, but it is also not advisable for a science manuscript in general. This sounds more like a seminar or talk than a paper at this point in the manuscript. I recommend a complete re-write of this section, as well as a thorough discussion of the meaning of a "cloud mask" (see comment above).**

**Reply:** The use of "you" is removed and sentences are rephrased. Section three has been thoroughly redesigned (see reply to **general comment, part 4**, especially points 1 and 2 which affects this section.

**p4,l148: "areal extension" -> "areal extent" Corrected**
**p5,l177-179: Revise English, hard to understand Rephrased**
**p5,l184: "possibly punish AVHRR-based methods in an unfortunate and undeserved way: : :" - see comment above Rephrased**

**p5,L199: "of which single shots that were removed: : :" fix English Rephrased**

**p8,L314: "navigation" -> "geolocation"? Corrected**
**p10: "Validation results are probably underestimated" -> What does that mean?**

**Reply:** We have rephrased this sentence (lines 686-689) and also related this to the new Figure 1 explaining the matching geometry (see **general comment, part 4**, point 2).

**p10: "compared to if only showing results based" -> fix English Rephrased**

**FINAL REMARKS:**

- We are extremely grateful for the suggested editorial, syntax and language improvements. These are invaluable for non-native English writers like us!

- We also express our appreciation of the reviewer's large effort leading to this very detailed review.

---

## Author Comment (AC5)

**"Detailed characterisation of AVHRR global cloud detection performance of the CM SAF CLARA-A2 climate data record based on CALIPSO-CALIOP cloud information"**

**by**

**Karl-Göran Karlsson and Nina Håkansson, SMHI**

Note: All line numbers referred to below are relevant for the revised manuscript version written in Word change track mode and named "CLARA\_A2\_validation\_AMT\_2017\_version2\_tracked\_changes".

**Repeating general comments:**

The paper presents an unprecedented evaluation of satellite-based cloud climatology (CMSAF's CLARA-A2) against **CALIPSO/CALIOP** performed at the global scale. Despite some limitations of CALIOP dataset discussed in the paper, it is the only currently considerable reference for cloud retrievals covering oceans, polar regions and other areas of very sparse cloud observations and measurements. Such evaluation has become possible with the sufficiently long CALIOP dataset. The authors also present an analysis of the CLARA-A2 cloud detection sensitivity, i.e. the threshold in the cloud optical thickness (COT) above which the cloud detection algorithm detects more than 50% of clouds. Screening the CALIOP data with COT below the globally-averaged detection sensitivity allows for "more realistic" evaluation, i.e. taking into account the difference between the sensitivity of CALIOP (active sensor) and AVHRR (passive sensor). Therefore, the paper will be an important first step towards proposing described validation methodology for the list of standard validation activities performed before releases of new cloud climate data records.

While the content of the paper is novel, valuable and appropriate for the publication in AMT, the paper structure should be significantly improved. Finally, the paper has some grammar and language issues, which should be addressed. They are mostly related to the syntax, i.e. sentence length and inappropriate word order. Some examples are indicated in the following, but the whole manuscript should be revised.

**Reply:**

We thank the reviewer for this positive evaluation. We notice the request for a reorganization of the paper (also demanded by other reviewers) and we have done our best to accomplish this. We reply to all specific comments below.

**Repeating specific comment 1:**

The title of the paper is a bit misleading. "Detailed characterization" suggests that the evaluation of the CDR is more detailed than the standard one, e.g. provided in CLARA-A2 validation report. However, the collocations of AVHRR and CALIOP are limited to NOAA-18 and NOAA - 19, afternoon orbits and 10-year period only (from 30y+ of the CDR). Taking into account that one of the challenges in deriving CDR is stable performance in time, the evaluation presented in the manuscript cannot serve as an evaluation of CLARA-A2 CDR.

**Reply:**

Yes, we understand this remark and we agree that the validation presented here cannot be fully representative of a validation of the entire 34-year CLARA-A2 data record. But we still argue that the validation presented here is improved and more detailed than the validation (i.e., the CALIPSO-CALIOP part) presented in the CLARA-A2 validation report. The reason is the use of CALIPSO version 4 datasets (version 3 was used in the CLARA-A2 validation report) and the introduction of the new concept evaluating the cloud detection sensitivity which is the core topic of this paper. So we are quite confident that this is the best validation effort that can be done from existing reference data (lines 85-87), at least if requiring global coverage. The validation based on SYNOP data in the CLARA-A2 validation report indeed covers the full 34-year period but it cannot present a result that is globally valid in the same sense as the CALIPSO-CALIOP validation. We have emphasized this situation on lines 50-58.

As regards the collocations with NOAA-18 and NOAA-19, these are exactly the same as for the standard CLARA-A2 validation (i.e., same number of collocations, about 5000 orbits). However, in this study we exclude collocations with the morning orbits of NOAA-17, Metop-A and Metop-B since these are only possible over a narrow latitude band close to 70 degrees. Thus, we want to focus on the global performance and that can best be studied based on afternoon orbit data. The exact content of the entire validation dataset is now described in the new section 3.6.

The point about the necessity to evaluate the stability of a long-term data record is indeed an important aspect but also one of the most difficult ones to deal with. How can we find a suitable reference dataset of cloud observations with global coverage to perform this stability analysis? To be honest, there is no such reference dataset offering the required length and coverage of observations. The only candidate is surface (SYNOP) observations of cloudiness but they cannot fulfill the requirement of global coverage (e.g. oceanic and polar regions are largely not covered) as mentioned on lines 50-58. They also have their own quality problems (e.g., lack of knowledge of the thinnest cloud being observed, low quality at night-time and also hampered by being subjective in their character in that different observers have different opinions on how to interpret clouds and their coverage). Furthermore, the surface observation network has undergone rapid changes during the last decades due to automatization and this has caused problems in maintaining stable observation quality over time. With this background, we are of the opinion that there is no better reference than the 10-year CALIPSO dataset for evaluating the CLARA-A2 (and similar) satellitederived data records, despite the fact that it only covers about one third of the CLARA-A2 observation period. It offers the global coverage (only excluding some areas in close proximity to the poles) and a high and stable quality of observations. Estimating the stability is still a challenge but we hope that on a longer term also this aspect will be properly dealt with assuming that the era of active cloud lidar observations from space can continue (e.g., with new data from EarthCARE and CATS replacing CALIPSO and hopefully also data from new lidar missions beyond the lifetime of EarthCARE). This aspect is mentioned at the end on lines 801-809.

Finally, we have also changed the title to the following:

"Characterization of AVHRR global cloud detection sensitivity based on CALIPSO-CALIOP cloud optical thickness information: Demonstration of results based on the CM SAF CLARA-A2 climate data record"

**Repeating specific comment 2:**

Objectives of the study should be described better in the Introduction. In relation to (1), it should be clear if the aim is to present new methodology using a subset of CLARA-A2 as an example or to evaluate CLARA-A2.

**Reply:**

Yes, we have done that on lines 76-89 (see also the reply to 1). The new title also emphasizes the presentation of a new methodology more than the presentation of new CLARA-A2 validation results.

The study intends to provide revised or upgraded validation results (compared to the validation reports from the standard CLARA-A2 validation) with some extended or additional features (like the Cloud Detection Sensitivity). The revision is partly required by the upgrade of the available CALIPSO-CALIOP datasets and the results of the impact of this change are also included as one separate (or preparatory) objective of the study (described in sections 3.3 and 4.1).

**Repeating specific comment 3:**

The current discussion section is a mix of discussion remarks and conclusions. I recommend to separate the two. In the results' section, there are also interpretations, which are hypothetical (they often start with "we believe", "we claim") and should be moved to the discussion. Otherwise it is often difficult to judge which statements are really supported by the results achieved in this study.

**Reply:**

Yes, we admit this weakness of the current manuscript. We have followed the recommendation and included both a Discussion section (section 5) and a Conclusion section (section 6).

**Repeating specific comment 4:**

The analysis of detection sensitivity reveals some interesting non-expected results. One is that CLARA performance is not better at dark and warm ocean surfaces (L374-375). The hypothesis this is due to sampling and geometry of AVHRR and CALIOP FOVs needs more explanation. The problem was detected here, because it leads to unexpected results. However, how to measure a possible effect of this issue on results in other situations, regions, etc.? I would consider a separate section (or paragraph) in the discussion..

**Reply:**

Yes, we admit that this result deserves more attention. We also got a similar remark from the other reviewers. We have improved the description in three ways:

- 1. We introduced a short summary (first part of Section 3.2) of the underlying basic method of matching AVHRR and CALIPSO data. It seems the current referencing to the original paper by Karlsson and Johansson (2013) (which introduces the matching method) is not enough for a full understanding. We need to recapitulate the method's most important aspects also in this paper.
- 2. We added an illustration (new Figure 1) of how matched high-resolution AVHRR FOVs relate to the CALIPSO-CALIOP FOVs within a nominal AVHRR GAC pixel. The consequences for the matching of the two datasets are described in the second part of Section 3.2.
- 3. We expanded the discussion of these results in the new Discussion section (Section 5, lines 642-695). However, we believe that further studies on the full (global and local) impact of the differences of matched AVHRR and CALIOP FOVs could indeed deserve a paper on its own. Thus, we cannot dwell too much on this seemingly unexpected result since this would risk leading to a much too long paper. We only want to highlight the existence of this problem which has (in our view) been largely overlooked in many previous papers using CALIPSO-CALIOP data as the main validation source.

**Repeating specific comment 5:**

Is the cloud detection sensitivity a measure of CDR performance itself? There is no discussion if 0.225 signifies good or bad CLARA performance. One can imagine the same analysis (i.e. evaluation against screened CALIOP data), but with the estimated cloud detection sensitivity of, say, 0.5. Please elaborate on that. In addition, since the authors recommend the methodology to be widely used (e.g. in CFMIP), more detailed guidelines would be appreciated. For instance, when applied to different passivesensor-based CDRs, should the cloud detection sensitivity be always recalculated?

**Reply:**

Yes, even if it only concerns cloud detection performance, we believe that it is at least one very important piece of information for characterizing the entire CDR performance. Despite of the fact that it only deals with the cloud masking quality and not specifically with the quality of other parameters of CLARA-A2 (e.g. other cloud properties, surface albedo and surface radiation budget parameters), we also know that errors in cloud masking definitely will affect the

quality of other parameters derived further down-stream in the processing of a data record. For example, incorrect cloud screening (missed clouds) over dark surfaces will inevitably lead to an overestimation of surface albedos. Exactly how the uncertainty in cloud masking is propagating into the uncertainty of other parameters is yet to be determined in more details than what is done today. However, to better describe this is one of the challenges in the CM SAF project when preparing the next version of the CLARA dataset (CLARA-A3). But for the current CLARA-A2 dataset (and which could also relevant for other similar type of datasets), this new description of the cloud detection performance can be seen as one important step towards a better uncertainty description.

The question whether the average cloud detection sensitivity at (cloud optical thickness) 0.225 represents a good or a bad performance has no clear answer. This is because this study is the first of its kind proposing such a measure defined in exactly this way (as described in the paper). However, one indication that it is probably not too bad is that the COSP (Cloud Feedback Model Intercomparison Project (CFMIP) Observation Simulator Package) satellite simulator for ISCCP uses a global cloud optical depth threshold of 0.3 to describe the cloud detection ability of the ISCCP dataset.

However, this quantity can only be evaluated when and if it is later put in relation to corresponding values (computed in the same way) for other datasets (like datasets from MODIS Collection 6, PATMOS-X, ISCCP or ESA-CLOUD-CCI). We encourage such studies since we think that this measure of performance is a universal one which has nothing to do with AVHRR data in particular. Instead, it should be applicable to any other global cloud dataset based on passive satellite imagery. And, yes, it should always be recalculated for every new dataset to be evaluated (answer to last question). These cloud detection sensitivities could then be inter-compared between different data records. This is the main point in promoting this method as a universal method.

The value 0.225 is only a global average calculated for CLARA-A2 (or to be strictly correct, for the 2006-2015 period of CLARA-A2) and it should only be inter-compared and evaluated with corresponding global averages derived for other cloud datasets. In that sense, the question about what happens if using the value 0.5 is not relevant. More interesting would rather be to compare the results of the global distribution of the cloud detection sensitivity (new Figure 12) with corresponding distributions for other cloud datasets. This would be the most interesting aspect for use in a wider context since this would be able to reveal global differences (at a rather fine resolution) in performance for different algorithms and data records. Examples of such inter-comparisons are still rather few (with the GEWEX inter-comparison study by Stubenrauch et al. in BAMS July 2013 as the best example). A tentative repeated GEWEX inter-comparison study in the future could be imagined to include such global performance and

difference maps valid for the entire period of CALIPSO data. That would really show how all these data records perform if using CALIPSO-CALIOP as representing the truth.

We have included some of these clarifications and proposals/suggestions in the new Discussion and Conclusion sections (e.g., lines 627-640, lines 787-799 etc).

**Reply to short comments and editorial remarks:**

L50, "be very accurate to be able.." - please be more specific, e.g. referring to GCOS recommendations

**Reply:** We are of the opinion that the reference Ohring et al. (2004) explains exactly what "very accurate" means. Their discussion also involves references to GCOS recommendations. We don't want to expand the discussion further here, especially when considering the need to expand other sections as a consequence of other more serious requests from reviewers.

**L82, "FOV resolution" - field of view does not have a resolution, I would keep FOV and remove 'resolution' (or 'size' in other places in the manuscript)**

**Reply:** OK, we may have used the wrong terminology here. The field of view (or sometimes being denoted "Instantaneous Field of View) can be defined as "*The area on the ground that is viewed by the instrument from a given altitude at any time.*" So, yes, this area is not equivalent to a resolution. The resolution we are thinking of is rather linked to the diameter of the FOV (assumed to be circular or elliptic in shape). This diameter, in turn, is then often used as the resolution of the image grid or image matrix defining a satellite image. In that sense, there is often some sort of relation between the FOV (diameter) and an image resolution.

However, to just remove resolution (or size) does not solve the problem here. For example, the sentence

"AVHRR is measuring in five spectral channels (two visible and three infrared channels) with an original horizontal field of view (FOV) resolution at nadir of 1.1 km." cannot be written as

"AVHRR is measuring in five spectral channels (two visible and three infrared channels) with an original horizontal field of view (FOV) at nadir of 1.1 km".

From the definition, FOV is an area and the modified sentence is therefore still wrong.

We propose kind of a compromise here so that we do not have to change too much of the text. We propose to use the expression "FOV size" to denote the approximate diameter of the FOV area. This requires that we explain this interpretation the first time we use it. Thus, we have added the following lines 99-100 after the introduction of AVHRR measurements:

"The size is defined in this context as the approximate diameter (assuming a circular or elliptic shape) of the FOV and this definition will be used throughout this paper."

We hope that this explanation will be enough for the reader to understand when we talk about the different FOV sizes (e.g., 70 m, 330 m, 1 km and 5 km) in the remainder of the paper.

**L92, 'various parameters retrieval' - be more precise**

**Reply:** CLARA-A2 contains more than just cloud parameters. There are also surface radiation and surface albedo products. The description is expanded slightly to explain this (lines 113-121).

**L117-119, "Thus CALIOP products..." - please provide a reference for this statement**

**Reply:** This is also described in the earlier mentioned reference Vaughan et al., (2009). Thus, we repeat it here (line 141).

**L126-127, "...claiming that useful...seems to be available" - based on which results?**

**Reply:** We also got a question on this from another reviewer. We repeat the reply to that question below:

We admit that we do not have good support in the literature for stretching the useful upper limit of CALIPSO-derived COD to 5. However, in the description of the upgrade to CALIPSO-CALIOP version 4 it is also emphasized that previous cloud optical thicknesses in version 3 were generally underestimated. This is also clearly indicated in Figure 2 (new Figure 3) in the manuscript. Whether this increase entirely justifies moving the upper limit to 5 is still not clear.

We do have more indications from our own investigations that an adjustment of the upper limit seems possible. In a study related to a paper by Riihelä et al. (2017) we investigated the correlation between CALIPSO-estimated and CLARA-A2 estimated CODs over various surfaces (with snow surfaces over Greenland as the main target). However, when isolating the collocated results over ice free ocean surfaces at high latitudes (noting that over a dark surface also the AVHRR-based estimations should be more accurate), we could clearly see a good correlation between the two estimations up to about COD=5 (see figure below):

Although this is not a perfect illustration (not included in Riihelä et al, 2017, but maybe considered for a follow-up paper) it shows how CLARA-A2-estimated optical depths compare to CALIOP-estimated optical depths in the range 0-15. Over a dark ocean surface the majority of values agree pretty well but what is clear is that an increasing number of cases (for higher optical depths) CLARA-A2 values saturates at 100 for CALIOP-values exceeding approximately 4 (noticeable at top of the figure). This reflects the inability of CALIOP to provide reasonable optical thicknesses for optically thick clouds. But, we made the conclusion that values compare pretty well even up to an optical thickness of 4-5 and this was one of the reasons why we decided to use the CALIOP interval 0-5 for this particular study (for AMT).

It is this finding that made us to use the maximum limit of 5 in this particular study. Unfortunately, in the end, we did not include this part of the intercomparison in the finally published paper by Riihelä et al. (2017).

We propose that we keep the original maximum value of 5 in our plots but add a remark that values near this upper end are uncertain (lines 146-154). The upper limit is not crucial for the findings of our study since in most cases the cloud detection sensitivity is considerably lower than 5. Only for some positions over Greenland and Antarctica we approach these high values but whether the value is 3 or 5 here does not really matter since it deviates anyhow very much from the values found on other places (which is the main message).

The mentioned reference is the following:

Riihelä, A., Key, J. R., Meirink, J. F., Munneke, P. K., Palo, T., & Karlsson, K.-G. (2017). An intercomparison and validation of satellite-based surface radiative energy flux estimates over the Arctic. Journal of Geophysical Research - Atmospheres, 122(9), 4829–4848. https://doi.org/10.1002/2016JD026443

**L140-L145, If these improvements are relevant for the study, please explain them better**

**Reply:** We are of the opinion that the three selected changes are obviously important for this study and that no further comments are needed. Full information about all changes is given by the link given before on line 138. Here we only highlight three selected changes which we think are most important.

The first selected change (line 170) points at a general improvement of the fundamental cloud-aerosol-discrimination method. This method is, of course, crucial for the quality of CALIOP cloud information.

The second selected change (line 171) points at a special problem that previously was noted for cloud-aerosol discrimination over certain regions. This is also crucial for our validation study since it reduces the risks that regional features in our validation results are due to weaknesses of the underlying CALIOP data.

The third change (line 173) is important in that it offers an alternative method to take into account some of the inconsistencies between fine resolution and low resolution CALIOP datasets. This is discussed more in detail in Section 3.2 (lines 250-280) and in Section 3.3.

Thus, we keep the text as it is. In our opinion, to add extended text is more important for more serious review points.

**L150, "...how thin or thick..." - do you mean optically, in height?**

**Reply:** We mean optically thin or thick. We have added this for clarity on line 134 and on several other places in the manuscript.

L151, "The second aspect..." - something is wrong with the syntax, please rephrase

**Reply:** We have rephrased the text considerably (lines 177-185).

L192, The investigation if the method used by Karlsson and Johansson (2013) is still applicable to the new CLAY version should be listed as one of the paper objectives (i.e. already in the introduction). The results (L206-223) should be moved from this paragraph to the Section 4.

**Reply:** Yes, we agree. We made the following changes:

- 1. A short sentence on the upgrade to CALIPSO-CALIOP version 4 and the impact of this change is added to the Introduction (lines 82-83).
- 2. We added a sentence (lines 297-298) explaining that the results of the preparatory study are given in (new) section 4.1.
- 3. The current description of results of the preparatory study is moved to (new) Section 4.1.

**L249, why 'CLARA-A2 cloud masks', i.e. in plural?**

**Reply:** Rephrased as follows (line 348-349):

"The results are computed by treating both CLARA-A2 and CALIOP cloud masks as binary values, ....."

**L250, "This approximation is acceptable.." - provide a reference**

**Reply:** Well, the simple answer is that there is no estimation of sub-pixel cloudiness in the CLARA-A2 case. Thus, we actually have no other choice. We have removed this sentence to avoid any confusion.

L288, Why 50% is an appropriate threshold for the cloud detection probability?

**Reply:** We do discuss this in the text (in the sub-sequent sentences after L288, which are lines 395-404 in the revised manuscript). The argument is that above this threshold, by definition we detect more clouds than we miss (in the statistical sense). A cloud detection scheme that misses more clouds than it detects is not an efficient scheme. So, a minimum requirement should be that it at least should detect 50 %. This is our point. If this is not a satisfying answer we wonder: How would you otherwise describe or define a measure or the cloud detection sensitivity? A threshold anywhere below the 50 % level can be questioned since the scheme then would generally fail here by missing more clouds than it detects. So, in our opinion, the 50 % level is the most sensible choice.

**L326, "...but we still believe..." - what if the authors are wrong?**

**Reply:** It is difficult to answer this question. In the ideal world you would always have an infinite number of samples to make the perfect statistical estimation. But in reality there are always limitations. The best thing to do here is probably to remove this rather speculative sentence and instead highlight that there might still be locations where estimations are uncertain. We reformulate the sentence in the following way (lines 432-433):

"...with only a few exceptions mainly located over the Pacific Ocean. In these locations the uncertainty in the results might be expected to be larger than for the rest of the globe."

L328 and L349, Please consider giving different section names. These two are not very informative.

**Reply:** OK, we suggest the following:

**4.2 Results based on original CALIOP cloud masks compared to results excluding contributions from very thin clouds**

**4.3 Additional validation scores**

L369, "This contributes..." - it's not clear what is meant. Please rephrase.

**Reply:** We suggest the following (line 644-645, also adjusting to new Figure numbers to reflect the new Figure 1):

"This explains to a large extent the fairly low values of the Kuipers' score over these regions (Figure 10) leading to a slightly different distribution of results in comparison to the Hitrate (Fig. 7)."

**L361-404 – It would be easier to follow the text divided in paragraphs**

**Reply:** OK, we have sub-divided the text into several paragraphs.

**L381, "We first conclude..." - is it based on actual results or it is a hypothesis?**

**Reply:** This follows from the actual geometries of the matched AVHRR GAC and CALIOP FOVs. We have commented this further in relation to discussion of the additional figure demonstrating the matching geometry (see point 2 in the reply to **specific comment 4**).

**L406-407, Wrong syntax, please rephrase**

**Reply:** Rephrased sentence (lines 577-579):

"We have here presented validation results after having 'removed' (in the sense of interpreting them as cloud-free cases) all clouds with smaller optical depths than the cloud detection sensitivity parameter. This leads undoubtedly to a clear improvement of results compared to if only showing results based on the original CALIOP cloud mask (i.e., comparing Figs. 5 and 7)."

L407, "...is undoubtedly a clear improvement", please explain why?

**Reply:** We think this is rather obvious when comparing results from the unfiltered (old Figure 4, new Figure 5) and the filtered case (old Figure 6, new Figure 7). Hitrates are considerably higher which is emphasized in section 4.3. The problem with the unfiltered case is highlighted in lines 332-334 in the original manuscript. Since CALIOP is a much more sensitive sensor than AVHRR there should be a certain fraction of clouds that are detectable by CALIOP but which never will be detected by any AVHRR-based method. The filtering approach is one way of trying to compensate for this.

We think we can rely on the current text and discussion here. No changes are made.

**L436-438, Please explain better, preferably in a separate paragraph in the Discussion**

**Reply:** We have done that (please see point 3 in the reply to **specific comment 4**).

Figure 11, it would be useful to have a different color scale (e.g. as in previous figures), with a shift between colours at 0.225. Otherwise it is difficult to see the 'edge' at 0.225

**Reply:** We definitely agree. This was one of the changes we had planned even before achieving review comments to the discussion paper. Here is our proposed new Figure 1 with blue colours denoting places where the detection sensitivity value is lower than the average value of 0.225 and where red colours show places where values are higher than the average. This new plot is also better in showing the high values over the poles (which were just masked out in grey colours in the previous figure).